# Numerical modelling of permafrost spring discharge and open-system pingo formation induced by basal permafrost aggradation

Mikkel T. Hornum[1,2], Andrew J. Hodson[1,3], Søren Jessen[2], Victor Bense[4], Kim Senger[1]

[1]Department of Arctic Geology, The University Centre in Svalbard (UNIS), N-9171 Longyearbyen, Norway.
[2]Department of Geosciences and Natural Resource Management, University of Copenhagen, 1350 Copenhagen K, Denmark.
[3]Department of Environmental Science, Western Norway University of Applied Sciences, N-6856 Sogndal, Norway
[4]Department of Environmental Sciences, Wageningen University, 6708PB Wageningen, Netherlands.

*Correspondence to*: Mikkel T. Hornum (mth@ign.ku.dk)

## Abstract

In the high Arctic valley of Adventdalen, Svalbard, sub-permafrost groundwater feeds several pingo springs distributed along the valley axis. The driving mechanism for groundwater discharge and associated pingo formation is enigmatic because wet-based glaciers are not present in the adjacent highlands and the presence of continuous permafrost seem to preclude recharge of the sub-permafrost groundwater system by either a sub-glacial source or a precipitation surplus. Since the pingo springs enable methane that has accumulated underneath the permafrost to escape directly to the atmosphere, our limited understanding

of the groundwater system brings significant uncertainty to predictions of how methane emissions will respond to changing climate. We address this problem with a new conceptual model for open-system pingo formation wherein pingo growth is sustained by sub-permafrost pressure effects, as related to the expansion of water upon freezing, during millennial scale basal permafrost aggradation. We test the viability of this mechanism for generating groundwater flow with decoupled heat (1D-transient) and groundwater (3D-steady-state) transport modelling experiments. Our results suggest that the conceptual model

represents a feasible mechanism for the formation of open-system pingos in lower Adventdalen and elsewhere. We also explore the potential for additional pressurisation, and find that methane production, and methane clathrate formation and dissolution, deserve particular attention on account of their likely effects upon the hydraulic pressure. Our model simulations also suggest that the generally low-permeability hydrogeological units cause groundwater residence times to exceed the duration of the Holocene. The likelihood of such pre-Holocene groundwater ages is supported by the geochemistry of the pingo springs, which

demonstrates an unexpected sea-ward freshening of groundwater, potentially caused by a paleo-subglacial melt water "wedge" from the Weichselian. Whereas permafrost thickness (and age) progressively increases inland, accordingly, the sub-permafrost melt water wedge thins and less unfrozen freshwater is available for mixing. Our observations imply that millennial-scale permafrost aggradation deserves more attention as a possible driver of sustained flow of sub-permafrost groundwater and methane to the surface, because, although the hydrological system in Adventdalen at first appears unusual, it is likely that

similar systems have developed in other uplifted valleys throughout the Arctic.

# 1    Introduction

Sub-permafrost groundwater systems are highly inaccessible and so their investigation usually relies on sparse data (van der Ploeg et al., 2012). However, cold regions increasingly become hydrogeologically active after surface warming and associated permafrost degradation. This implies, for example, an increased outflow of deeper groundwater to rivers and lakes (Bense et al., 2012), increased rates of biogeochemical processes (Grosse et al., 2016), and potentially increased fluxes of methane or other compounds into the surface environment and atmosphere (Schuster et al., 2018). The surface discharge of sub-permafrost groundwater is currently exemplified by springs in the high Arctic (Andersen et al., 2002; Grasby et al., 2012; Haldorsen et al., 1996; Williams, 1970). If conditions are favourable, spring outflow may instead freeze below the active layer and initiate the growth of an ice-cored hill or pingo. By definition, this would classify as an open-system pingo because of the open connection to the sub-permafrost groundwater (Liestøl, 1996). Considerable methane stocks may exist below continuous permafrost and where no such springs exist, the only rapid escape route goes to the ocean, where methane oxidation prevents much of it from reaching the atmosphere (Mau et al., 2017; Myhre et al., 2016). Where springs do exist, however, sub-permafrost methane may escape directly to the atmosphere, contributing significantly to the total landscape methane emissions (Betlem et al., 2019; Hodson et al., 2019).

The hydrogeological mechanisms causing the sustained flow of sub-permafrost groundwater to surface springs remain elusive (Scheidegger et al., 2012). Earlier it was proposed that subglacial meltwater from underneath warm-based ice sheets or glaciers would sufficiently recharge a sub-permafrost aquifer (Demidov et al., 2019; Liestøl, 1977; Scheidegger et al., 2012; Scheidegger and Bense, 2014). However, in regions of continuous permafrost lacking warm-based glaciers or other groundwater recharge pathways, such models do not seem applicable (Ballantyne, 2018; Grasby et al., 2014; Woo, 2012). An alternative model to explain the existence of perennial springs in such environments is that hydraulic head gradients in the sub-permafrost hydrogeological system are maintained by artesian pressure generated by past or current aggradation of basal permafrost (Fig. 1). This would remove the need to invoke groundwater recharge from the surface as spring outflow derives from relict groundwater. Furthermore, it might explain the formation of emergence-related open system pingos in coastal lowlands (Burr et al., 2009; Yoshikawa and Harada, 1995).

In this paper, we use field data from Adventdalen, Svalbard, in combination with numerical modelling of heat and groundwater flow to evaluate the hypothesis that perennial spring flow through continuous permafrost can be driven by sub-permafrost artesian pressure produced by basal permafrost aggradation. Alternative causes of anomalous pressures are also discussed, including overpressure remaining from past perturbations (e.g., glacial loading) or contemporary processes such as equilibration of groundwater density contrasts and gas production. The investigation of the above hypothesis is based on decoupled heat and groundwater modelling, but in the discussion, this is combined with inverse analyses of spring geochemistry from a series of open-system pingos within the valley. A 1D heat transfer model forced by reconstructed paleo-temperatures serves to simulate Holocene ground temperatures and permafrost development in the valley floor of Adventdalen. By considering the expansion of water upon freezing, the simulated rates of present permafrost aggradation must cause a loss

in aquifer volume equivalent to a recharge rate. This apparent recharge defines the only inflow term to a steady-state 3D groundwater model that simulates the present state of the sub-permafrost groundwater system. The modelling results are then discussed in relation to the hydrochemical observations of the pingo spring waters.

## 2        Conceptual model of permafrost aggradation-driven pingo formation

When permafrost aggrades into the ground, water in the pore space freezes, and hence expands. At shallower depths this is evident from ice lenses and other types of visible ground ice, resulting in ground heave, but from a certain depth downwards (e.g., ~ 5 m in Adventdalen: Gilbert et al., 2018), these cryostructures are no longer observed (French, 2017). Instead, the lithostatic pressure prevents ground heave and the ice expansion induces an overpressure (with regard to the hydrostatic pressure) on the sub-permafrost groundwater, especially where pressures cannot dissipate in an efficient manner. The process is well known from closed-system pingos, where groundwater is enclosed by aggrading permafrost and expelled to the surface from a closed talik (i.e., a perennially unfrozen part of the permafrost) (Mackay, 1998).

In contrast to closed-system pingos, an open-system pingo is sourced from a body of groundwater that is not enclosed by frozen ground. Liestøl (1977) suggests that an open-system pingo-spring can be driven by recharge from subglacial melting of warm-based glaciers. Scheidegger et al. (2012) developed a coupled model of transient permafrost formation and showed how hydraulic heads can maintain spring outflow for millennia even when permafrost is aggrading (disregarding ice expansion). In our study, we test the hypothesis that permafrost aggradation itself can generate such excess head.

Figure 1 illustrates our conceptual model for open-system pingo formation by basal permafrost aggradation and presents the additional conditions that also have to be met. The assumed starting point is a coastal landscape with no permafrost and a subsurface consisting of a hydrogeological unit in which hydraulic pressures dissipate poorly (Fig. 1a). Figures 1b and 1c illustrate that a negative shift in the surface energy balance results in permafrost aggradation. Freezing pressure is induced at the freezing front resulting in hydraulic head gradients. The shift in the surface energy balance may occur due to a drop in the mean annual air temperature (MAAT) (Figs. 1b–c), a regressing shoreline (Fig. 1c), rapid erosion (not illustrated), or a combination of any of these. Close to the sea, groundwater flows towards the shoreface, but at some distance inland, higher advective heat transfer, associated with higher groundwater velocities, prevents frozen ground formation. As a consequence, groundwater may flow through a talik that perforates the permafrost (i.e., a through-talik) towards the surface along the most hydraulically conductive path, resulting in a spring (or pingo) (as modelled by Scheidegger et al., 2012). Freeze-up of the through-talik is further restricted if permafrost aggradation lowers the melting point by increasing pressure and/or salinity. Figure 1d illustrates that when ground temperatures are eventually in equilibrium with the MAAT, permafrost aggradation has stagnated, and so groundwater flow to the pingos has ceased. Due to the lack of advective heat transfer, the through-talik might therefore close and, if so, irreversibly de-activate pingo spring discharge. However, salinity may keep the through-talik open, in spite of no-flow.

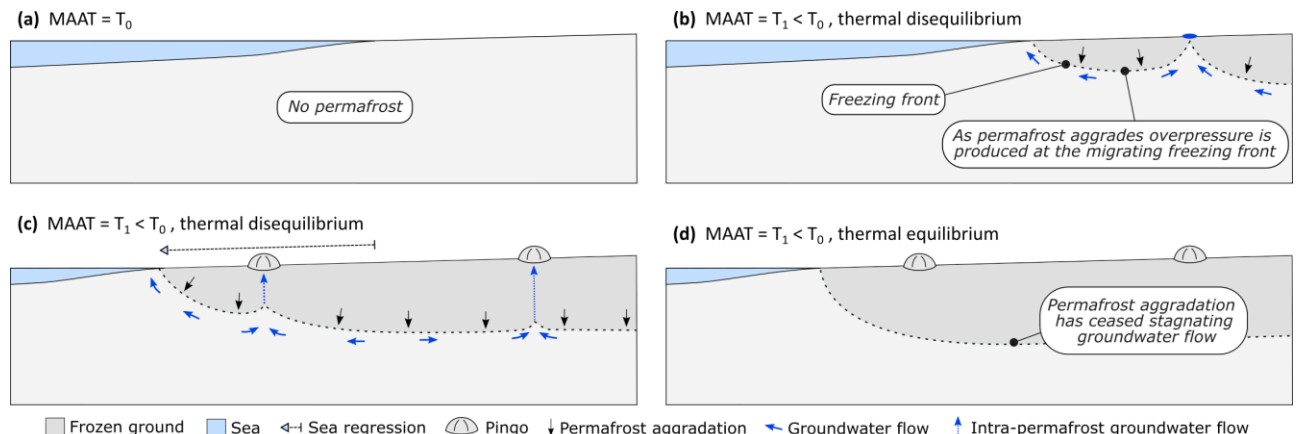


**Figure 1** Conceptual model of pingo formation driven by permafrost aggradation in a low-permeable system. **(a)** No permafrost is present. **(b)** and **(c)** A negative shift in the surface energy balance results in permafrost aggradation. Freezing pressure is induced at the freezing front and this results in hydraulic head gradients. At some distance inland, higher advective heat transfer, associated with higher groundwater

velocities, prevents the ground from freezing and groundwater flows to the surface where a pingo forms at the spring. **(d)** The ground is in thermal equilibrium with the MAAT, permafrost aggradation has stagnated, and groundwater flow to the pingos has ceased.

## 3      Study site

Adventdalen is a ~ 30 km long glacially cut valley in central Spitsbergen, Svalbard (Fig. 2a). Its high Arctic climate is characterised as polar tundra (Kottek et al., 2006) and the ground is dominated by continuous permafrost (Humlum et al.,

2003). Because of the dry climate with a mean annual precipitation of ~ 200 mm (Hanssen-Bauer et al., 2018) only a few, small glaciers exist today, the largest being Drønbreen, 9 km long and up to 200 m thick.

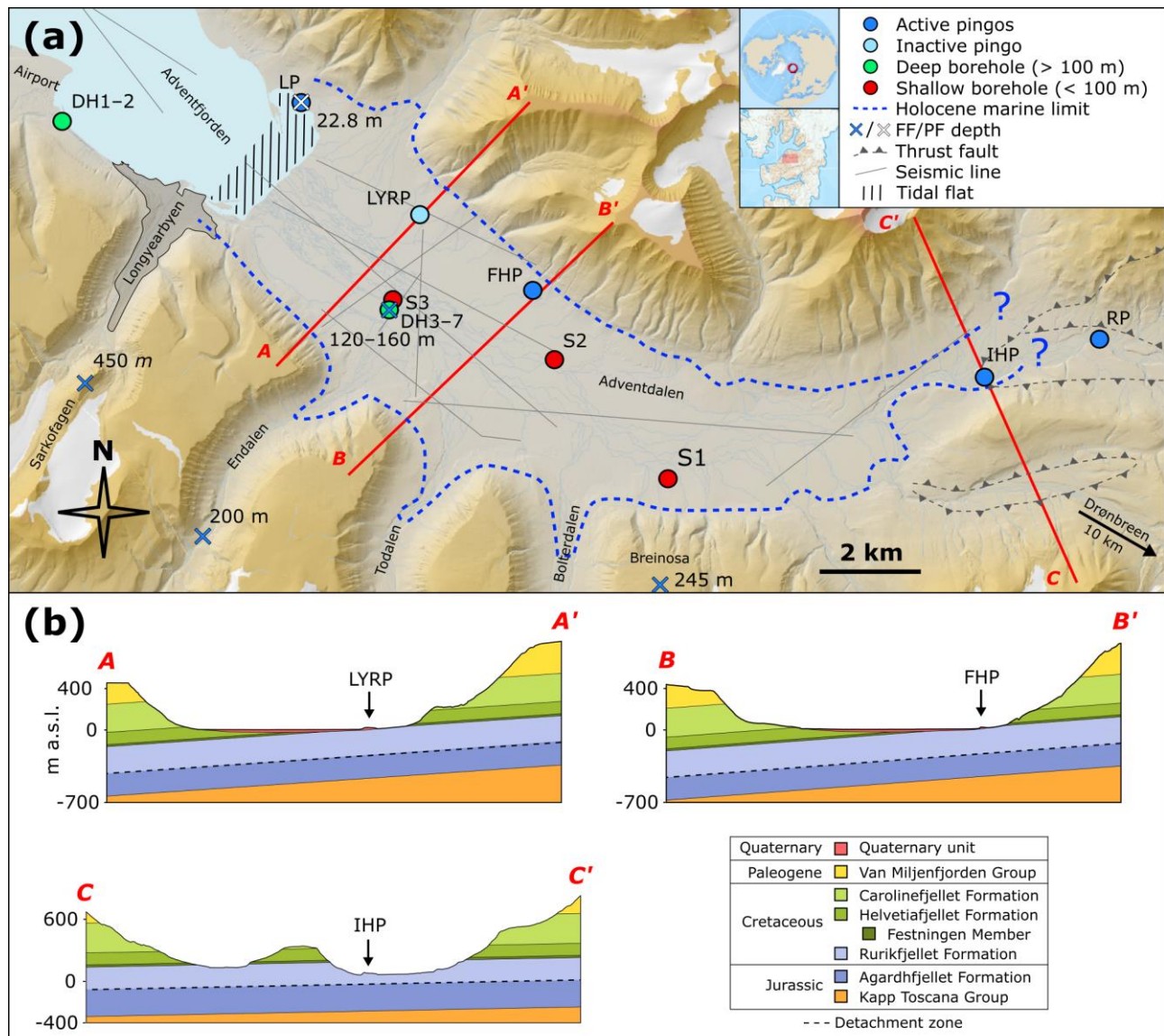

**Figure 2 (a)** Map of Lower Adventdalen with the location of data resources, pingos and the Holocene marine limit. LP = Lagoon Pingo, LYRP = Longyear Pingo, FHP = Førstehytte Pingo, IHP = Innerhytte Pingo, RP = River Pingo. Core logs from boreholes S1–3 and D1–D7 (respectively, Gilbert et al., 2018, and Olaussen et al., 2020, and references therein), seismic lines (Bælum et al., 2012, and unpublished commercial lines from Norsk Hydro) and a geological map (Norwegian Polar Institute, 2019) were used to build the geological model (Fig. 5a) (see details in Hornum, 2018). Permafrost depth measurements at the Sarkofagen, DH4 and Breinosa sites are from Liestøl (1977), Braathen et al. (2012), and Christiansen et al. (2005), respectively. The freezing front depth at LP is from Yoshikawa and Harada (1995). Data used to develop the map including topography, glacial extent, and fluvial network by courtesy of Norwegian Polar Institute (2019). **(b)** Geological cross sections constructed based on the resources mentioned above. The Quaternary unit overly well-consolidated sedimentary

strata of pre-Cenozoic age (i.e., Cretaceous or older). See Sect. 3.1 for a (hydro)geological description of the layers shown in the cross sections A, B and C.

## 3.1    Geology and hydrogeology in Adventdalen

In Adventdalen, fine-grained Quaternary sediments (< 70 m thick Gilbert et al., 2018) overly pre-Cenozoic, well-consolidated
sedimentary strata (Fig. 2b), which are likely the best described in Svalbard largely thanks to the Longyearbyen CO2 Laboratory Project (Olaussen et al., 2020, and references therein). Together, all these units form a low-permeability groundwater system.

The sedimentary strata gently dip in a west-south-westerly direction (Fig. 2b) and the youngest strata are thus found closest to Longyearbyen and the present-day coastline. The uppermost unit, the Early Cretaceous Carolinefjellet Fm (~ 300 m
thick, Fig. 2b), consists of sandstone intercalated with shale beds and overlies the fluvial sandstones of the Helvetiafjellet Fm (59–72 m thick, Grundvåg et al., 2019). The Festningen Member of the Helvetiafjellet Fm (11–18 m thick), comprises fractured sandstones and is relatively hydraulically conductive, as proven by cross-well water injection tests (Bælum et al., 2012). Directly below is the ~ 450 m thick Janusfjellet Subgroup (Fig. 2b) that comprises two shale-dominated units, the Rurikfjellet (201–232 m thick, Grundvåg et al., 2019) Agardhfjellet Formations (253–264 m thick, Koevoets et al., 2018). A regional
detachment zone with extensive fracturing and swelling clays propagates near the boundary of these two formations (Braathen et al., 2012) and is considered a major barrier to fluid migration (Olaussen et al., 2020). The tectonic disturbances, as reflected in Festningen Sandstone and the detachment zone, make it possible that minor secondary permeability development is present elsewhere in the stratigraphy. This would have implications for groundwater movement in the system, which is otherwise predominantly through rocks with low hydraulic conductivity (Table 3).

The glacier advances during the last glacial maximum (LGM) on Svalbard (~ 20 ka (ka = $10^3$ yr before present)) are thought to have completely eroded any pre-existing glacial deposits in the inner fjords of Svalbard (Elverhøi et al., 1995) and the Quaternary succession in Adventdalen thus postdates this event. Optically stimulated luminescence (OSL) datings in three cores from Adventdalen (S1, S2, and S3, Fig. 2a) support this (Gilbert et al., 2018). The Quaternary succession overlies fractures in the underlying sedimentary strata, possibly explained by glacier load/unload and freeze-thaw processes (Benn and
Evans, 2010; Gilbert et al., 2018), as well as the significant tectonic uplift in the area. The presumed high hydraulic conductivity of this fracture zone is in contrast to the generally low-permeability of the Quarternary succession. The lowermost Quaternary unit is a < 5 m thick, subglacial lodgement till deposited during the last Weichselian glacial advance. Overlying the till, a shallowing-upwards trend is observed in the gradually changing succession of marine muds (< 20 m), pro-deltaic to deltaic muds and very fine-grained sands (< 35 m), tidally influenced (saline) fluvial fine-grained sands (< 35 m), and aeolian loess
deposits (< 5 m). This reflects Holocene progradation of the present delta-system (Cable et al., 2018; Gilbert et al., 2018).

The sub-permafrost groundwater often has hydraulic heads above hydrostatic, which results in artesian conditions. This is evident from the occurrence of several pingo springs and from artesian outflow from deep boreholes (> 100 m below ground level (b.g.l.)) at the confluence of Bolterdalen and Adventdalen (Malte Jochmann, pers. comm.; SNSK, Unpublished

Reports SN1981_008 and SN1983_004). Based on artesian outflow during a drilling experiment nearby, Braathen et al. (2012) deduce a hydraulic pressure of 18 to 23 bars at a depth of ~ 175 m b.g.l. in well DH4 (Festningen Member, Fig. 2), which corresponds to a hydraulic head of 9 to 60 m above hydrostatic when the potential pressure effects of the dissolved gasses are excluded. Significant under-pressure ~ 50 bar below hydrostatic is recognised in deeper stratigraphical layers (~ 800 m b.g.l.) and is believed to relate to glacial unloading, extensive fracturing and matrix expansion (Birchall et al., 2020; Braathen et al., 2012; Wangen et al., 2016). The low pressure indicates hydrogeological separation from upper groundwaters immediately beneath the permafrost, in line with Sr-isotope analyses from the drill cores (Huq et al., 2017; Ohm et al., 2019).

### 3.2 Late Weichselian and Holocene climate history

Air temperatures on Svalbard have been continuously recorded at Longyearbyen Airport since 1911 (Nordli et al., 2014). Until the 1990s, the 30-yr running mean of MAATs was -6.0 °C, while it has increased to -4.2 °C in the period 1988 to 2017 (Hanssen-Bauer et al., 2018). For the entire temperature record, the mean summer air temperature (MSAT) has consistently been 10 °C warmer than the MAAT (on the 30-yr scale, Førland et al., 1997). Further back in time, Holocene mean summer sea temperatures (MSST) in and around Svalbard are relatively well constrained by fossil-based temperature reconstructions (Fig. 3a, van der Bilt et al., 2018; Hald et al., 2007; Mangerud and Svendsen, 2017). Mangerud and Svendsen (2017) point out that the MSST is essentially identical to the MSAT.

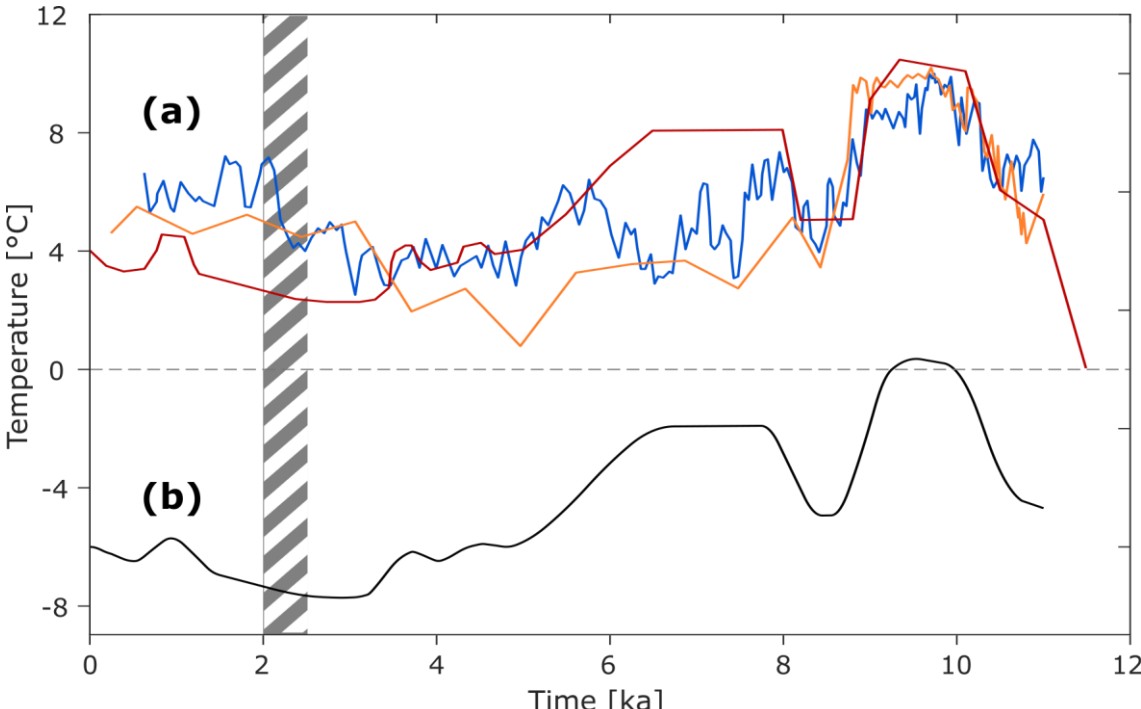

**Figure 3** Holocene temperature reconstructions in and around Svalbard. Dashed grey area indicate time of minimal driftwood arrival (Farnsworth et al., 2020). The unit of the time axis is ka = 10³ years before present. **(a)** MSST curves (= MSAT, see text). Red line from

Mangerud and Svendsen (2017). Orange line from Hald et al. (2007). Blue line from van der Bilt et al. (2018). **(b)** MAAT used in this work. Based on Mangerud and Svendsen (2017) and Førland et al. (1997) (see Sect. 4.3.3).

At the last glacial maximum (LGM) occurring at ~ 20 ka, glaciers covered all fjords in Spitsbergen (Ingólfsson and Landvik, 2013). By ~ 11.5 ka the central parts of Isfjorden were glacier-free and its inner tributaries followed at ~ 11.2 ka (Forwick and Vorren, 2011; Gilbert et al., 2018). If any glacier ice remained in Adventdalen after then, it was certainly gone by ~ 10 ka when non-glacial sediments were deposited in the valley head (Lønne and Nemec, 2004). The eustatic sea level rise caused by northern hemispheric deglaciation during Late Quaternary and Early Holocene was surpassed on Svalbard by

the rate of its postglacial rebound. The Holocene marine limit (HML) ranges from ~ 20 m above sea level (a.s.l.) in the northwestern part of Spitsbergen to ~ 90 m a.s.l. in the central part (Forman, 1990). In Adventdalen, raised marine sediments suggest a HML of ~ 62 m a.s.l. and ~ 70 m a.s.l. in the inner and outer part of the valley, respectively (Lønne and Nemec, 2004). Although not well constrained, the relative sea level is estimated to have fallen pseudo-exponentially until reaching close to present levels ~ 5 ka (Lønne and Nemec, 2004). Despite the uncertainty of sea level fall, the fjord retreat and the

associated exposure of new valley floor are relatively well constrained by absolute dating presented in previous work (Table 1).

**Table 1** Absolute ages from Adventdalen constraining delta propagation.

| Site/Event[a] | Distance to modern delta front [m] | Dating method | Dating material[b] | Age of valley floor [yr BP][b] |
|---|---|---|---|---|
| LP | ~340 | C-14 | Peat | > 240 (±50)[I] |
| LYRP | ~3650 | C-14 | Driftwood | < 2650 (±55)[II] |
| S3 | ~4300 | Quartz OSL | Quartz | ~ 3000 (±200)[III] |
| FHP | ~6400 | C-14 | Shell | < 6980 (±70)[IV] |
| S2 | ~7500 | C-14 | Plant matter | < 9178 (±153)[III] |
| [c]HML | ~16000 | C-14 | Shell | < 10025 (±160)[V] |

[a]See Fig. 2 for site locations. [b]Depending on the dated material and the host sediment, the dating indicates minimum, approximate or

maximum valley floor ages. [c]HML = Holocene marine limit. Compiled from [I]Åhman (1973), [II]Svensson (1970), [III]Gilbert et al. (2018), [IV]Yoshikawa and Nakamura (1996) and [V]Lønne and Nemec (2004).

### 3.3    Permafrost, pingos and the apparent lack of groundwater recharge

In the valley floor of Adventdalen, the permafrost thickness ranges from ~ 0 m at the coast to ~ 200 m inland. In the adjacent mountains it increases to > 450 m (Christiansen et al., 2005; Humlum et al., 2003; Liestøl, 1977). One observation of the

freezing front depth at Lagoon Pingo (Harada and Yoshikawa, 1996) and permafrost depth observations at well DH4 (Braathen et al., 2012), Endalen, Sarkofagen (both Liestøl, 1977), and Breinosa (Christiansen et al., 2005) support this regional characterisation (Fig. 2a). Mountain permafrost is presumably of Weichselian age, while permafrost in valleys postdates the Late Holocene (Humlum, 2005). The continuous permafrost and a lack of warm-based glaciers in the adjacent highlands most likely hinder subglacial recharge to the sub-permafrost aquifer due to the impervious frozen ground (Burt and Williams, 1976;

Haldorsen et al., 2010; McCauley et al., 2002; Walvoord and Kurylyk, 2016; Woo, 2012).

Sub-zero temperatures are a prerequisite for permafrost formation and its thickness essentially reflects equilibration to the geothermal heat flow (French, 2017). In the Adventdalen area, measurements of the geothermal gradient range from 0.02 °C m$^{-1}$ in the highlands to 0.03 °C m$^{-1}$ in the valley bottoms (Betlem et al., 2019; Liestøl, 1977).

Open-system pingos are a common feature in Svalbard and Adventdalen (Humlum et al., 2003), where five of them are distributed parallel to the valley axis (Fig. 2a). The three outermost; Lagoon, Longyear and Førstehytte Pingos, are all located on the northeastern side of the valley. All three have formed in Quaternary marine muds (Yoshikawa and Harada, 1995) and close to the sedimentary bedrock boundary (Fig. 2b). In the valley head, two additional pingos are located close to the boundary of HML; Innerhytte and River Pingos. They have formed in shales above a major fault and are situated in the valley floor in the path of the river Adventelva (Yoshikawa and Harada, 1995). With the exception of the seemingly inactive Longyear Pingo, groundwater has discharged perennially from springs located at the pingos, at least since the earliest recordings in the 1920s (Orvin, 1944). However, visible spring outflow (or winter icing) at Førstehytte Pingo was not observed from summer 2018 until October 2019. Presumably, during that time, groundwater flow through the permafrost continued, but instead of discharging to the surface, groundwater froze within the pingo and added to its growth. From Lagoon Pingo, Yoshikawa and Harada (1995) reported a spring discharge of 0.013 to 0.016 L s$^{-1}$, Hodson et al. (2019) estimated ca. 0.3 L s$^{-1}$ during the 2017 summer, and we measured 0.26 L s$^{-1}$ in August 2019. At Innerhytte Pingo, Yoshikawa and Harada (1995) measured a discharge rate of 0.11 L s$^{-1}$, which is somewhat smaller than Liestøl's (1977) estimate in 1976 of ~ 1 L s$^{-1}$. Based on our own "by the eye" estimates involving no quantitative measurements, the discharge rate at Førstehytte Pingo was in orders of 0.1 L s$^{-1}$ when visited in fall of 2015, 2016 and 2017, and less than 0.01 L s$^{-1}$ when rediscovered in October 2019.

## 4      Method - Numerical modelling

Several numerical model codes are capable of simulating coupled heat and groundwater transport in permafrost environments (Grenier et al., 2018). However, the benchmark models do not consider the overpressure produced by ice expansion. To include this process, we decoupled the modelling of heat flow from sub-permafrost groundwater flow and made use of a source term to mimic the pressure effects of permafrost aggradation (Fig. 4). Ground temperatures and permafrost dynamics were simulated in the vertical dimension using a custom made finite-difference 1D transient heat transfer model (hereafter just 1DHT) coded in MATLAB R2019b (MathWorks®, 2019). The 1DHT model script is publicly available at DOI:10.5281/zenodo.4240594. Groundwater flow was modelled 3D steady-state with MODFLOW in the groundwater modelling software GMS 10.4 (AQUAVEO™, 2019). The connection between the two models was the permafrost aggradation rates simulated by the 1DHT, which determined the only water source for the groundwater model. The decoupled approach does not simulate advective nor lateral heat transport. The potential limitations will be assessed in the Discussion (Sect. 6.2.1).

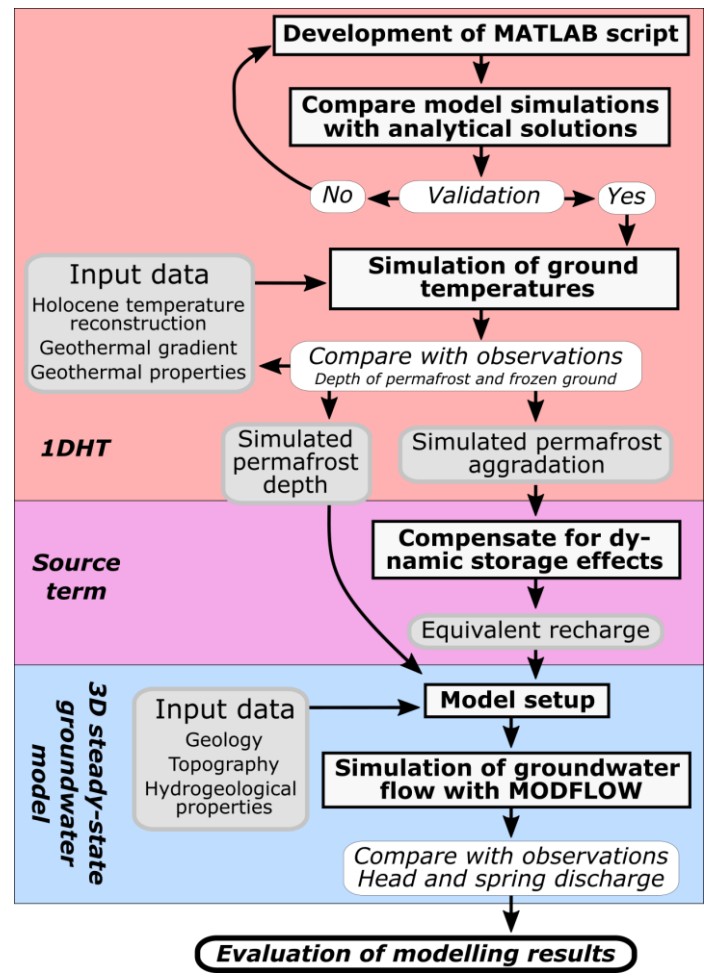


**Figure 4** Schematic overview of the inner workings of the decoupled heat and groundwater model. Model setup, calculations and algorithms are indicated with sharp corners and bold text. Validation and comparison of simulations and observations are indicated with round corners and italics. Input and output data are indicated with rounded corners, grey background and normal text.

## 4.1    Heat flow

Heat flow was modelled one-dimensionally in the vertical dimension, which is a common approach for permafrost models (Riseborough et al., 2008). Following from Fourier's law and conservation of energy, the 1D conductive heat flow equation states that:

$$\frac{\delta T}{\delta t} = \frac{k_e}{\rho_e \cdot c_e} \cdot \frac{\delta^2 T}{\delta z^2} \tag{1}$$

where $T$ is temperature [K], $t$ is time [s], $z$ is distance [m], and $k_e$, $\rho_e$, and $c_e$ are the effective values of the thermal conductivity

[W (m K)$^{-1}$], density [kg m$^{-3}$], and specific heat capacity [J (kg K)$^{-1}$], respectively. The term $\frac{k_e}{\rho_e \cdot c_e}$ equates to $\alpha_e$, the effective

thermal diffusivity [$m^2$ $s^{-1}$]. As heat transfer was modelled one-dimensionally, simulating advective heat transfer was not possible. The ratio of advection to conduction heat transfer rates may be quantified by a Peclet number (Bergman et al., 2011):

$$P_{e_L} = \frac{v\,L}{\alpha_e}$$ (2)

where $v$ is the pore water velocity and $L$ is the characteristic length.

In the case of a saturated medium, heat will be conducted through a matrix of solids (i.e., sediment or rock) and liquid water, ice or a mixture. The effective thermal parameters were assumed independent of temperature, but the fractions of water and ice changed between the solidus and liquidus temperatures, $T_S$ and $T_L$ respectively. The fraction of liquid water within the pore space, $f_w$, was determined using a smoothed step function (same approach as Mottaghy and Rath, 2006):

$$f_w = \begin{cases} \exp\left(-\left(\frac{T-T_L}{w}\right)^2\right) & if \quad T < T_L \\ 1 & if \quad T > T_L \end{cases}$$ (3)

where $w$ [K] determines the shape of the freezing curve. For this work $w \approx 0.96$, implying that $T_S = -2$ °C and $T_L = 0$ °C. Bonacina and Comini (1973) and Mottaghy and Rath (2006) note that the exact shape of the freezing curve is of little importance for the calculated temperatures, but that a smoother function generally improves the performance of a numerical model due to a more efficient convergence of the numerical approximation. The total fractions of soil or rock ($F_s$), water ($F_w$), and ice ($F_{ice}$) are described respectively as; $F_s = 1 - n$, $F_w = f_w \cdot n$, and $F_{ice} = n - F_w$ with $n$ being the total porosity. The

effective thermal conductivity was calculated as root-square-mean, as done by Mottaghy and Rath (2006) and Govaerts et al. (2016). When temperature change occurs between $T_S$ and $T_L$, freezing or thawing results in the release or absorption of latent heat, $L = 333.6$ kJ $kg^{-1}$ (Mottaghy and Rath, 2006). The latent heat of fusion was included in the expression of the equivalent volumetric heat capacity [J ($m^3$ $K^{-1}$], $C_{eq}$ (same approach as Govaerts et al., 2016):

$$C_{eq} = c_e \cdot \rho_e = F_s \cdot \rho_s \cdot c_s + F_w \cdot \rho_w \cdot \left(c_w + \frac{\delta f_w}{\delta T} \cdot L\right) + F_{ice} \cdot \rho_{ice} \cdot \left(c_{ice} + \frac{\delta f_w}{\delta T} \cdot L\right)$$ (4)

where subscripts $s$, $w$, and $ice$ indicate parameters of soil or rock, water, and ice, respectively.

     To validate the model code we compared simulations with two analytical solutions; Neumann's solution as presented by Carslaw and Jaeger (1959) and Mottaghy and Rath (2006); and an analytical solution of a step change in temperature neglecting latent heat effects as presented by Carslaw and Jaeger (1959) and Eppelbaum et al. (2014). The model code was able to reproduce the analytical results with root-mean-square errors of respectively $1.1 \cdot 10^{-2}$ and $1.3 \cdot 10^{-5}$ and these numbers

were regarded to represent an acceptable level of accuracy for the given purpose. The model code validation is described in detail in the Supplement.

## 4.2      Groundwater flow

Following from Darcy's law and the conservation of mass, the 3D groundwater flow equation fundamental for groundwater modelling can be described as (Fitts, 2002):

$$K_x \frac{\delta^2 h}{\delta x^2} + K_y \frac{\delta^2 h}{\delta y^2} + K_z \frac{\delta^2 h}{\delta z^2} + Q_N = S_s \frac{\delta h}{\delta t}$$ (5)

where $x$, $y$ and $z$ are distances [m] in the three dimensions, $K$'s are hydraulic conductivities [m s$^{-1}$] in those dimensions, $h$ is the hydraulic head [m], $Q_N$ is a term representing any potential sink or source [m$^3$ s$^{-1}$] (i.e., recharge, seepage, etc.), $S_s$ is the specific storage, and $t$ is time [s]. In this work, groundwater flow was modelled as a steady-state implying that the right-hand side of Eq. (5) equals 0. However, in the discussion of the model simulation results, hydrodynamic storage effects resulting from the glacial loading and unloading are considered.

Groundwater flow was simulated with MODFLOW, which solves the 3D groundwater flow equation with the finite-difference method (McDonald and Harbaugh, 1988). We approximated the pressure build-up from the simulated rate of permafrost aggradation, $R_{PF}$, by considering that it must correspond to some equivalent recharge rate (or source term, $Q_N$ in Eq. 5), $REq$. Assuming no expansion or compression of the matrix, $REq$ is specifically proportional to $R_{PF}$, the total porosity, $n$, and the expansion of water upon freezing, $X_w$:

$$REq = R_{PF} \cdot n \cdot X_w \qquad \text{[m s}^{-1}\text{]} \qquad\qquad (6)$$

$$Q_{REq} = REq \cdot A \qquad \text{[m}^3\text{ s}^{-1}\text{]}$$

where $Q_{REq}$ is equivalent to the source term $Q_N$ in Eq. (5), and $A$ is an area [m$^2$].

## 4.3    Setup and boundary conditions

### 4.3.1    Geological model

In order to define proper geothermal and hydrogeological properties, a geological model of the subsurface in Adventdalen was built. Towards northwest, the model covers the tidal flat (Fig. 2a), but no other sea-covered areas were included. Elsewhere, the horizontal model boundary was a simplified outline of the HML. The lower boundary is at 300 m b.g.l. (Fig. 5a). We used data from boreholes, seismic lines (see locations and references on Fig. 2a), and a geological map with DEM provided by the Norwegian Polar Institute (2019). The general workflow was to map relevant geological boundaries in 3D with the petroleum industry software Petrel v2016 (Schlumberger©, 2019), then use them to build geological layers with the TIN and SOLIDS editor functions GMS 10.4 (AQUAVEO™, 2019). For the 1DHT model, the geology was simplified into one-dimensional columns for a total of 12 zones of the model area. The zones were defined as follows: The age of the valley floor (Table 1) was used to infer isochrones of valley floor exposure with intervals of 1000 yrs (Fig. 5a). The isochrones defined boundaries between the zones and the names of the zones (i.e., the zone located between the 5 ka and 6 ka isochrones was named zone 5-6, Fig. 5b). The area between isochrones 9 ka and 10 ka was divided into two zones (9-10a and 9-10b) to incorporate geological variation.

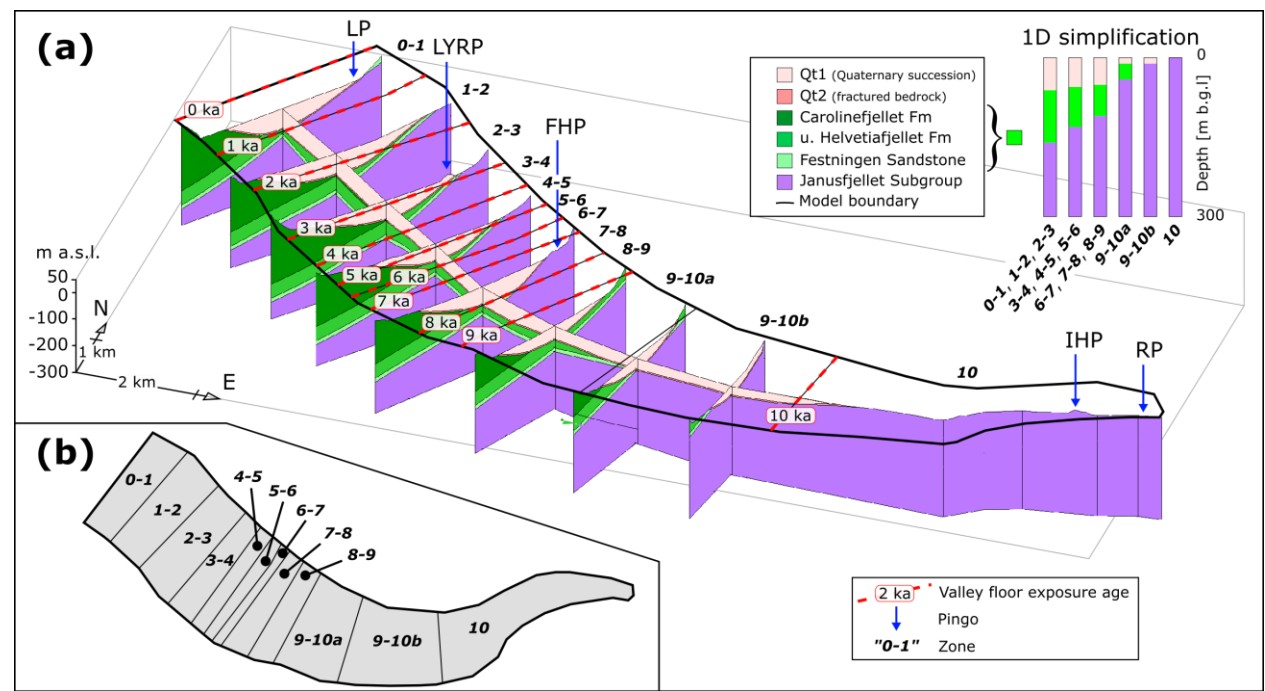

**Figure 5 (a)** 3D geological model of the subsurface below the valley floor in Adventdalen and vertical 1D simplifications (inset in upper right corner) below zones of the model area. The former determines the hydrogeological properties in the groundwater model (Table 3), whereas the latter determines the geothermal properties in the 1DHT model (Table 2). The domain of the 1DHT model extends to 1000 m b.g.l. (not shown). Deeper than 300 m b.g.l., geothermal properties were defined as for Janusfjellet Subgroup. The sea retreat reconstruction was inferred from absolute datings (Table 1) and is illustrated by valley floor exposure isochrones (red dashed lines). Pingos are indicated with blue arrows. **(b)** Vertical view of the model area showing the zonation. The aforementioned isochrones defined the zone names so that the valley floor exposure age of a zone is apparent from its name (i.e., zone 0-1 became sub-aerially exposed between 1 and 0 ka, zone 1-2 between 2 and 1 ka, etc.).

### 4.3.2 Geothermal and hydrogeological properties

Due to the sparse data available from the field area, geothermal and hydrogeological properties of the lithologies in the model domain (Tables 2 and 3) were largely based on the available literature. The considerable contrast between the thermal properties of water and ice implied that porosity was the most important parameter for permafrost growth, and realistic minimum, intermediate, and maximum values were therefore defined for the 1DHT model (Table 2). The permafrost base is presently located within the upper two thirds of the Janusfjellet Subgroup (Figs. 2b and 5a). Estimated burial depths and thicknesses of overlying units (Grundvåg et al., 2019; Marshall et al., 2015) indicate that this strata has been buried to maximum depths between 2150 to 2600 m b.g.l. corresponding to effective vertical stresses between 34 to 41 MPa (assuming a rock density of 2.6 kg m$^{-3}$ and hydrostatic equilibrium). Different studies on the compressibility of lithologically and age equivalent rocks in the North Sea thus suggest porosities between 0.08 to 0.3 (Burland, 1990; Okiongbo, 2011; Skempton, 1969; Yang and Aplin, 2004). We therefore used this range in our modelling experiments. An exception to the purely literature-

based values is the sandstone units of which the matrix porosity and vertical permeability, $\kappa_v$, were measured as part of the Longyearbyen CO$_2$ Laboratory Project (Olaussen et al., 2020, and references therein). The small-scale horizontal
permeability, $\kappa_h$, for sandstones is typically a factor two higher than $\kappa_v$ (Domenico and Schwartz, 1998). The horizontal hydraulic conductivity, $K_h$, was therefore calculated using the measurements of $\kappa_v$ by Braathen et al. (2012) as

$$K_h = C_{Kh/Kv} \cdot \frac{\kappa_v \cdot \rho_w \cdot g}{\mu} \tag{7}$$

where $C_{Kh/Kv}$ is the conversion factor (i.e., 2 for this work), $\kappa_v$ is permeability [m$^2$], $\rho_w$ is the density of water [kg m$^{-3}$], $g$ is the gravitational acceleration [m s$^{-2}$], and $\mu$ is the dynamic viscosity of water [kg (m s) $^{-1}$]. The range of the hydraulic
conductivity values of the Carolinefjellet and Helvetiafjellet formations (Festningen Sandstone not included) was defined by the 25 %, 50 % and 75 % percentiles of a log-logistic probability fit of the measured values (details provided in the Supplement). Ranges of hydraulic conductivity for the fluvio-deltaic succession were based on literature values from Fitts (2002). For the remaining bedrock units we also regarded the influence of fractures (Singhal and Gupta, 2010). All within these ranges, three sets of hydraulic conductivity defined the different model scenarios Sc1–3$x$ (Table 3). For steady-state
simulations, the porosity does not affect the net groundwater fluxes (i.e., discharge and Darcy fluxes, Eq. 5). To evaluate the pore water velocities, however, the effective porosity is of importance. We used the same values of effective porosity for all groundwater model simulations (Table 3).

**Table 2** Geothermal material properties used in the heat transfer model.

| Material | Thermal conductivity [J (yr m K)$^{-1}$] | Specific heat capacity [J (kg K)$^{-1}$] | Density [kg m$^{-3}$] | Thermal diffusivity [m$^2$ s$^{-1}$] | Total porosity [m$^3$ m$^{-3}$] | | |
|---|---|---|---|---|---|---|---|
| | | | | | Min | Intermediate | Max |
| Water | [I]1.77·10$^7$ | [I]4180 | [I]1000 | 1.34·10$^{-7}$ | - | - | - |
| Ice | [I]7.06·10$^7$ | [I]2100 | [I]917 | 1.16·10$^{-6}$ | - | - | - |
| Silty sand (Qt) | [II]1.58·10$^7$ | [I]850 | [III]2400 | 2.46·10$^{-7}$ | [IV]0.3 | [IV]0.4 | [IV]0.5 |
| Sandstone | [II]7.88·10$^7$ | [II]900 | [III]2600 | 1.01·10$^{-6}$ | [V]0.06 | [V]0.1 | [V]0.15 |
| Shale | [II]4.73·10$^7$ | [II]800 | [III]2600 | 7.21·10$^{-7}$ | [VI]0.08 | [VI]0.19 | [VI]0.3 |

Density and thermal properties compiled from [I]Williams and Smith (1989), [II]Robertson (1988), and [III]Manger (1963). Porosities from [IV]Fitts (2002), [V]Braathen et al. (2012), and based on works by [VI]Burland (1990), Grundvåg et al. (2019), Marshall et al. (2015), Okiongbo (2011), and Yang and Aplin (2004) (see text for more detail).

**Table 3** Properties of the hydrogeological units used in the groundwater model.

| Hydrogeological unit | Lithology | *Hydraulic conductivity* [m day$^{-1}$] | | | *Effective porosity* |
|---|---|---|---|---|---|
| | | Sc1$x$ | Sc2$x$ | Sc3$x$ | All |
| [I]Qt1 | Clay, silt and sand | 10$^{-4}$ | 10$^{-3}$ | 10$^{-2}$ | 0.4 |
| [I]Qt2 | Heavily fractured bedrock | 10$^{-2}$ | 0.1 | 1 | 0.4 |
| [III]Carolinefjellet Fm | Sandstone | 2·10$^{-4}$ | 5·10$^{-4}$ | 10$^{-3}$ | 0.1 |
| [III]u. Helvetiafjellet Fm | Sandstone | 2·10$^{-4}$ | 5·10$^{-4}$ | 10$^{-3}$ | 0.1 |
| [I, II]Festingen Sandstone | Fractured sandstone | 5·10$^{-2}$ | 7.5·10$^{-2}$ | 0.1 | 0.1 |
| [I, II]Janusfjellet Subgroub | Shale | 5·10$^{-4}$ | 7.5·10$^{-4}$ | 10$^{-3}$ | 0.1 |
| [I, II]Detachment zone | Fractured shale | 5·10$^{-3}$ | 7.5·10$^{-3}$ | 10$^{-2}$ | 0.1 |

Property values based on [I]Fitts (2002) and [II]Singhal and Gupta (2010) or evaluated from [III]Braathen et al. (2012) (details provided in the Supplement).

### 4.3.3 1D transient heat transfer model (1DHT)

The model domain contained 12 columns, each 1000 m long and consisting of 500 cells with a height of 2 m. One column was associated to each of the model area zones and the geothermal properties were defined according to the associated geological 1D simplifications (see insert on Fig. 5a). Deeper than 300 m b.g.l., the properties were that of the Janusfjellet Subgroup. The simulation run time was defined by the valley floor age inferred for that zone (Fig. 5), so that, for zone 0-1 the simulation period was 0.5 to 0 ka, for zone 1-2 it was 1.5 to 0 ka, etc. For zone 10, the simulation period was 10 to 0 ka. The initial ground temperature distribution followed the geothermal gradient reported by Liestøl (1977) (0.025 °C m$^{-1}$) from a surface temperature of 0 °C. At any subsequent time, the lower boundary condition was defined from the same geothermal gradient resulting in a basal temperature change of less than 0.65°C. The upper boundary condition was defined by the Holocene MAAT curve presented in Fig. 3b. Assuming that the present 10 °C difference between MAAT and MSAT (Førland et al., 1997) was alike for the entire Holocene, we constructed this curve (Fig. 3b) by subtracting 10 °C from the MSST curve by Mangerud and Svendsen (2017) (Fig. 3a). As illustrated on Fig. 3a, their MSST curve is largely in agreement with MSST temperature reconstructions from west and southwest of Svalbard (van der Bilt et al., 2018; Hald et al., 2007). We chose to rely on Mangerud and Svendsen (2017) because; a) their curve is more local to our field area than the alternatives and; b) the suggested timing of the Holocene thermal minimum at ~ 3 to 2 ka is in agreement with the maximum of perennial or semi-permanent land fast sea ice at ~ 2.5 to 2 ka inferred from the minimal occurrence of dated driftwood (Dyke et al., 1997; Farnsworth et al., 2020; Funder et al., 2011). Furthermore, their curve is better supported by geomorphological evidence of glacier dynamics (Farnsworth et al., 2020).

### 4.3.4 Groundwater model

For the groundwater model, each grid cell measured 100 by 100 by 5 m (*x y z*) and their hydrogeological properties were defined from the geology (Fig. 5a, Table 3). The model domain was defined from the geological model therefore covering an elongated ground surface area of 59 km$^2$ that is < 4 km broad and ~ 18 km long (Fig. 5b). Frozen ground was considered impervious and cells shallower than the simulated freezing front depth were thus de-activated. The lower boundary was at 300 m b.g.l. The fjord was simulated with a general head of 0 m a.s.l. and the conductance was determined according to the hydraulic conductivity (Table 3). The area assigned with these boundary conditions comprised 24 cells located within the uppermost layer at the northwestern end of the model domain. The MODFLOW drain package was used to simulate pingo springs. Because the cells located at the springs were inactivated, the drains were assigned to the uppermost active cells located closest to springs, but within the conductive Festningen Sandstone if present in the underlying stratigraphy (i.e., Lagoon and Førstehytte Pingos, Fig. 5a). Drain levels were set according to spring elevations (i.e., 1, 20, 65, and 77 m a.s.l. for Lagoon, Førstehytte, Innerhytte, and River Pingos, respectively). The simulated springs were able drain more water than the cells they were situated in could transmit. That is, the conductance was set high enough not to restrict any discharge. Except for the fjord and the springs, all outer model boundaries were assigned no-flow conditions.

The only source of water in the groundwater model was defined from the basal permafrost aggradation rate simulated
by the 1DHT model and assigned as recharge to the uppermost active cells in the model domain. To compensate for the lack
of dynamic storage effects in the steady-state model, we applied a moving time-average to the simulated basal permafrost
growth (or decay) before calculating the recharge equivalent (Eq. 6). The time window of the moving average was based on
the possible range of the adjustment time, $t_a$, which is the time needed for fluids to redistribute to a pressure perturbation (e.g.,
Neuzil, 2012; Šuklje, 1969):

$$t_a = l^2 \, S_s \, K^{-1} \tag{8}$$

where $l$ is half of the shortest dimension of the system (the characteristic length), $S_s$ is the specific storage, and $K$ is the
hydraulic conductivity. We found $t_a$ to be shortest in the vertical dimension, but assumed that hydraulic pressures could only
dissipate in the horizontal dimension after the formation of continuous permafrost no earlier than 6 ka (Humlum, 2005; this
research). Specifically, we estimated the horizontal $t_a$ to be between 20 and 19000 yrs. To quantify this, we used a
characteristic length of 1 km. For $S_s$, we used a matrix compressibility of $7 \cdot 10^{-10}$ to $7 \cdot 10^{-8}$ Pa$^{-1}$ (based on common estimates for
fractured rocks, e.g., Domenico and Mifflin, 1965; Domenico and Schwartz, 1998; Fitts, 2002) yielding a $S_s$ of $7 \cdot 10^{-6}$ to $7 \cdot 10^{-4}$ m$^{-1}$ (in line with literature values, c.f. Singhal and Gupta, 2010). For $K$, we used the values estimated for the dominating
geological unit (Janusfjellet Subgroup, Table 3). The time window used to compensate for dynamic storage effects were
defined from the above, but no longer than the age of permafrost (i.e., 6000 yrs or less).

To represent the uncertainty of how permafrost aggradation affects sub-permafrost groundwater flow, we simulated
nine different scenarios that were defined by having three sets of values for the two fundamental parameters in any
combination; hydraulic conductivity (Scenarios Sc1–3$x$, Table 3) and equivalent recharge (Scenarios Sc$X$a–b, values
calculated as described above and by Eq. 6). The nine scenarios are all labelled Sc$Xx$ where $X$ and $x$ indicates the minimum,
intermediate or maximum value sets of hydraulic conductivity and equivalent recharge, respectively. We further simulated a
tenth scenario that takes additional pressure sources into account.

## 5    Results

### 5.1    1DHT model results

The direct output from running the 1DHT model code was matrices containing the temperatures throughout the model domain
for each time step in the simulation period. For each time step, the permafrost and freezing front depths were calculated by
interpolating the depth at which the associated temperatures occurred. The greatest phase change rate in the 1DHT model
occurs at a temperature of -0.7 °C (following from Eq. 3) and it was therefore the progression rate of this isotherm that was
used for the calculation of the equivalent recharge (by net pore space loss, Eq. 6). Hereafter, 'permafrost aggradation' therefore
means the downwards progression rate of this isotherm (-0.7 °C), although this is not entirely congruent with the thermal
definition of permafrost (ground perennially below 0 °C, French, 2017). Since phase change occurs over a temperature range,

the porosity used to calculate the equivalent recharge was taken as a weighted mean of the porosities in the corresponding cell range, where the weight was proportional to the phase change rate.

### 5.1.1     Permafrost and freezing front depth

In Fig. 6, the simulated Holocene ground temperature development in Adventdalen is exemplified by the growth of frozen ground and permafrost in zones 2-3, 7-8 and 9-10b when using the intermediate porosity values (Table 2). For the oldest part

of the model area (most inland), an early occurrence of frozen ground was simulated during the Early Holocene cooling of 9 to 8 ka (Figs. 3 and 5). However, due to the subsequent warming at 8 to 6.5 ka, frozen ground was thawed and not re-established until ~ 6.5 ka. Since this time, deep ground temperatures were simulated to be cooling until present. The pattern was identical for simulations with lower and higher porosities (not shown) although the depths were different (as illustrated on Fig. 7a).

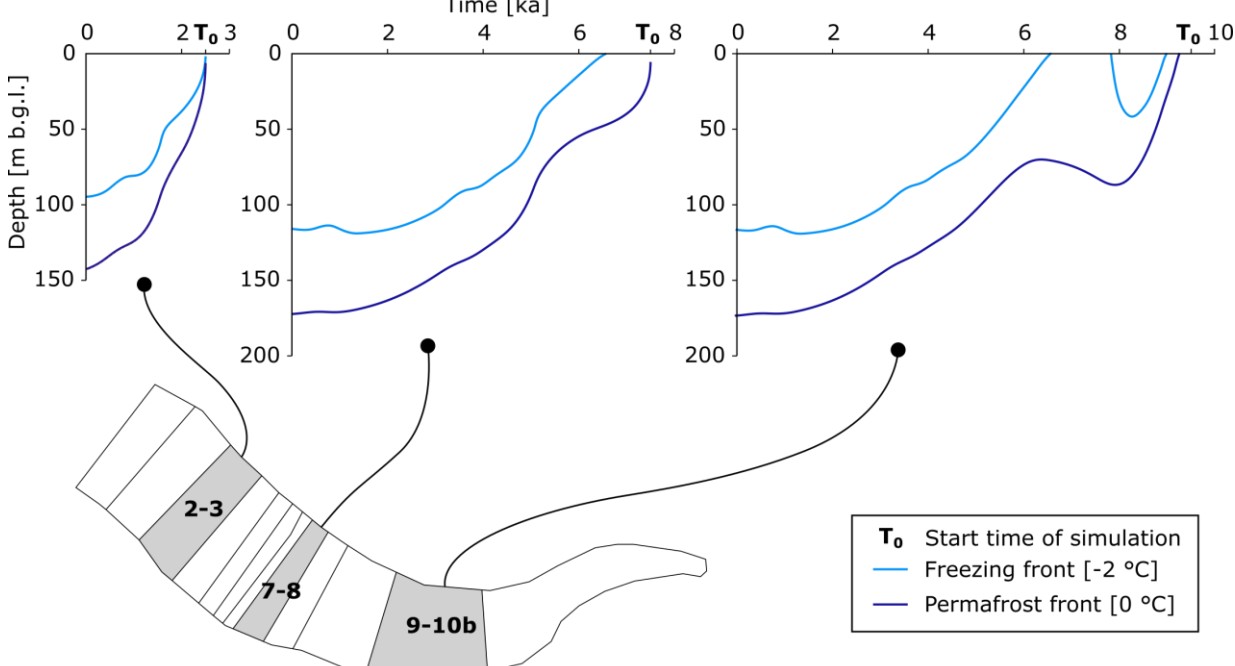

**Figure 6** Development of simulated freezing front and permafrost front depths from zones 2-3, 7-8 and 9-10b of the model area (see Fig. 5). These simulation results derive when using the intermediate porosity values (Table 2). Note that completely frozen ground does not establish permanently until ~ 6.5 ka.

Depending on the scenario, the 1DHT model simulated present day permafrost and freezing front depths of respectively 165 to 184 and 110 to 124 m b.g.l. at distances further than 6 km from the delta front (Fig. 7). Closer to the delta front both

isotherms are located at shallower depths and decrease to 58 to 65 m b.g.l. and 31 to 34 m b.g.l. The porosity plays an important role for the temperature development due to latent heat of the water (ice) filling the pore space and because water and ice account for the minimum and maximum thermal diffusivities in the model domain, respectively. In panel (a) of Fig. 7, an uncertainty field is drawn (shading) as derived from the applied porosity range (Table 2). The upper and lower edge of the

shaded area corresponds to the maximum and minimum porosity. The line corresponds to the intermediate porosity (Table 2).

In panel (b), the simulated present permafrost aggradation rate is plotted by point symbols. The shaded area was drawn by applying the moving time-average correction defined using the estimated adjustment times (see Sect. 4.3.4, Eq. 8) and indicates the values used to calculate the equivalent recharge rates in panel (c). Here, the line indicates intermediate values of recharge added to the groundwater model, while the edges of the shaded area represent the lower and upper estimates. The scarcity of ground temperature observations does not allow for model calibration, but observations both at Lagoon Pingo (freezing front)

and well DH4 (permafrost) agree relatively well with simulations.

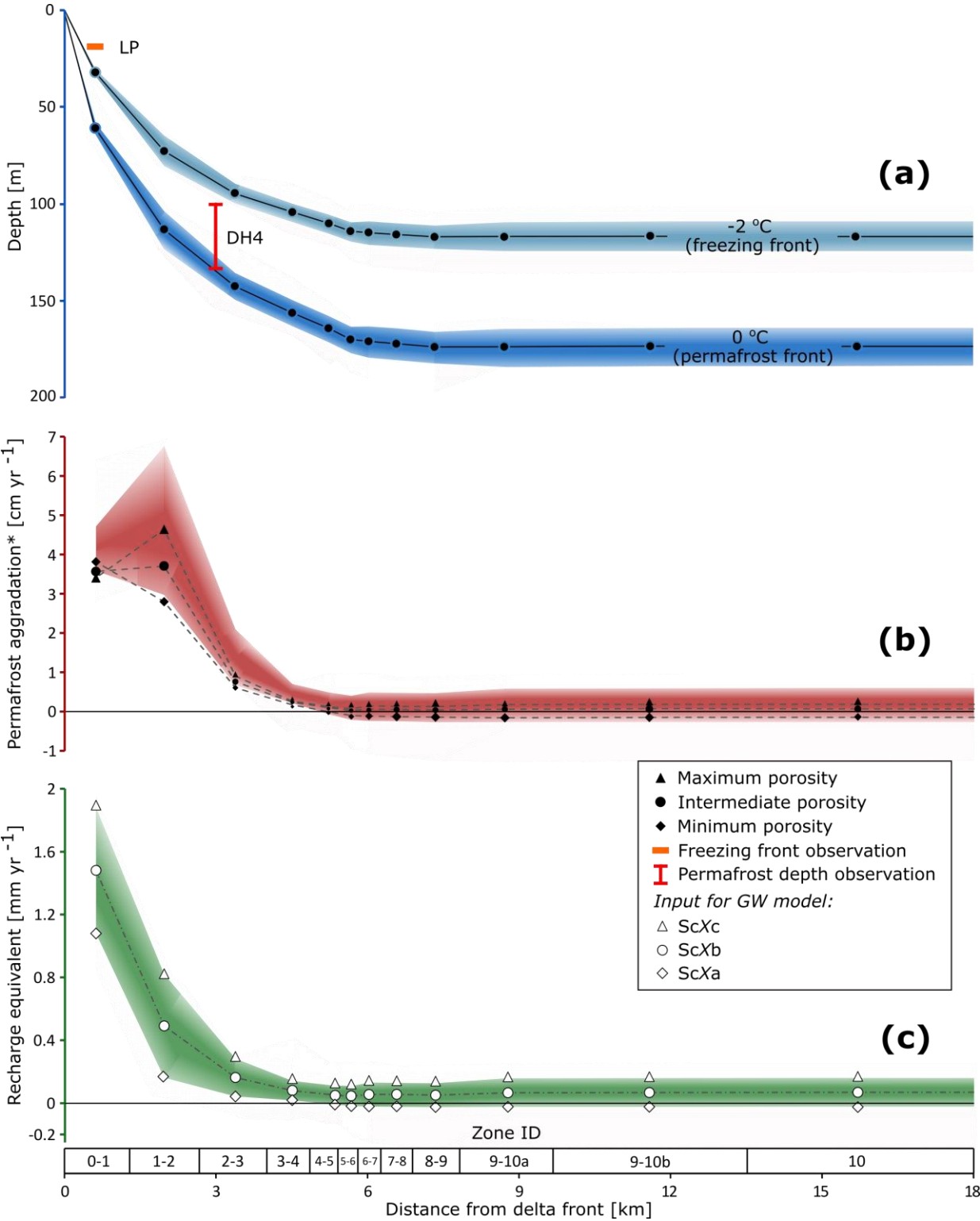

**Figure 7** (previous page) Present-day permafrost conditions as simulated by the 1DHT model. **(a)** Freezing front and permafrost depths. The upper and lower edge of the shaded area corresponds to the maximum and minimum limits of the porosity range, while the curve and the points corresponds to the intermediate (Table 2). Permafrost depth and freezing front depth observations are from Braathen et al. (2012) and Harada and Yoshikawa (1996), respectively. **(b)** Rate of permafrost aggradation. The point symbols represent the present permafrost aggradation, while the shaded area is drawn by applying a time-moving average as described in Sect. 4.3.4. and represents the values used to calculate the equivalent recharge rates. *It is the -0.7 °C isotherm, which has been used to calculate this as the greatest phase change rate takes place at that temperature. **(c)** Equivalent recharge rate (net rate of loss in pore space). The points on the curve and the outer edges of the shaded area represent the recharge rates, which were assigned to the corresponding zones in the groundwater model.

### 5.1.2 Permafrost aggradation and recharge equivalent

Simulated rates of present permafrost aggradation ranged from -0.02 – 5 cm yr$^{-1}$ and generally decreased up-valley with older exposure ages and for scenarios with lower porosity (Fig. 7b). The correction for dynamic storage effects generally increased the "effective" permafrost aggradation rate that was used to calculate the recharge equivalent. The reason for this was that basal permafrost growth was generally faster in the past (Fig. 6). For a homogeneous medium with one-sided freezing, the freezing front progression rate will decrease with time (as exemplified by Figure S2). At first, it therefore seems surprising that the model simulations generally did not suggest that the highest aggradation rate occurred where permafrost is youngest (zone 0-1), but instead at zone 1-2. This was due to the heterogeneity in the model domain as expressed by the different properties of the sediments and bedrock undergoing freezing. In zone 0-1, closest to the shore, phase change took place at < 60 m b.g.l. corresponding to the most porous and least thermally diffusive unit (Qt1, insert on Fig. 5a and Table 2). Thus, a relatively high amount of latent heat had to be released for the freezing front to aggrade. By contrast, the opposite was the case in zone 1-2, where the freezing front just entered the sandstone unit that possessed the lowest porosity (Carolinefjellet Fm, insert on Fig. 5a and Table 2). For the same reason, Fig. 7c shows that the pattern of the equivalent recharge decreases from 1.1 – 1.9 mm yr$^{-1}$ closest to delta front (zone 0-1), to -0.02 – 0.16 mm yr$^{-1}$ in zone 10 furthest up-valley.

### 5.2 Simulated groundwater flow paths and flow velocity distributions

The above simulated recharge equivalent rates were assigned to cells within the corresponding zones of the model area (Fig. 5b). The resulting outputs of the groundwater model for all scenarios are shown in Fig. 8. The equivalent recharge rates, $REq$, calculated from the 1DHT model simulations resulted in a total inflow of water, $Q_{REq}$ to the groundwater system of 20, 40.5 and 63.4 m$^3$ day$^{-1}$ for the minimum, intermediate and maximum effective permafrost aggradation rate scenarios, respectively. For the minimum recharge equivalent scenarios (Fig. 8a), the slightly negative recharge equivalent rates (i.e., permafrost thaw) simulated at distances further than 6 km from the delta front result in a total hydraulic pressure loss that corresponds to a discharge rate of 2.4 m$^3$ day$^{-1}$.

Simulated hydraulic heads ranged from sea level to maxima between 119 m a.s.l. (maximum $Q_{REq}$ and minimum $K$ scenario, Fig. 8-1c) and 10 m a.s.l. (minimum $Q_{REq}$ and maximum $K$ scenario, Fig. 8-3a). The only two exceptions were the

minimum $Q_{REq}$ and intermediate to maximum $K$ scenarios (Figs. 8-2a and 8-3a) for which hydraulic heads went down to -6 and -10 m a.s.l., respectively, in the up-valley part of the system. The simulated hydraulic head in well DH4 was generally within the range deduced from well outflow (Braathen et al., 2012), but significantly above for the maximum $Q_{REq}$ and minimum $K$ scenario (Fig. 8-1c). For two other scenarios, the simulated head fell slightly outside the deduced range (above and below, respectively for the intermediate $Q_{REq}$ and minimum $K$ scenario and the minimum $Q_{REq}$ and maximum $K$ scenario,

Figs. 8-1b and 8-3a). For the remaining six scenarios, the groundwater model simulated heads within the uncertainty range of the observation. As illustrated by the colour fill on Fig. 8, entirely or almost entirely artesian conditions were simulated for all but four scenarios (all small $Q_{REq}$ scenarios and the intermediate $Q_{REq}$ and maximum $K$ scenario, Figs. 8-1a, 8-2a and 8-3a–b), where the up-valley part of the system has hydraulic pressures below ground level.

        With the porosities listed in Table 3, groundwater flow paths and pore water velocities were evaluated from all

simulations using particle tracking (Pollock, 2016). In order to visualise groundwater movement towards the outlet points, we used the particle tracking to draw 3 kyr catchment zones (the term 'catchment' is somewhat misleading in this context as no actual recharge takes place). The 3 kyr duration was chosen because it is on that order of time that the modelled permafrost and groundwater conditions likely existed. For most scenarios, water particle path lines (blue lines, Fig. 8) depicted a multidirectional flow pattern with local catchment zones for each outlet point. A more uniform down-valley-directed flow

pattern was simulated for the intermediate $Q_{REq}$ and maximum $K$ scenario (Figs. 8-3b). For the minimum $Q_{REq}$ scenarios (Sc$X$a, Figs. 8-1a, 8-2a and 8-3a), the negative equivalent recharge rates in the up-valley part of the system (Fig. 7c) resulted in a bidirectional flow pattern with groundwater flowing away from a groundwater divide located ~ 2 km from the delta front. The uniform and bidirectional flow patterns coincide with the partly non-artesian conditions and the lack of discharge at the up-valley pingo springs. As also illustrated by the size of the 3 kyr catchment zones, the simulated mean pore water velocities

ranged from 0.05 to 0.14 m yr$^{-1}$ (Figs. 8-1a and 8-3c, respectively) and suggest a relatively stagnant groundwater system. This is in accordance with mean residence times that ranged from 60 to 950 kyr (respectively, maximum $K$ and $Q_{REq}$ and minimum $K$ and $Q_{REq}$ scenarios, Figs. 8-3c and 8-3a) and by far exceed the duration of the Holocene.

        The colour fill and the bar charts on Fig. 8 together show that the hydraulic pressure below a pingo site needs to be artesian in order for outflow to take place at the pingo. Outflow from all pingo sites were simulated for four scenarios including

all three with maximum $Q_{REq}$ and the one with intermediate $Q_{REq}$ and minimum $K$ (Figs. 8-1b–c, 8-2c and 8-3c). The simulated spring discharge rates (bar charts, Fig. 8) increased with increasing $Q_{REq}$ and decreasing $K$, and had a maximum value of 0.31 L s$^{-1}$ at FHP for the scenario with maximum $Q_{REq}$ and minimum $K$ (Fig. 8-1c). The simulated proportion of the total outflow not discharging at the springs varied from 30 % to 89 % (the extremes respectively illustrated by Figs. 8-1c and 8-3a). For intermediate and maximum $Q_{REq}$ scenarios (Sc$X$b–c, Fig. 8), all of this discharged to the fjord, while, for the minimum $Q_{REq}$

scenarios (Sc$X$a, Fig. 8), a minor proportion of the outflow represents phase change associated with permafrost thaw, which causes a storage redistribution within the model domain. Larger proportions of outflow to the fjord were simulated in the case of maximum $K$ and a small total inflow $Q_{REq}$, with the former being the most important parameter.

**Figure 8** (next page) Groundwater model simulations from all 9 scenarios. The individual diagrams are sorted so that the hydraulic conductivity increase along the right-hand axis (scenarios **Sc1**–**3x**) and the equivalent recharge produced by permafrost aggradation increase along the left-hand axis (scenarios **Sc*X*a**–**c**). On each diagram, the following simulation results are illustrated: Heads from the uppermost grid layer are shown in m a.s.l. by isopotential contours (note that the colour scales are different) and in m a.g.l. by the colour fill (see scale at the bottom of the figure). The latter indicates if artesian conditions are simulated (reddish) or not (blueish). The groundwater outflow rates

and distribution are illustrated by bar charts with the location of the discharge points (pingos and fjord) indicated on **(1c)**. For minimum $Q_{REq}$ scenarios (**1a**, **2a** and **3a**) part of the outflow represents phase change associated with permafrost thaw (Figs. 7b–c), which causes a storage redistribution within the model domain. Flow patterns are illustrated by thin blue lines, which each depict pathways of particles released in the uppermost grid layer. The mean pore water velocities, shown in the upper left-hand corners, were calculated from the aforementioned particles and using the same porosities for all scenarios (Table 3). The thick dashed lines outline the areas (catchments) that

contributed water to the outlet points during 3 kyr of simulation time. In the lower right-hand corner, the simulated head in well DH4 is illustrated on a range plot with the range defined as deduced from Braathen et al. (2012). The colour of the bar indicates if the simulated head falls within (green) or outside the deduced range with less (yellow) or more (red) than 10 % of the total range. The location of DH4 and the pingo springs is marked on **(1c)**.

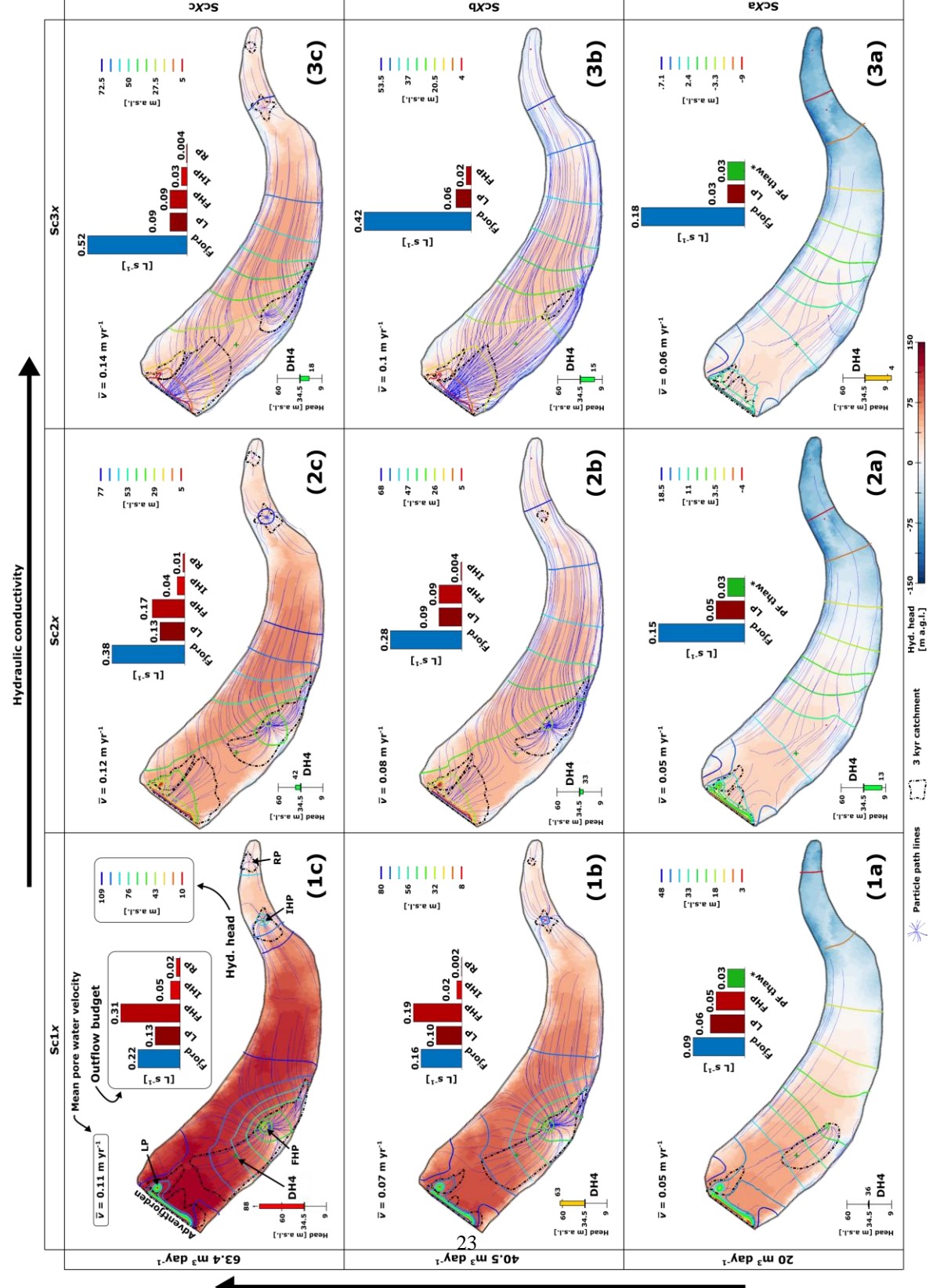

## 6 Discussion

### 6.1 Alternative processes producing sub-permafrost overpressures

In this work, we contemplate the implication of deep (sub-permafrost) groundwater systems being restricted from recharge by impervious continuous permafrost. It seems enigmatic that anomalous overpressures and springs still persist in such landscapes, where the lack of warm-based ice at the bottom of glaciers or ice-sheets, or other features capable of maintaining through-taliks as pathways for groundwater flow, seem to rule out recharge from precipitation. Directly supporting the main research hypothesis, the outcomes of our investigation suggest that basal permafrost aggradation due to equilibration of ground temperatures may produce sufficient artesian pressures to sustain such a system. However, other mechanisms producing anomalous pressures relative to hydrostatic conditions cannot be excluded and are therefore considered below.

Anomalous pressures may occur in groundwater systems that are either hydrodynamically equilibrated or disequilibrated (Neuzil, 1995). To clarify the meanings of those two systems states in this context, equilibrated systems are in a steady-state condition with the geological and hydrological setting, while disequilibrated systems are not. For the former, anomalous pressures are typically produced by topography-driven head gradients, but as the permafrost conditions in Adventdalen seem to rule out this process, we focus on the disequilibrium-type of anomalous pressures. These result from geological or glacial processes and can be further classified into systems where anomalous pressures equilibrate to past or ongoing perturbations. Anomalous pressure produced by permafrost aggradation is an example of the latter and we will discuss this in Sect. 6.3 using the model simulation results. First we consider the alternative processes.

#### 6.1.1 Equilibration to past perturbations – glacial loading

A possible interpretation of anomalous overpressure is that a previous perturbation was long-lived enough to redistribute groundwater (and other fluids) and recent enough for groundwater not to have adjusted to the present conditions (Bahr et al., 1994). The notion of adjustment time (i.e., $t_a$, see Sect. 4.3.4 and Eq. 8) becomes convenient when assessing whether an ice-load removed $> 10^4$ yr ago could be responsible for present anomalous pressures. For shallow, low-permeability and well-consolidated bedrock systems like the one investigated in this work, we found the vertical $t_a$ to be between 80 and 7500. To calculate this, we used the lowest $K$-estimate of the dominant hydrogeological unit (Janusfjellet Subgroup, Table 3) and a characteristic length of 200 (approximating half of the thickness of the aforementioned unit). The specific storage was defined like when calculating the horizontal adjustment time (Sect. 4.3.4). Conclusively, we argue that overpressures in systems like the investigated case cannot be explained by equilibration to past glacial loading.

#### 6.1.2 Equilibration to ongoing perturbations – density contrasts and gases

Past geological or climatic events may be indirectly responsible for ongoing pressure perturbations in systems that adjust to active hydrodynamic processes (Neuzil, 1995). Here, we consider the potential pressure contribution derived from groundwater density contrasts and the occurrence and possible production of gases and gas hydrates.

For a coastal setting like our field site, it is relevant to consider how a potential disequilibrium of the freshwater-saltwater interface could drive groundwater flow. During glaciation, the equilibrium depth of this interface must have been considerably deeper than today if subglacial melting could replenish the groundwater system. Following glacial unloading, the fresh groundwater body is in disequilibrium with the decreased surficial pressure and the density contrast directs hydraulic gradients upwards. Specifically, the freshwater head approximates to 0.025 times the depth of the seawater-freshwater interface, assuming hydrostatic conditions and densities of 1025 and 1000 kg m$^{-3}$ respectively (following the Ghyben-Herzberg relation, Verruijt, 1968). In other terms, to explain artesian pressures resulting in outflow from springs situated at elevations of up to 77 m a.s.l., the seawater-freshwater interface has to be situated at depths down to 3 km b.g.l. unless extensive brine is present. The actual seawater-freshwater interface is most certainly located above this depth, and the observed low-pressures at ~ 800 m b.g.l. (Braathen et al., 2012) definitively exclude density contrasts as a main cause for the overpressures in the shallower system. We conclude that density contrasts may theoretically contribute to the artesian pressures, but only insignificantly.

The presence of gases, either free in solution or bound in gas hydrate complexes (i.e., clathrates), may affect hydraulic pressures (Neuzil, 2003). Pingo spring waters and sub-permafrost groundwater in Adventdalen indeed contain both $CH_4$ and $CO_2$ (Hodson et al., 2020). The methane is dominated by a biogenic fingerprint (Hodson et al., 2019) and contemporary methanogenesis is probable (Huq et al., 2017). In order to explain pressure build-up, the essential question is whether any ongoing process produces or releases gas to the groundwater, thereby increasing pressure. Such processes may be exemplified by methanogenesis or clathrate dissolution, which cause the partial gas pressure to exceed the hydrostatic pressure, so that free gas forms and replaces groundwater in the pore space. We speculate that, over time, groundwater flow driven by this process is limited, as it represents neither a groundwater source nor a net loss in pore space. In Adventdalen, the pressure and temperature conditions at the base of permafrost are at the threshold for gas hydrate stability, and controlled mainly by the gas composition (Betlem et al., 2019). Whether partly responsible for groundwater flow or not, this proximity to the boundary of hydrate stability means gas clathrates may currently represent a pressure buffer. This is because should clathrates be present, any decrease in pressure to conditions below thermodynamic stability initiates clathrate dissolution, thereby releasing gases and increasing the pressure. It is as yet unclear which form the methane predominantly takes below the permafrost in Adventdalen. However, the near-stability conditions, the documented sub-permafrost gas accumulation, and the recent history of climate warming do make the buffering effects of clathrate dissolution more likely.

## 6.2    Model limitations, extent and uncertainties

For sub-permafrost groundwater systems, an extraordinary amount of relevant data and research exists for Adventdalen (van der Ploeg et al., 2012) and this arguably makes Adventdalen an optimal case for investigation. There were, nevertheless, too few observations for calibrating the numerical models in a statistical way, and so in the following, we consider the model limitations carefully before drawing conclusions from the simulation results.

### 6.2.1 Limitations related to model approach

Using an approach where transient one-dimensional heat flow modelling was decoupled from steady-state three-dimensional groundwater modelling required an array of assumptions that deserve attention. Modelling heat transfer one-dimensionally in the vertical dimension implies that no lateral conduction was considered. The inherent assumption was thus that the isotherms are horizontally parallel. This assumption holds inland, but for a coastal setting the slope of an isotherm is expected to increase seawards and in some cases cause a thermal "bulge" under the sea floor (c.f. Gregersen and Eidsmoen,
1988; Taniguchi, 2000). The 1DHT model simulations indicated that even the largest isotherm slope gradients in Adventdalen are quite small (Fig. 7a). Between the two most seawards points, the gradients for the -2 °C and 0 °C isotherms are -25 and -35 m km$^{-1}$ ($\Delta z\ \Delta x^{-1}$), respectively, in landwards direction. Effectively, if heat transfer in this work had been modelled in 2D, the simulated permafrost aggradation rate would have been slightly slower close to the coast due to lateral heat transfer. Nevertheless, considering how small the isotherm slope gradients are, the lateral heat flow component was considered
negligible and we found the one-dimensional modelling approach appropriate.

        The inability to model advective heat transfer represents an uncertainty proportional to the importance of this process over the time scale in question. From the hydrogeological properties and the spring water chemistry discussed later on, we realised that the groundwater system must be relatively stagnant. We therefore assumed that the only uncertainty related to the omission of advective heat transfer arose from neglecting the energy leaving the system with the groundwater. By definition,
the greatest discrepancy between simulations and actual conditions must occur locally at the outflow points to which the advective heat transfer rate is greatest. On the regional scale, however, we infer that advective heat transfer played an insignificant role. Disregarding shallow formation of visible ground ice, the total pore water expulsion by freezing approximates to 9 % of the volume of ice in the pore space. The average pore water velocity of this water when it got expelled from the system can be approximated by half of the frozen ground depth (assuming this represents the mean travelled distanced)
divided by the porosity and the time it took for frozen ground to establish. For the investigated system this yields a Peclet number of $P_{e_L} < 0.02$, implying that advective heat transfer played an insignificant role on the regional scale (taking 100 m as an average frozen ground thickness and characteristic length, and the thermal diffusivity of frozen ground).

        The steady-state approach of groundwater modelling implied that dynamic storage effects could not be simulated ($S_s$, Eq. 5). To account for the present pressure contributions from previous permafrost dynamics, we instead applied the time-
moving average to the development of the permafrost base before calculating the recharge equivalent (Fig. 4). Another drawback from the steady-state approach, which was not accounted for in the model setup, is the possible overestimation of the pressure contribution that arrives from not considering the effect of the Holocene sea level fall. Neglecting this effect represents an uncertainty in the simulation results, but because sea levels already reached levels close to the present day by 5 ka, we regard it as insignificant.

## 6.2.2    Model extent uncertainty

The boundary conditions of the groundwater model define a bathtub-like system with the pingo springs and the fjord as the only discharge points, and with the expansion of water upon freezing within the model regime as the only source of hydraulic pressure. In reality, the hydrological system in Adventdalen may not entirely conform to this description as groundwater flow across the model boundaries cannot be rejected. Additional recharge could, for example, occur through microcracks below the valley floor induced during glaciation (Leith et al., 2014). Likewise, hydraulic pressures may, to a greater extent than simulated, dissipate directly to the fjord through unknown pathways. In this respect, our model serves to isolate the pressure effect of freezing expansion in Adventdalen and systems like it. These conditions should be taken into account before drawing site-specific conclusions from the modelling results.

Due to Early Holocene warming (Fig. 3), the 1DHT simulation results showed that continuous frozen ground in Adventdalen is likely younger than 6.5 ka even where the valley floor is older (Fig. 6). This is supported by geomorphological and geochronological evidence (Humlum, 2005). As such, there seems to be no reason why permafrost dynamics in the valley bottom outside the HML should be markedly different from that in the up-valley part of the model area. Based on the above, it is possible that basal permafrost aggradation goes on beyond the model area (HML) and model simulations may have underestimated the freezing-induced pressures affecting spring discharge.

The dominantly low-permeability groundwater system challenged a physically determined lower boundary for the model domain. From the significant low-pressures observed in deeper stratigraphic layers (~ 800 m b.g.l. at DH4, Braathen et al., 2012), we inferred isolation of the investigated groundwater system from that below and simply assigned the base to a depth of 300 m b.g.l. By simulating scenarios with a lower base of 250 and 400 m b.g.l., we found that simulation results did not change significantly (< 1 % deviation of simulated heads and discharge rates).

## 6.3    Do model simulations represent the processes in the groundwater system in Adventdalen?

The amount of hydrogeological data from Adventdalen was insufficient for automatic calibration of model parameters and the model simulations should therefore at best be taken as possible scenarios for the conditions in Adventdalen. However, some scenarios yielded simulations that must be considered at odds with the available observations. For the minimum $K$ and maximum $Q_{REq}$ scenario (Figs. 8-1c), the simulated hydraulic heads were almost certainly too high. The only head observation supports this view and we therefore suggest that the hydraulic conductivities in reality must be higher if the recharge rate (real and/or equivalent) is as employed here (or higher). Further, discharge from the up-valley pingo springs was not simulated for scenarios with minimum $Q_{REq}$ and maximum $K$ (Sc$X$a and Sc3b, Figs. 8-1a, 8-2a, 8-3a–b). This could indicate that the real $Q_{REq}$-values are in effect higher than those employed for these scenarios. However, since alternative processes contributing to hydraulic pressures were not incorporated in the model, such conclusion is speculative. If permafrost aggradation is the main driver of pingo spring outflow in Adventdalen, the most plausible representation of the system is likely to be found within the scenarios shown in Figs. 8-1b, 8-2b–c and 8-3c).

For the flow pattern, a more certain conclusion could be drawn. The flow was simulated to be multidirectional, with local sub-catchments appearing whenever the groundwater model was able to simulate artesian spring discharges from the pingos. This indicates that regional groundwater flow across the model area is very limited.

On Fig. 9, simulated spring outflows from all model simulations are plotted together with the few available observations. Validation of any particular model scenario was not possible due to the variability and paucity of observations. Nevertheless, assuming that only a small fraction of the discharge (if any) freezes within the pingo and becomes part of its core, the comparable observed and simulated discharges suggest that permafrost aggradation alone may explain the presence of the pingos and their springs in low-permeability systems comparable to the one modelled. To investigate the effect of

additional pressure sources, we ran the groundwater model with maximum $K$-values (Table 3) and different, uniformly distributed recharge rates, all resulting in a greater total inflow of groundwater than from permafrost aggradation alone (Fig. 8). We used the maximum $K$-values to allow for the highest amount of recharge to enter the system. Figure 10 illustrates such a scenario with a recharge rate of 1 mm yr$^{-1}$. We found that, if the pressure production exceeds that equivalent to a recharge rate of 1 mm yr$^{-1}$, hydraulic heads rise unrealistically (i.e., > 200 m a.g.l.). Within the model limitations (Sect. 6.2), Fig. 10

may thus be regarded as an approximation of the upper limit of possible total inflow (or pore space loss) rates (161.4 m$^3$ day$^{-1}$). The corresponding spring water discharge rates range from 0.12 to 0.56 L s$^{-1}$ (Fig. 9) and the mean residence time is 24 kyr.

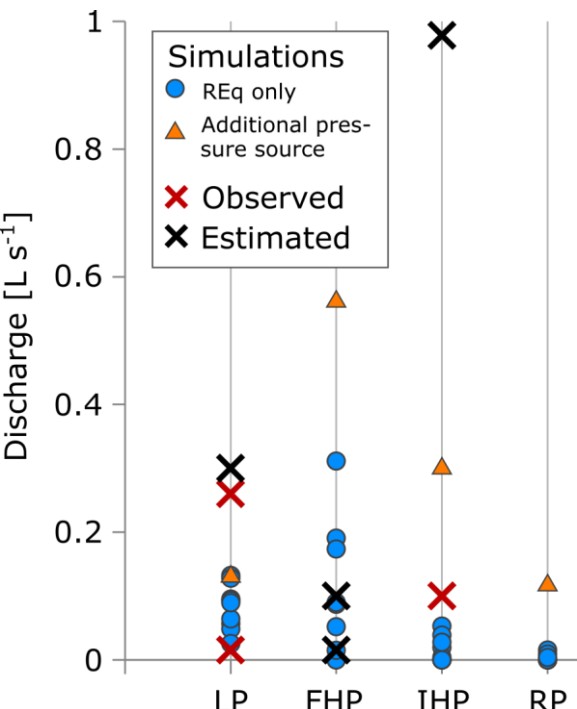

**Figure 9** Simulated, observed (measured) and estimated ("by the eye") spring discharge rates. Blue dots are discharge rates simulated for groundwater model scenarios where the source term is defined by the rate of basal permafrost aggradation (Figs. 7 and 8). The orange dots are discharge rates simulated for a groundwater model scenario with an additional unknown pressure source (Fig. 10).

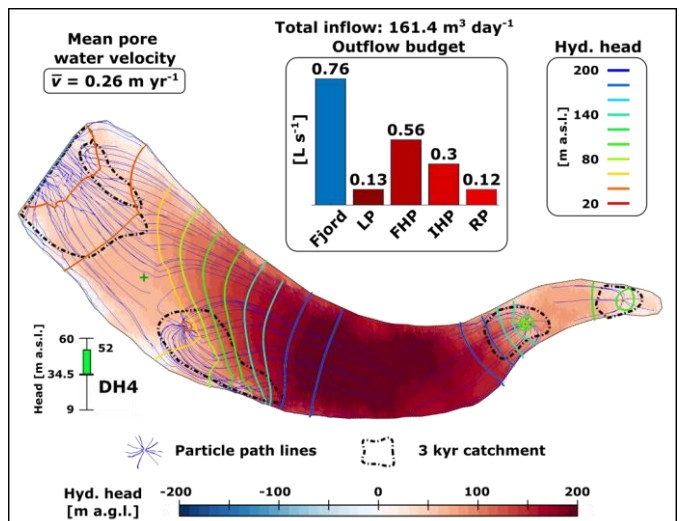

**Figure 10** Groundwater model simulation representing additional pressure sources. Recharge is distributed uniformly with a rate of 1 mm yr[-1]. This figure uses the same structure and legend as in Figure 8.

## 6.4    Comparison with hydrological processes inferred from pingo spring water chemistry

The lack of hydrological data for model calibration make comparison with other information on the groundwater system ever so important. In this context, hydrochemical data from 25 pingo spring water samples from 2014 to 2017 (Hodson et al., 2020: see DOI:10.5285/3d82fd3f-884b-47b6-b11c-6c96d66b950d) give additional insights into the groundwater system. Accordingly, water samples from LP, FHP, IHP and RP reveal that all these pingo springs share the same sodium-bicarbonate ($NaHCO_3$) water type (illustrated in the Supplement), which is commonly associated with freshening of a saline groundwater system (e.g., Giménez-Forcada, 2010). The only exceptions are four samples taken near River Pingo in 2017 of a magnesium-sulfate water type. These four samples were excluded from the following discussion because they might not be associated with a pingo according to Hodson et al. (2020).

Among the 21 $NaHCO_3$-dominated samples a few distinct trends were observed in the hydrochemistry. Specifically, the concentration of $Cl^-$ and heavy stable water isotopes both increase in the up-valley direction. To illustrate this, the concentration of $Cl^-$ is plotted against $\delta^{18}O_{H_2O}$ in Fig. 11. The former has a relatively constant concentration at each site compared to the latter. We inferred that the variation of the $Cl^-$ concentrations between the different springs reflects an up-valley variation in the sub-permafrost groundwater system, and not processes acting locally along the flow paths towards the individual pingos. If the latter had been the case, we would expect to see greater intra-site variation.

In order to explain the increasing $Cl^-$ concentration and $\delta^{18}O_{H_2O}$ we considered solute rejection and isotope fractionation associated with freezing as well as mixing between seawater and freshwater. For the former, we found that the positive relationship between the two parameters was incongruent with this being a major control in the investigated system: when ice forms, water molecules containing the heavier isotope $^{18}O$ (and D) are preferentially included in the ice, while the residual water becomes isotopically lighter. For freezing in a well-mixed and closed reservoir, the isotopic ratio in the residual

water, $\delta_w$, may be expressed by a Rayleigh-type fractionation according to the following equation (Lacelle, 2011):

$$\delta_w = \delta_0 + \ln(\alpha_{i-w}) \cdot 1000 \cdot \ln(f) \qquad\qquad (8)$$

where $\delta_0$ is the initial isotopic ratio in the water, $\alpha_{i-w}$ is the fractionation coefficient between ice and water, and $f$ is the fraction of residual water. $\alpha_{i-w} = 1.003$ (Lehmann and Siegenthaler, 1991) at equilibrium and approaches 1 for faster freezing rates (non-equilibrium). The decrease of $\delta^{18}O_{H_2O}$ in the residual water resulting from freezing is exemplified in Fig. 11 by

taking a typical LP water sample (i.e., with the lowest salinity) and assuming that $Cl^-$ is completely excluded from the ice. The shading indicates the possible range of $\delta^{18}O_{H_2O}$ that can be derived from this process. Although two samples from FHP fall within the shaded area, it was hard to explain the composition of these with freezing from a LP-type water, as the remaining samples did not comply with this interpretation (Fig. 11).

        Another model line on Fig. 11 is drawn by assuming that the LP samples result from mixing of seawater and a

freshwater end-member with zero $Cl^-$ concentration. Generally, the remaining samples agree with this model line and hence we argue that mixing is the more feasible explanation. This was also in line with the interpretation that the $NaHCO_3$-dominated hydrochemistry reflects freshening.

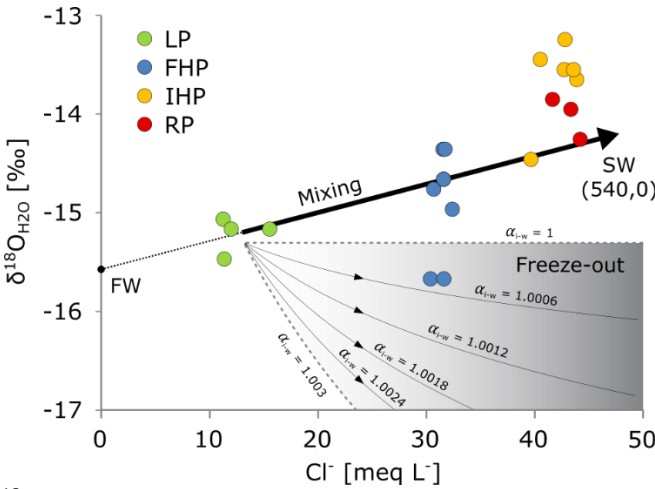

**Figure 11** $Cl^-$ concentration and $\delta^{18}O_{H_2O}$ of pingo spring water samples. The thick black model line is drawn by assuming that the LP samples result from mixing of seawater and a freshwater endmember with a zero $Cl^-$ concentration. The range of possible $\delta^{18}O_{H_2O}$-values following freeze-out (Eq. 8) from water with an initial composition alike LP is illustrated with the shaded area. Here, $Cl^-$ is assumed to be

completely excluded from the ice. The lower and upper edges of the shading represent equilibrium fractionation and no fractionation, respectively, while the intermediate model lines illustrate fractionation at 20 %, 40 %, 60 % and 80 % (top–down) of equilibrium conditions.


For the above mixing scenario, field surveys suggested a somewhat unusual trend with greater fractions of freshwater towards the sea, where $\delta^{18}O_{H_2O}$ and Cl⁻ concentrations approach those of the inferred freshwater end-member in Fig. 11. In the case of present-day recharge from the adjacent mountains (as demonstrated at a pingo 35 km South West of Adventdalen by Demidov et al., 2019), we would not expect to observe such a systematic trend along the valley axis, as this would be unlikely

in a system dependent upon localised areas for infiltration. We therefore suggest that the unexpected landwards increase in Cl⁻ is difficult to explain without the sequence of events illustrated in Fig. 12. During glaciation, the groundwater system was recharged by subglacial melting (Fig. 12a). Despite being covered by the sea during the Early Holocene, seawater could not infiltrate substantially into the groundwater system to replace the fresh groundwater (Fig. 12b). On Fig. 12c, the body of freshwater forms a wedge that thins inland below the freezing front due to the density difference with seawater. Moving away

from the sea, the springs expel more saline groundwater because the wedge thins in this direction and the permafrost thickness increases (Fig. 12c). Therefore both the sources of water and the hydraulic conditions of the groundwater system seem intricately linked to landscape evolution throughout the Holocene.

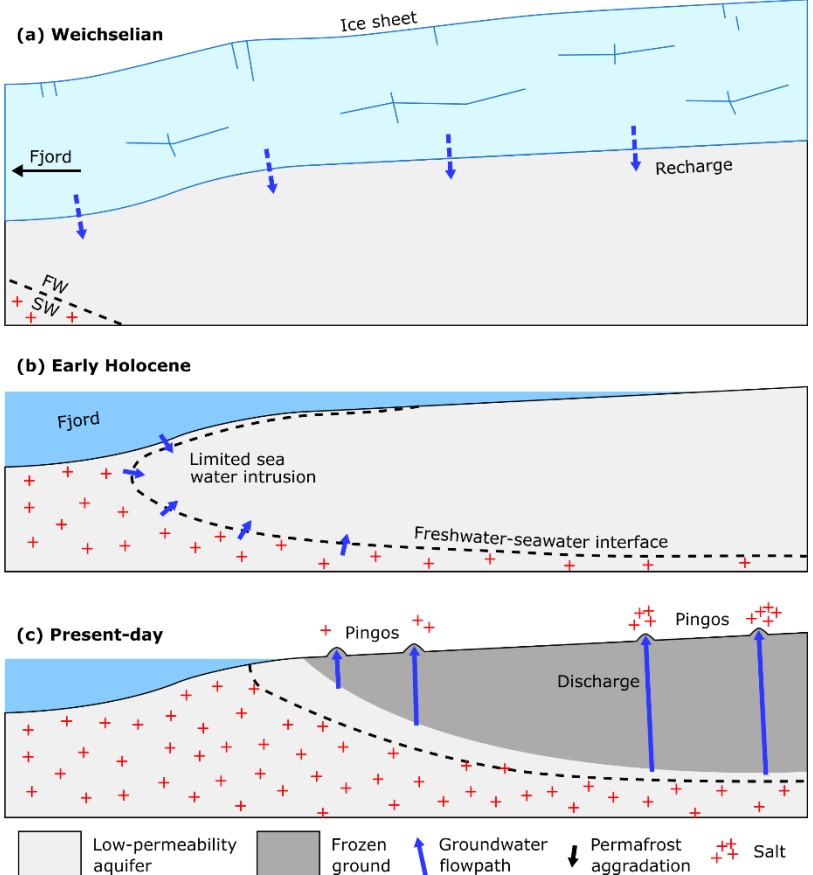

**Figure 12** Possible interpretation of hydrochemical trends observed in pingo spring waters. **(a)** Subglacial melting from the Weichselian ice sheet recharged the groundwater system with freshwater. **(b)** Although covered by the sea during Early Holocene, low-permeability rocks limited infiltration of sea water and the fresh groundwater body persisted. **(c)** Due to the shallower permafrost depth towards the sea, spring water sampled in this direction holds a lower concentration of sea water (salt).

## 6.5    Pan-Arctic significance

The hydrological system proposed here, where time-transgressive basal permafrost aggradation in an uplifted Arctic valley is able to drive flow of deep groundwater to the surface, may at first appear unusual. However, the combination of Holocene marine regression and climate cooling, which have resulted in this system, is neither unique to Adventalen nor to Svalbard. As exemplified by Holocene temperatures on Svalbard (Figure 3), a general climate cooling from the Holocene thermal maximum to pre-industrial age occurred throughout the Arctic (Kaufman et al., 2020). Marine regression took place where the rate of postglacial rebound or sediment supply surpassed the Holocene eustatic sea level rise (Kidson and Heyworth, 1978). Virtually all valleys on Svalbard and Greenland located close to sea level have been exposed to such postglacial rebound (Ingólfsson and Landvik, 2013; Weidick and Bennike, 2007), and other examples are found in the Canadian high Arctic (Aitken and Gilbert, 1989; Bell, 1996; Lemmen et al., 1994). Considering the frequency of these fundamental conditions required for the

proposed mechanism, we argue that millennial-scale permafrost aggradation may also drive flow of sub-permafrost
groundwater to the surface in other uplifted Arctic valleys.

**7        Conclusions**

Results from the decoupled heat and groundwater model show that millennial-scale basal permafrost aggradation may alone produce hydraulic pressures sufficient for the formation of pingos and their spring water outflows when the right conditions are met. In addition to the climate cooling necessary for permafrost aggradation, a relatively low-permeability groundwater
system with limited dissipation of hydraulic pressures are also required for pingo formation. Pingos formed in this way do not conform to the traditional differentiation between open-system and closed-system types, but constitute a borderline case: by definition, they classify as open-system pingos, because the groundwater body from which spring water is expelled is not enclosed in permafrost. Generically though, they relate more to closed-system pingos, because the causal mechanism of hydraulic pressures is essentially similar, although operating over much longer time-scales. We emphasise that this conceptual
model for pingo formation represents an end-member of open-system pingo-forming processes, which is not exclusive, but may act in combination with others, such as those reported nearby by Demidov et al. (2019).

The simulation results from the 1DHT model suggested that basal permafrost aggradation in Adventdalen presently induces head gradients corresponding to the effect of a recharge rate of ~ 0.1 mm yr$^{-1}$ furthest up-valley, and increasing to ~ 1 mm yr$^{-1}$ towards the sea. By applying these rates to the groundwater model, we simulated spring outflow rates of the order of
$10^{-1}$ L s$^{-1}$. Due to the probable occurrence of basal permafrost aggradation outside the model area, these may be underestimations. Nevertheless, the simulated and observed spring outflow rates at Adventdalen pingos were of the same order of magnitude, suggesting that they likely form at least partly in accordance with our conceptual model. This further suggests that overpressures induced by water expansion during freezing in other sub-permafrost groundwater systems can result from permafrost aggradation.
The simulated aquifer flow paths and flow velocity distributions suggested that sub-permafrost groundwater flow in Adventdalen is characterised by slow mean pore water velocities (<0.25 m yr$^{-1}$) and long residence times (> 2.5·$10^{4}$ yr) that exceed the duration of the Holocene. The groundwater system most likely has a multidirectional flow pattern with individual catchments around each pingo.

Alternative and non-recharge-related processes that may also affect sub-permafrost pressures were considered. The
role of gases may be particularly important in this context, because it is likely that methane hydrates have influenced the groundwater system under investigation. However, methane hydrate dissolution may in fact act as a pressure buffer and prolong artesian pressures after permafrost ceases aggrading (or starts thawing). This represents an uncertainty in forecasting how groundwater and methane fluxes will react to climate change. Unresolved questions regarding the occurrence and formation of gases in sub-permafrost groundwater systems therefore constitute an ongoing challenge for Arctic science.

The presence of a positive relationship between Cl$^-$ and $\delta^{18}O_{H_2O}$ in the pingo spring water samples suggests that mixing between seawater and freshwater is the major control of hydrogeochemistry in the sub-permafrost groundwater system prior to aggradation. As a result, a somewhat unexpected but clear trend of increasing salinity in an up-valley direction was found. Therefore, given the relatively stagnant groundwater system, Weichselian subglacial meltwater could endure the Early Holocene inundation and result in the present-day situation where a freshwater body forms a wedge that thins in the inland

direction below the permafrost. This possible interpretation readily explains the inland increase of the spring salinity because the inland springs expel groundwater from greater and hence more saline depths.

        To our knowledge, this is the first study to show that time-transgressive basal permafrost aggradation in an uplifted Arctic valley represents a feasible mechanism for driving the flow of deep groundwater to the surface, and so the hydrological system in Adventdalen might appear unusual at first. However, considering that isostatic uplift in permafrost valley systems

is common across the Arctic, millennial-scale permafrost aggradation deserves more attention as a driver of sustained gas and groundwater flows from the sub-permafrost environment to the land surface.

**Code availability**

The 1DHT model code is publicly available at DOI:10.5281/zenodo.4240594.

**Supplement link**


**Author contributions**

MTH developed the 1DHT model code and designed the modelling experiments with contributions from VB. AJH, SJ and MTH analysed the hydrochemical data. MTH prepared the manuscript with contributions from all the co-authors.

**Competing financial interest**

The authors declare no competing financial interests.

**Acknowledgements**

This paper largely emanated from work initiated with MTH's MSc thesis (Hornum, 2018) and the authors acknowledge Prof. Peter Engesgaard for supervising the thesis work. The authors also acknowledge The Research Council of Norway grant

294764. The authors are grateful to the editor (Peter Morse) and two reviewers (Melissa Bunn and an anonymous reviewer) whose constructive reviews were very helpful for improving this manuscript.

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
