# Peer review of "Numerical modelling of permafrost spring discharge and open-system pingo formation induced by basal permafrost aggradation"

_The Cryosphere, 2020_

## Referee Comment (RC1) · Melissa Bunn (Referee) · 11 Mar 2020

Summary:

In this paper, the authors investigate permafrost aggradation and the associated increase in subpermafrost groundwater pressures over millennial scales as the potential cause of pingo springs in a high Arctic valley (Aventdalen, Svalbard). Continuous permafrost, high desert conditions, and a lack of wet-based glaciers in the adjacent highlands preclude recent groundwater recharge as a source of the spring water. Using a 1D heat flow model the authors quantify potential rates of permafrost aggradation. This aggradation is then related to a water flux which is applied as recharge in a 3D groundwater flow model. These processes are fully decoupled. The groundwater flow model represents the steady state flow of groundwater to the pingos (and the adjacent Fjord) that results from the additional subpermafrost water flux. Although validating field data is limited to sporadic spring flow measurements and hydraulic head measurements at a single borehole, what is available is compared to this data to support the development of the model and the proposed conceptualization of the pingo spring flow.

The mechanism proposed by the authors is new, and their use of numerical models to illustrate and quantify this mechanism is of value. The discharge of subpermafrost groundwater to the surface has the potential to introduce methane to the environment and other solutes to freshwater systems. Understanding of the mechanism that generates the driving hydraulic heads in a variety of geological settings improves our ability to forecast future conditions under a changing climate. Overall the conceptual model and the numerical approach is well presented. However, the paper gives the sense that the modelling work proves the conceptual model to be correct. In general, assumptions are made in numerical modelling to align the numerics with the conceptualization. The modelling presented in this paper is no different, and as such, the model output does not prove the conceptual model to be correct. Some factors that warrant further investigation to support the numerical modelling assumptions are detailed below. Value could be added to the paper by using the model to further explore the physical factors required to form pingo springs under this conceptual model.

The boundaries of the numerical groundwater flow model create a closed "bathtub" like system. Although the amount of recharge added is quite low (<1 mm/year), the presence of permafrost throughout the top of the model domain, a lower boundary that is within 100 m of the base of the permafrost, and no flow across lateral boundaries, leaves only the Fjord and the pingo springs as discharge points. Additional details, or discussion would be useful in supporting the boundary selection for the groundwater flow model as follows:

- The paper investigates the effect of the lower boundary position by lowering the model depth by 100 m. A more illustrative demonstration of the effect of the lower boundary may be to lower the boundary to the detachment zone that separates the upper overpressured and lower underpressured groundwater flow systems (i.e., to a depth where there is field evidence of a no-flow boundary);

- There is no evidence provided for the presence of hydraulic divides at the flanks of the valley that would support the use of lateral no flow boundaries in the groundwater flow model as specified. This boundary prevents any regional flow in to or out of the valley. Were groundwater flow to follow the modest slope of the formations, flux through the Festningen Member may be sufficient to dissipate the recharge flux specified in the model. If field data is not available, the sensitivity of model results to deeper regional flow across the valley should be explored;

- As stated by the authors, the drain boundaries used to represent the pingo springs are placed within the upper most active cells closest to the spring, but within the Festningen Sandstone. Additional details and discussion of this placement would be of use. Figure 2(a) and Figure 4 indicate that the sandstone is not present at the Innerhytte Pingo. Figure 7 shows that the drain associated with the Førstehytte Pingo is opposite the valley axis from the surface expression. It is understood that the fractured nature of this sandstone could permit the required subhorizontal flow to the pingo; however the paper would benefit from additional discussion of this conceptualization (what is the inferred orientation of the fracturing that allows formation of the pingos). The sensitivity of the model results to the geological unit that the drain boundary is placed in should be discussed;

- The boundary representing the Fjord is described as being applied to the relevant cells. Please describe this assignment in more detail (are the boundaries assigned to the upper most active layer only, or are they assigned to several layers to the approximate seafloor depth in the Fjord). As the boundary at the Fjord

represents the highest flux from the model (approximately 40% to 90% of the total flux) model results can be expected to be very sensitive to the vertical location of this boundary, and should be investigated further; and,

- The potential for subpermafrost discharge to the Adventdalen River has not been considered in the conceptual model development. Rossi et al. (2017) suggest that this discharge may occur near the Innerhytte Pingo. Although the rate of subpermafrost discharge to this River may be low, the potential for it to occur should be considered in the overall balance of flows.

In general, further investigation of the effect of these various boundary conditions on the groundwater model results are required before it could be concluded that the model results show that the basal permafrost aggradation produces the hydraulic pressures to sustain the pingo spring water outflows as the authors have stated at the start of Section 7.

The observed hydraulic head at DH4 is stated to range from 9 m to 60 m above hydrostatic. It is unclear if this range is due to temporal variability, or the range in the correction for the effect of dissolved gasses. With limited field data for model validation, this warrants further discussion. As shown on Figure 7 (2a, 3b, 3a) simulations in which the hydraulic head at DH4 is on the lower end of this range (and with the lower to middle recharge flux) do not produce sufficient flow at the pingo springs.

The equivalent recharge applied to the model ranges from 25.4 m$^3$ /day to 56.7 m$^3$ /day. This range is related to the porosity of the formation through which permafrost aggradation is occurring. Based on the 1D columns shown on Figure 4, and the model results shown on Figure 5, much of this aggradation would occur within the shale units. The porosity of the Janusfjellet subgroup has been derived from Manger (1963). How was the range from 0.1 to 0.3 selected from the values provided in Manger (1963). The higher porosity units in this reference are related to high clay content shales or claystones. Is this valid for the Janusfjellet formation? Given that the higher porosity

ranges were required to produce a water flux that could sustain the pingo flows, further details should be provided on the derivation of these values.

How is the lower boundary of the 1D heat transfer model specified? If the geothermal gradient from surface is maintained, does that imply that temperature of the lower boundary changes with time? How would the rate of permafrost aggradation be changed if the depth of the 1D model was extended such that the heat flux at the bottom boundary could be kept constant through time? How would a cessation of permafrost aggradation up valley effect results?

In Table 3, the rock unit hydraulic conductivities derived from literature (the Festingen sandstone, the Janusfjellet subgroup, and the detachment zone) range within one order of magnitude across the three scenarios. While this could be considered a large range in this type of study, comparison to the observed hydraulic head at DH4 indicates that Scenario 1 (low hydraulic conductivity) values are unlikely, leaving a more reasonable half order of magnitude range. For the rock units with field data (the Carolinefjellet and Hevetiafjellet formations), the hydraulic conductivities applied range over two orders of magnitude. Does this range represent the maximum and minimum of tested values? It would be of value to plot the probability density function for the hydraulic conductivity values for each formation, selecting the geometric mean as scenario 2, and a more realistic percentile as scenarios 1 and 3 to tighten the potential range for these formations. As stated on line 406, the hydraulic conductivity is the most important parameter in determining the distribution of outflows between the pingos and the Fjord. Assignment of this parameter should be constrained where possible.

On line 515 it is stated that the steady-state assumption for the groundwater flow model results in an underestimate of the present-day pressures. This statement may oversimplify the transient groundwater dynamics that could occur as permafrost aggrades and the sea level retreats. Permafrost aggradation is highest in proximity to the Fjord, which is also where the greatest potential for discharge to the Fjord occurs. It is possible that any excess pressure would be dissipated as the sea level retreats, and that a transient

simulation may not show higher pressures.

On line 635 it was stated that simulated flows to the pingo springs are likely underestimated as basal permafrost aggradation outside of the model domain is not included. Were this aggradation to contribute to the pingo spring flows, it would imply that the lateral boundaries of the model domain are not hydraulic divides (i.e., no flow boundaries). This statement should be reconciled with the boundary selection.

Technical Corrections:

Line 13: ...wet-based glaciers **are not present** in the adjacent highlands

Line 18: ..and groundwater (**3D** -Steady-state)

Line 229 Equation 1: The $\delta z$ in the denominator should be $\delta z^2$

Line 235: ...heat conduction will flow **heat will be conducted** through a matrix of **solids (i.e sediment or rock)** and **liquid water**, ice or a mixture

Line 288: ..The fraction of **liquid** water

Line 247: ...When temperature change occur**s**

References

Rossi, G., Accaino, F., Boaga, J., Petronio, L., Romeo, R., Wheeler, W., 2018. Seismic survey on an open pingo system in Adventdalen Valley, Spitsbergen, Svalbard. Near Surface Geophysics 16, 89–103. https://doi.org/10.3997/1873-0604.2017037

---

## Author Comment (AC1) · 22 May 2020

We thank the reviewer for at very detailed, constructive and thorough review. The manuscript has been revised according to comments and suggestions made the referee.

The review of our manuscript clearly shows that the referee has studied our work carefully and evaluated it very professionally. We are pleased to see acknowledgement of the novelty of the pingo-forming mechanism we propose and that our research is well presented.

[Figure]

The reviewer is of the impression that the paper "gives the sense that the modelling work proves the conceptual model to be correct" and this leads to several specific comments on how limitations to the modelling mean it falls short of being able to provide such a confirmation. However, we did not intend to make such a claim, and merely wished to reveal the plausibility of our conceptual model through the modelling work. We therefore hope that our revisions make this message clearer. We also note that addressing the specific points raised by the reviewer have been of great value for improving our paper.

Some of the reviewer's critique concern the boundary conditions of the numerical models and she asks for data supporting our boundary selection. Despite the relatively extraordinary amount of relevant data available for Adventdalen, the boundary selection still relied on little information. As such, the boundary conditions are somewhat speculative and at best representative of one possible state of the investigated groundwater system. More data and greater certainty would be preferable, but we are left with this approach in lack of better options. This is a common challenge for modelling deep groundwater flow in permafrost areas (van der Ploeg et al., 2012).

Please see the supplement to this comment for a point-to-point answer to the referee's comments.

Please also note the supplement to this comment:
https://www.the-cryosphere-discuss.net/tc-2020-7/tc-2020-7-AC1-supplement.pdf
* * *

---

## Referee Comment (RC2) · Anonymous Referee #2 · 8 Jun 2020

In this paper, the authors investigate the question: "Why are there pingos in Adventdalen when there seem to be no groundwater recharge? They do this with a conceptual model, and find that the observed pingos are sustained by groundwater supply with residence times exceeding the duration of Holocene. In addition to the modeling, the work relies on a wide range of field observations.

The work presented in this paper is impressive. A new model is developed and data from many different sources are used in the study. However, the manuscript would benefit from revisions to improve readability and clarity. Currently, the manuscript contains many terms and context that need to be explained to expand readership beyond

immediate experts and to the diverse backgrounds of the readers of The Cryosphere. The minor comments below provide many examples.

Major comments ===========

1. Many terms and concepts are provided without much context. I give several examples in the minor comments.

2. Organize the manuscript into sections according to the traditional structure of a scientific paper (Abstract, intro, study site, methods and data, results, discussion, conclusions). In the current paper, there is no clearly defined method section and sections 2 to 4 gradually move from introduction to results. a. For example, the section 3.2 title suggests this section contains background information about the study site. However, besides that background information it also explains how climate history was reconstructed. Separate these two so that the background info on lines 150-154 goes in a study site description section, and the rest in a methods subsection about reconstructed temperatures b. Another benefit of a dedicated method section is that you can provide an overview of the methods at the beginning providing the reader with a sort of road map.

3. None of the tables has captions. It should be possible for a reader to understand tables and figures based on the caption.

4. Adding a description of the model simulations (number of simulations, parameterizations etc). in the methods would be helpful for the reader to anticipate the results.

5. The simulations with the groundwater model need to be better explained. Clarify what scenarios were run and why. What is the significance of the 3 kyr catchment. How were they drawn in the first place? Figure 7 is very complicated and should be simplified and ideally split up into several figures. In the text, it is stated that each of the 12 zones were run with a different REq. However, the y-axis suggests that only three values were used. Explain why head is visualized with two different types of units?

What is the purpose of the pie charts and all the information provided in the figure. It is difficult to cross-reference with the explanations in the text. Also provide context for the scenarios (max, min, mean). It is difficult to see the evidence in support of the conclusion that pingo's co-occur with regions of overpressure.

Minor comments ===========

L13: "methane emissions/release"

L40: Clarify what "they" refers to, e.g. springs

L47-L50. Clarify where the liquid water is coming from if not from the surface (e.g. relic groundwater?)

L70: Here and other places. Be careful with using words like "this" without being more specific about what "this" refers too (here pressure). It is easy for the reader to get lost.

L72: Define "talik"

L73: "suggest that and open..."

L65-72: Somewhere here it would be good to explain the difference between open and closed pingo systems.

L75: be more specific about what "the system" refers to.

L82: Avoid abbreviations as much as possible. If you need the, make sure to write them out the first time they are mentioned.

L89: Explain what a "through-talik" is and how is it different than a "talik".

L112: Reformulate. It is not clear from Fig 2b that the sediments are fine-grained or pre-Cenozoic.

L116: What does "these" refer to. All layers or just the surface layers < 70 m? Needs clarification.

L131: Explain the term "OSL"

L154: Specify period

L191: Clarify the source of these depth observations and the permafrost depths inferred from those observations.

L220: Expand abbrevations, i.e. explain what GMS 10.4 is

L252: Rephrase "validate the model we" (you are validating the model, not the code itself)

L253: Briefly explain the limitations of the analytical solutions (i.e. answering the question why can't you use these model for your study). Also summarize the findings about the model performance. Make sure to provide quantitative estimates of model performance (e.g. RMSE as in Supp. L79). Statements such as "relatively good performance" (Supp. L58) are not sufficient.

L271: Explain the term "A" in equation 6.

L281: Better define subzones. The text says that 12 subzones are defined. However, figure 12 does not show these clearly. Table 1 says nothing about subzones.

L293: Explain how you found out that porosity was the most important parameter

L311: Can a grid be one dimensional? Isn't it by definition 2D? I suggest rephrasing.

L312: It is not clear from Figure 4 where these 12 grids/columns are located. I suggest adding a point or arrow to identify the locations of the 12 grid/columns

L313. Table 1 shows site locations, not geological units. Do you mean table 3? Either way, explain how the names of the zones refer to the age. It is unclear. Provide a better connection to Figure 4. I suggest making a table with parameters for each of the 12 zones.

L319: Clarify that this grid is different from the one in the previous section.
L331: Clarify how you dealt with the active layer at the surface (i.e. unfrozen too, but also inactivated?)

L331: Clarify the meaning of "raw" and how it differs from non-raw simulations

L333: The interpolation is unclear. What was interpolated, time or space? Explain.

L335: Be more specific and clarify. Isn't it the permafrost aggradation in m/s that is inferred from the freezing front?

L340: Table 2 does not mentioned "intermediate" values, only, min/mean/max for three materials. Clarify what porosity values was used, and why those were used over other values.

L371: Cross reference with tables so that the reader can check this statement. Also provide the name of the layer.

L374: What about the thermal properties. Also cross-reference with table.

L389: Explain how you know this was artesian.

L390: Explain how you know that hydraulic pressures were below hydrostatic as well as the significance of this.

L392: Explain the 3 kyr catchments. Why 3 and not 4 kyr? What is the significance of these "catchments"

L401: Rephrase and clarify. It reads as if both artesian and non-artesian determine spring sites and has small QREq and high K. Earlier it is written that most of the area was artesian.

L445: Clarify if the characteristic length is equal to l, and why 200 m was selected.

L453: Clarify. Pertubations of what

L536: Be specific about which scenarios you are referring to here.

L547: Explain why max K values where used. In the previous text you argued for intermediate K values to be the most reasonable.

Comments on figures and tables ===================

Tables: Add captions

Table 2: Be more specific about what source you sued for what properties by using. The way it is done for porosity is great, do this for all parameters and other tables as well.

Figure S1 should be in the main article. However, I suggest creating different types of boxes in the flow chart to distinguish between data outputs and inputs, calculations/algorithms, and decisions. Make sure the groundwater model MODFLOW in the diagram

Figure 1: This is a great figure to illustrate the conceptual model. To improve., consider using another shading to identify permafrost

Figure 2 caption. Several words, terms, sites mentioned within parenthesis are unclear without context. Clarify the meaning/context of these, including: "temperature loggings", "Sarkofagen", "Breinosa", "geophysics". Be specific about what "Map data" you refer to, e.g. topographic data in panel a. Add at the end "description of the layers shown in cross sections A, B, and C". Add compass to the map since directions are discussed in the text later. Also point out the direction towards Longyearbyen

Figure 3. Explain the time axis. Time in relation to what (i.e. what is time 0 = present day). Use colors instead of dashed lines. The grey shaded are of driftwood arrival is almost impossible to see, use colors for this too.

Figure 4: Clarify that the inset maps shows the 1D simplifications of the model area. Choose one word to describe the 1D simplification (simplifications or interpretation). Explain the numbering of the subzones. Why does it start with zero and why is there a 10b. Why not go from 1 to 12? Explain everything shown in the figure, e.g. the red

arrows pointing out pingo locations.

Figure 5: This figures shows better what the zones are. Something like this is needed in figure 4 but indicating all zones. Why not just label the zones 1, 2, 3 and so on. But you also need to explain in the caption what all elements in the figure are (including the zone map)

Figure 6: Clarify that the uncertainty fields are uncertainty due to porosity. Why are symbols include in the middle chart, but not in the top and bottom chart? Make the chart constant.

Figure 7: The nine scenarios need to be explained in the method section. Explain how were they selected and parameterized. The figure is difficult to understand. Simplify and split up into multiple figures and tables. What does the colors mean for the DH4 borehole?

---

## Author Comment (AC2) · 9 Jun 2020

We would like to thank Referee#2 for a very constructive and thorough review that will help improve the quality of our manuscript. In this answer, we reply broadly to the general and major comments. We look forward to addressing the review in greater detail in a revised version of our manuscript.

We are very pleased to read that Referee#2 acknowledge our work as impressive and its use of many different data sources. Further, the Referee#2's view is that the manuscript has room for improvement when it comes to clarity and readability, and a number of specific changes to the manuscript are suggested. We agree that the

manuscript would benefit from the majority of the suggested changes and so, we will be pleased to abide by these in the revised manuscript.

Point-to-point reply to major comments:

1. In the revised manuscript, we will strive to use a clearer and more concise writing and further explain terms and concepts when needed.

2. We will reorganize the manuscript closer to a traditional structure as suggested.

3. Table captions will be provided in the revised manuscript.

4. We agree that adding a description of the model simulations in the method section would improve the overall readability and we will include such in the revised manuscript.

5. We will improve the explanation of the groundwater model simulations and their results according to the following:

- A descriptive overview of the model simulations and model will be added to the methods section as also suggested in comment 4.

- A description of how the 3 kyr was drawn with particle back-tracking will be provided and it will be stated that their significance is to visually illustrate slow groundwater velocities.

- Referee#2 is not sure which equivalent recharge rates (REq) were used for which scenario of the groundwater model. We read this confusion as due to a misunderstanding that the subzones of the groundwater model were run individually ("it is stated that each of the 12 zones were run with a different REq"). This was not the case (without specific reference to the text, we can also not find this statement in our manuscript). Instead, the entire surface of the model domain was assigned with the maximum, intermediate or minimum estimates of the REq rate as calculated from heat transfer model results (Fig. 6c). As illustrated on Fig. 6c, the maximum, intermediate or minimum REq value was not uniform, but decreased in the inland direction, and each subzone was thus

assigned with the relevant REq rate. As indicated by the y-axis on Fig. 7 and as stated in the text (Old version, L381-382) this resulted in three different total inflow rates of water to the groundwater model. In the revised manuscript, we will also express this clearly in the caption for Fig. 7.

- Referee#2 suggests that Fig. 7 is simplified and split up to several different figures. In our answer to Referee#1 (posted in the interactive discussion forum: https://www.the-cryosphere-discuss.net/tc-2020-7/), we presented a simplified version of Fig. 7. In order to ease comparison between the different scenarios, we would like to keep them in the same figure unless the editor agrees with Referee#2 that splitting it up is better.

With some insignificant exceptions, we will happily correct our manuscript according to the minor changes suggested by Referee#2.

---

## Author Response (AR1)

**Contents of this document**

**List of summarized manuscript changes**

We have summarized the manuscript changes in the list below. Subsequently, a list of changes to figures is provided.

- Model setup
  - The lower boundary of our heat transfer model (1DHT) was defined by a constant geothermal gradient at 300 m below ground level. As pointed out by referee#1, this definition allowed for considerable temperature changes at the model base, which implies that the geothermal heat flux was not constant. In the revised manuscript, we have extended the lower boundary to a depth of 1 km so that the temperature change here is close to constant (throughout the simulations). The new setup resulted in different simulated permafrost aggradation rates, which in turn caused changes to input for the groundwater model.
  - In the old version of the manuscript, the modelling approach did not consider dynamic storage effects. Due to the steady-state setup of the groundwater model, such are not possible to incorporate directly in the modelling of groundwater flow. In the revised manuscript, we have compensated for dynamic storage by applying a time-moving average to the estimated permafrost aggradation rates before calculating the present pressure contribution on the groundwater model. This new step in the modelling scheme is explained in a new paragraph added to the end of Sect. 4.3.4.
- As suggested by referee#2, the flow chart describing the model approach has been updated and moved from the Supplement to the main article as a new Fig. 4.
- Material properties
  - Hydraulic conductivities – The likely ranges of hydraulic conductivity for all pre-Cenozoic units have been tightened as suggested by referee#1:
    - Festningen Sandstone, Janusfjellet Subgroup and the detachment zone - The former lower end of the ranges is considered unlikely and the range now span across half an order of magnitude.
    - Carolinefjellet and u. Helvetiafjellet formations narrowed by applying a probability density function to the values measured by the Longyearbyen CO2 Laboratory Project. See details in the new version of the Supplement (S2).
  - Porosity of Janusfjellet Subgroup – In the first submission, we based the likely porosity range of Janusfjellet Subgroup on a single reference (Manger, 1963). As pointed out by referee#1, this parameter deserved more attention, because of its crucial importance when calculating the equivalent recharge from the simulated permafrost aggradation rates. In the revised manuscript, we improve the derivation of the likely porosity range by considering the estimated burial depth of Janusfjellet Subgroup and empirical works on the relationship between compressibility of clays and mudstones and effective stress. See further explanation and six new references in Sect. 4.3.2.
- More clearly defined manuscript structure
  - Section titles more clearly indicating intro, results, methods, etc.

- - Paragraphs moved to different sections
- Description of the groundwater model simulations is added to the methods section
- Tables
  - Captions added to tables
  - References to values in tables are indicated more clearly
- Corrections
  - A handful of errors pointed out by the reviewers have now been corrected.
- Improved readability
  - Many terms and concepts are now better explained to expand readership.
  - Sentences and phrasing that appeared unclear have now been rephrased.
  - To avoid confusion and improve readability, the usage of words like 'this' have been constrained where possible.
  - All abbreviations not used in common language are now explained.
  - Using the same word to describe different things or using different words to describe the same thing is now avoided.
  - Figures are made easier to read by explaining their content more thoroughly in the captions and by improving symbology and adding new components (see details in 'List of changes to figures' below).

**List of changes to figures**

- Figure 4 (old figure S1)
  - Moved to main article as a new figure 4
  - Modelling scheme clarified by using different box types and colour codes
  - New caption:
    - **"Figure 4** Schematic overview of the inner workings of the decoupled heat and groundwater model. Model setup, calculations and algorithms are indicated with sharp corners and bold text. Validation and comparison of simulations and observations are indicated with round corners and italics. Input and output data are indicated with rounded corners, grey background and normal text."
- Figure 1
  - Panel labels and descriptive texts concatenated and moved to upper left corners.
  - Shading added to indicate frozen ground
- Figure 2
  - 2a
    - The location of Longyearbyen is shown
    - Compass added
    - Tidal flat drawn
  - 2b
    - Simple geological time scale added to legend
  - New caption:
    - "**Figure 2 (a)** Map of Lower Adventdalen with the location of data resources, pingos and the Holocene marine limit. LP=Lagoon Pingo, LYRP=Longyear Pingo, FHP=Førstehytte Pingo, IHP=Innerhytte Pingo, RP=River Pingo. Core logs from boreholes S1–3 and D1–D7 (respectively, Gilbert et al., 2018, and Olaussen et al., 2020, and references therein), seismic lines (Bælum et al., 2012, and unpublished commercial lines from Norsk Hydro) and a geological map (Norwegian Polar Institute, 2019) were used to build the geological model (Fig. 5a) (see details in Hornum, 2018). Permafrost depth measurements at the Sarkofagen, DH4 and Breinosa sites are from Liestøl (1977), Braathen et al. (2012), and Christiansen et al. (2005), respectively. The freezing front depth at LP is from Yoshikawa and Harada (1995). Data used to develop the map including topography, glacial extent, and fluvial network by courtesy of Norwegian Polar Institute (2019). **(b)** Geological cross sections constructed based on the resources mentioned above. The Quaternary unit overly well-consolidated sedimentary strata of pre-Cenozoic age (i.e. Cretaceous or older). See Sect. 3.1 for a (hydro)geological description of the layers shown in the cross sections A, B and C."
- Figure 3
  - Curves drawn in different colours – not dash styles
  - Shaded area (minimal arrival of driftwood) made more visible
  - New caption:

- "**Figure 3** Holocene temperature reconstructions in and around Svalbard. Dashed grey area indicate time of minimal driftwood arrival (Farnsworth et al., 2020). The unit of the time axis is ka = $10^3$ years before present. **(a)** MSST curves (= MSAT, see text). Red line from Mangerud and Svendsen (2017). Orange line from Hald et al. (2007). Blue line from van der Bilt et al. (2018). **(b)** MAAT used in this work. Based on Mangerud and Svendsen (2017) and Førland et al. (1997) (see Sect. 4.3.3)."
- Figure 5 (old version Fig. 4)
  - We have clarified the meaning of the inset by making the labelling of the zones (now only called 'zones') more visible.
  - 1D simplifications are now only called so.
  - The labelling of the zones is now explained in the caption. Isochrones added to the figures.
  - Legend added to make the figure easier to read.
  - A zone map is now provided in a new panel b.
  - New caption:
    - "**Figure 5 (a)** 3D geological model of the subsurface below the valley floor in Adventdalen and vertical 1D simplifications (inset in upper right corner) below zones of the model area. The former determines the hydrogeological properties in the groundwater model (Table 3), whereas the latter determines the geothermal properties in the 1DHT model (Table 2). The domain of the 1DHT model extends to 1000 m b.g.l. (not shown). Deeper than 300 m b.g.l., geothermal properties were defined as for Janusfjellet Subgroup. The sea retreat reconstruction was inferred from absolute datings (Table 1) and is illustrated by valley floor exposure isochrones (red dashed lines). Pingos are indicated with blue arrows. **(b)** Vertical view of the model area showing the zonation. The aforementioned isochrones defined the zone names so that the valley floor exposure age of a zone is apparent from its name (i.e. zone 0-1 became sub-aerially exposed between 1 and 0 ka, zone 1-2 between 2 and 1 ka, etc.). "
- Figure 6 (old version Figure 5)
  - No changes
- Figure 7 (old version Figure 6)
  - Changed due to new model setup.
  - New caption:
    - "**Figure 7** (previous page) Present-day permafrost conditions as simulated by the 1DHT model. **(a)** Freezing front and permafrost depths. The upper and lower edge of the shaded area corresponds to the maximum and minimum limits of the porosity range, while the curve and the points corresponds to the intermediate (Table 2). **(b)** Rate of permafrost aggradation. The point symbols represent the present permafrost aggradation, while the shaded area is drawn by applying a time-moving average as described in Sect. 4.3.4. and represents the values used to calculate the equivalent recharge rates. *It is the -0.7 °C isotherm, which has been used to calculate this as the greatest phase change rate takes place at that temperature. **(c)** Equivalent recharge rate (net rate of loss in pore space). The

points on the curve and the outer edges of the shaded area represent the recharge rates, which were assigned to the corresponding zones in the groundwater model."

- Figure 8 (old version figure 7)
  - Changed due to new model setup
  - Legend for the colour fill indicating heads in m a.g.l. is made identical for all scenarios and moved to bottom of the figure.
  - New caption:
    - **"Figure 8** (next page) Groundwater model simulations from all 9 scenarios. The individual diagrams are sorted so that the hydraulic conductivity increase along the right-hand axis **(scenarios Sc1**–**3*x*)** and the equivalent recharge produced by permafrost aggradation increase along the left-hand axis **(scenarios Sc*X*a**–**c)**. On each diagram, the following simulation results are illustrated: Heads from the uppermost grid layer are shown in m a.s.l. by isopotential contours (note that the colour scales are different) and in m a.g.l. by the colour fill (see scale at the bottom of the figure). The latter indicates if artesian conditions are simulated (reddish) or not (blueish). The outflow distribution and discharge rates are illustrated by pie charts with the location of the discharge points (pingos and fjord) indicated on **(1c)**. *For minimun $Q_{REq}$ scenarios (**1a**, **2a** and **3a**) part of the outflow is caused by basal permafrost thaw (Figs. 7b–c). Flow patterns are illustrated by thin blue lines, which each depict pathways of particles released in the uppermost grid layer. The mean pore water velocities, shown in the upper left-hand corners, were calculated from the aforementioned particles and using the same porosities for all scenarios (Table 3). The areas outlined with thick dashed lines show where each outlet point were simulated to receive water from during 3 kyr. In the lower right-hand corner, the simulated head in well DH4 is illustrated on a range plot with the range defined as deduced by Braathen et al. (2012). The colour of the bar indicates if the simulated head falls within (green) or outside the deduced range with less (yellow) or more (red) than 10 % of the total range. The location of DH4 and the pingo springs is marked on **(1c)**."
- Figure 9 (old version figure 8)
  - Changed due to new model setup
- Figure 10 (old version Figure 9)
  - Changed due to new model setup
- Figure 11 (old version figure 10)
- Figure 12 (old version figure 11)

**Point-to-point reply to reviews**

On the following pages, we give a point-to-point reply to the reviews. Please note that the original referee comments are written in **bold red text**, answers to comments are written in *italics*, and changes to the manuscript are written in *blue italics.*

**Reply to comments by referee 1# (Melissa Bunn)**

**Summary:**

**In this paper, the authors investigate permafrost aggradation and the associated increase in subpermafrost groundwater pressures over millennial scales as the potential cause of pingo springs in a high Arctic valley (Aventdalen, Svalbard). Continuous permafrost, high desert conditions, and a lack of wet-based glaciers in the adjacent highlands preclude recent groundwater recharge as a source of the spring water. Using a 1D heat flow model the authors quantify potential rates of permafrost aggradation. This aggradation is then related to a water flux which is applied as recharge in a 3D groundwater flow model. These processes are fully decoupled. The groundwater flow model represents the steady state flow of groundwater to the pingos (and the adjacent Fjord) that results from the additional subpermafrost water flux. Although validating field data is limited to sporadic spring flow measurements and hydraulic head measurements at a single borehole, what is available is compared to this data to support the development of the model and the proposed conceptualization of the pingo spring flow.**

**The mechanism proposed by the authors is new, and their use of numerical models to illustrate and quantify this mechanism is of value. The discharge of subpermafrost groundwater to the surface has the potential to introduce methane to the environment and other solutes to freshwater systems. Understanding of the mechanism that generates the driving hydraulic heads in a variety of geological settings improves our ability to forecast future conditions under a changing climate. Overall the conceptual model and the numerical approach is well presented. However, the paper gives the sense that the modelling work proves the conceptual model to be correct. In general, assumptions are made in numerical modelling to align the numerics with the conceptualization. The modelling presented in this paper is no different, and as such, the model output does not prove the conceptual model to be correct. Some factors that warrant further investigation to support the numerical modelling assumptions are detailed below. Value could be added to the paper by using the model to further explore the physical factors required to form pingo springs under this conceptual model.**

*Melissa Bunn's review of our manuscript clearly shows that she has studied our work carefully and evaluated it very professionally. We are pleased to read that the reviewer acknowledges the novelty of the pingo-forming mechanism we propose and that she overall find that our research is well presented.*

*The reviewer is of the impression that the paper "gives the sense that the modelling work proves the conceptual model to be correct". We did not intend to make such claim, but we acknowledge the reviewer's impression and hope that this revision improves that. It appears to us that a number of the reviewer's quite specific comments regarding the numerical modeling work arise from the above impression. Although this was not intended, these comments are still of great value for improving our paper.*

*Freezing pressure from permafrost aggradation is well known from closed-system pingos, but has not previously been considered for open-system pingos as this research does. This is, as also noted by the reviewer, a novelty. Based on our investigation, we argue that permafrost aggradation deserves*

*attention as a driver for deep permafrost springs and open-system pingos. We do so because the modelling results suggest that the conceptual model represents a feasible mechanism for how open-system pingos can form. However, we do not think that it is the only option for the pingos in Adventdalen and we have tried to clarify this view in the revised manuscript. All of the following changes and corrections were done with this in mind and the ones directly below are examples of this.*

Old version, Abstract, Lines 19:

*"Our results show that the pingos in lower Adventdalen easily conform to this conceptual model."*

New version:

*"Our results suggest that the conceptual model represents a feasible mechanism for the formation of open-system pingos in lower Adventdalen and elsewhere."*

Old version, Sect. 6.3, Line 530:

*"The amount of hydrogeological data from Adventdalen was insufficient for automatic calibration of model parameters."*

New version:

*"The amount of hydrogeological data from Adventdalen was insufficient for automatic calibration of model parameters and the model simulations should therefore at best taken as possible scenarios for the conditions in Adventdalen."*

**The boundaries of the numerical groundwater flow model create a closed "bathtub" like system. Although the amount of recharge added is quite low (<1 mm/year), the presence of permafrost throughout the top of the model domain, a lower boundary that is within 100 m of the base of the permafrost, and no flow across lateral boundaries, leaves only the Fjord and the pingo springs as discharge points. Additional details, or discussion would be useful in supporting the boundary selection for the groundwater flow model as follows:**

- **The paper investigates the effect of the lower boundary position by lowering the model depth by 100 m. A more illustrative demonstration of the effect of the lower boundary may be to lower the boundary to the detachment zone that separates the upper overpressured and lower underpressured groundwater flow systems (i.e., to a depth where there is field evidence of a no-flow boundary);**

*While we agree with the reviewer's comment that a lower boundary defined by field evidence would be preferred to the somewhat arbitrary approach, we have used, we do not find such in the investigated system. The reviewer suggests that the detachment zone could define the lower model boundary, but this is found at such shallow depths in the Eastern part of the model area ( ~ 50 m b.g.l., Fig. 2b) that this would not be part of the groundwater model domain (after the permafrost layer has been removed). Seismic investigations at IHP suggest that pingo spring water rise vertically from rocks deeper than 50 m (Rossi et al., 2018) and thus likely from below the detachment zone. At the same time, hydrogeochemical*

similarities and consistent trends (water type and concentrations of Cl- and stable water isotopes) from all pingo springs suggest that they formed in the same hydrogeological system (Hodson et al., In review). Complete hydrological separation between the lower (NW) and upper (SE) part of the system therefore appears unlikely.

Although no field evidence exists for a uniform model domain depth, it may be justified if we assume that groundwater flow at least partly is controlled by secondary permeability induced by glacial loading and unloading (Leith et al., 2014). We have added this consideration in the discussion of the model extent (see changes in the revised manuscript below the answer to the next comment).

- **There is no evidence provided for the presence of hydraulic divides at the flanks of the valley that would support the use of lateral no flow boundaries in the groundwater flow model as specified. This boundary prevents any regional flow in to or out of the valley. Were groundwater flow to follow the modest slope of the formations, flux through the Festningen Member may be sufficient to dissipate the recharge flux specified in the model. If field data is not available, the sensitivity of model results to deeper regional flow across the valley should be explored;**

There is indeed no field evidence for the presence of hydraulic divides at the flanks of valley and we hence write that the lateral model extent is "somewhat arbitrary" (Old version, Section Line 519).

The reviewer suggests that we investigate how regional flow through Festningen member or other presumably permeable hydrogeological units (i.e. fractured bedrock) may affect dissipation of the recharge flux. However, do to its stratigraphical orientation, the Festningen member does not cross the model domain and as such, it cannot facilitate regional flow across the system (Figs. 2b and 4): Eastwards, it is not present, and in the Northeastern part of the domain, it is cut by the surface/Quaternary strata.

Although no field evidence exists for a "bathtub" system, there is likewise no evidence for the contrary. As should be clear from our manuscript, we focus our study on effects of freezing expansion. Our model design attempts to isolate this mechanism (taking variable Holocene temperatures and sensitivity analyses into account) and we attempt to emphasize this in the revised manuscript:

Old version, Section 6.2.2., Lines 518-529:

 "Given the lack of known geological boundaries or groundwater divides, the lateral extent of the model domain was defined using a simplified outline of the HML, but this may be somewhat arbitrary. Due to Early Holocene warming (Fig. 3), the 1DHT simulation results showed that continuous frozen ground in Adventdalen is likely younger than 6.5 ka even where the valley floor is older (Fig. 5). This is supported by geomorphological and geochronological evidence (Humlum, 2005). As such, there seems to be no reason why permafrost dynamics outside the HML should be markedly different from that in the up-valley part of the model area. Based on the above, it is possible that basal permafrost aggradation goes on beyond the model area (HML) and model simulations may have underestimated the freezing-induced pressures affecting spring discharge.

*The dominantly low-permeable groundwater system challenged a physically determined lower boundary for the model domain. From the significant low-pressures observed in deeper stratigraphic layers (~ 800 m b.g.l., Braathen et al., 2012), we inferred isolation of the investigated groundwater system from that below and simply assigned the base to a depth of 300 m b.g.l. By simulating scenarios with a lower base of 250 and 400 m b.g.l., we found that simulation results did not change significantly (< 1 % deviation of simulated heads and discharge rates)."*

New version:

*"The boundary conditions of the groundwater model define a bathtub-like system with the pingo springs and the fjord as the only discharge points, and with the expansion of water upon freezing within the model regime as the only source of hydraulic pressure. In reality, the hydrological system in Adventdalen may not entirely conform to this description as groundwater flow across the model boundaries cannot be rejected. Additional recharge could, for example, occur through microcracks below the valley floor induced during glaciation (Leith et al., 2014). Likewise, hydraulic pressures may, to a greater extent than simulated, dissipate directly to the fjord through unknown pathways. In this respect, our model serves to isolate the pressure effect of freezing expansion in Adventdalen and systems like it. This should be taken into account before drawing site-specific conclusions from the modelling results.*

*Due to Early Holocene warming (Fig. 3), the 1DHT simulation results showed that continuous frozen ground in Adventdalen is likely younger than 6.5 ka even where the valley floor is older (Fig. 6). This is supported by geomorphological and geochronological evidence (Humlum, 2005). As such, there seems to be no reason why permafrost dynamics in the valley bottom outside the HML should be markedly different from that in the up-valley part of the model area. Based on the above, it is possible that basal permafrost aggradation goes on beyond the model area (HML) and model simulations may have underestimated the freezing-induced pressures affecting spring discharge.*

*The dominantly low-permeability groundwater system challenged a physically determined lower boundary for the model domain. From the significant low-pressures observed in deeper stratigraphic layers (~ 800 m b.g.l. at DH4, Braathen et al., 2012), we inferred isolation of the investigated groundwater system from that below and simply assigned the base to a depth of 300 m b.g.l. By simulating scenarios with a lower base of 250 and 400 m b.g.l., we found that simulation results did not change significantly (< 1 % deviation of simulated heads and discharge rates)."*

- **As stated by the authors, the drain boundaries used to represent the pingo springs are placed within the upper most active cells closest to the spring, but within the Festningen Sandstone. Additional details and discussion of this placement would be of use. Figure 2(a) and Figure 4 indicate that the sandstone is not present at the Innerhytte Pingo. Figure 7 shows that the drain associated with the Førstehytte Pingo is opposite the valley axis from the surface expression. It is understood that the fractured nature of this sandstone could permit the required subhorizontal flow to the pingo; however the paper would benefit from additional discussion of this conceptualization (what is the inferred orientation of the fracturing that allows formation of the pingos). The sensitivity of the model results to the geological unit that the drain boundary is placed in should be discussed;**

*The reviewer here point to an error in the first version of our manuscript. The drains representing the pingo springs were placed within the uppermost active cells closest to the spring, but within Festningen Sandstone if present in the underlying stratigraphy. As the reviewer rightfully notes, Festningen Sandstone is not present at Innerhytte and River Pingos, and the drains were here placed in the uppermost active cells. The reasoning behind placing the drains within Festningen Sandstone is that the permeability of this unit presumably makes it a pathway for groundwater flow. Assessing the fracture orientation within this unit is beyond the scope of this paper, and its hydraulic conductivity was thus assumed uniform.*

Old version, Section 4.3.4., Lines 325-326:

*" […] the drains were assigned to the uppermost active cells located closest to springs, but within the conductive Festningen Sandstone."*

New version:

*" […] the drains were assigned to the uppermost active cells located closest to springs, but within the conductive Festningen Sandstone if present in the underlying stratigraphy (i.e. Lagoon and Førstehytte Pingos, Fig. 5a)."*

- **The boundary representing the Fjord is described as being applied to the relevant cells. Please describe this assignment in more detail (are the boundaries assigned to the upper most active layer only, or are they assigned to several layers to the approximate seafloor depth in the Fjord). As the boundary at the Fjord represents the highest flux from the model (approximately 40% to 90% of the total flux) model results can be expected to be very sensitive to the vertical location of this boundary, and should be investigated further; and,**

*The reviewer is not sure whether the fjord BC is assigned only to the uppermost active cells only or extended to the approximate depth of the seafloor. Both are true because the fjord depth is less than 5 m. It is not stated in the first submission, but the model extent towards the fjord aligns with the approximate terminus of the tidal flat. We have now drawn the tidal flat on Fig. 2a and made the following correction:*

Old version, Section 4.3.1., Lines 276-277:

*"The horizontal model boundary was a simplified outline of the valley bottom and the lower boundary was at 300 m b.g.l. (Fig. 4)."*

New version

 *"Towards northwest, the model covers the tidal flat (Fig. 2a), but no other sea-covered areas were included. Elsewhere, the horizontal model boundary was a simplified outline of the HML. The lower boundary is at 300 m b.g.l. (Fig. 5a)."*

- **The potential for subpermafrost discharge to the Adventdalen River has not been considered in the conceptual model development. Rossi et al. (2017) suggest that this discharge may occur**

**near the Innerhytte Pingo. Although the rate of subpermafrost discharge to this River may be low, the potential for it to occur should be considered in the overall balance of flows.**

*We are aware of (Rossi et al., 2018) and have now included this reference. The reviewer writes that the potential for subpermafrost discharge to the Adventdalen River has not been considered. Based on seismic surveys (Rossi et al., 2018) suggest the discharge area at Innerhytte Pingo (IHP) may extent some meters below Adventdalen River. In the summer when Adventelva is active, a minor amount of spring discharge directly to the river could very well be overseen, but during winter when the river is dry, all perennial discharge points are easily recognized by continuously growing icings. At IHP, there is only one icing observed (covering the pingo apex and extending down to the river). If the reviewer would like it, we will be happy to include a speculative uncertainty bar on Fig. 8, which could account for the potential minor discharge to the river. However, based on our observations of outflow from the winter icing, this would be less than Liestøl's (1977) estimate of 1 L/s, which is already considered in the balance of flows.*

*Elsewhere than at IHP, perennial discharge to the river would results in winter season icings. As these have not been observed, we do not regard discharge of subpermafrost groundwater to the river to be plausible.*

*No changes to the manuscript were made based on this comment.*

**In general, further investigation of the effect of these various boundary conditions on the groundwater model results are required before it could be concluded that the model results show that the basal permafrost aggradation produces the hydraulic pressures to sustain the pingo spring water outflows as the authors have stated at the start of Section 7.**

*The reviewer here refers to the first paragraph of Sect. 7. This paragraph was intended to refer to open-system pingo formation in general, and not specifically to the pingos in Adventdalen. As such, we did not mean to give the impression that the basal permafrost aggradation, per se, produces hydraulic pressures that sustain the outflow in Adventdalen. Instead, we argue that the model experiments show that permafrost aggradation alone may drive open-system pingo and subpermafrost spring systems, if conditions like for the modeling experiments exists.*

Old version, Sect. 7, Lines 622-623:

*"Results from the decoupled heat and groundwater model show that millennial-scale basal permafrost aggradation may alone produce hydraulic pressures sufficient for the formation of pingos and their spring water outflows."*

New version

*"Results from the decoupled heat and groundwater model show that millennial-scale basal permafrost aggradation may alone produce hydraulic pressures sufficient for the formation of pingos and their spring water outflows when the right conditions are met."*

**The observed hydraulic head at DH4 is stated to range from 9 m to 60 m above hydrostatic. It is unclear if this range is due to temporal variability, or the range in the correction for the effect of dissolved gasses. With limited field data for model validation, this warrants further discussion. As shown on Figure 7 (2a, 3b, 3a) simulations in which the hydraulic head at DH4 is on the lower end of this range (and with the lower to middle recharge flux) do not produce sufficient flow at the pingo springs.**

*The range of the "observed" head at DH4 is uncertainty. The head range is calculated from a single hydraulic pressure (range) estimate by (Braathen et al., 2012) excluding the potential pressure effect of dissolved gasses. This was already stated in the first submission in Sect. 3.1 lines 145-148. (Braathen et al., 2012) deduce the plausible hydraulic pressures based on outflow from the well during drilling. We have now expressed this explicitly:*

Old version, Line 144:

*"Nearby, […]"*

New version:

 *"Based on artesian outflow during a drilling experiment nearby, …"*

**The equivalent recharge applied to the model ranges from 25.4 m3 /day to 56.7 m3 /day. This range is related to the porosity of the formation through which permafrost aggradation is occurring. Based on the 1D columns shown on Figure 4, and the model results shown on Figure 5, much of this aggradation would occur within the shale units. The porosity of the Janusfjellet subgroup has been derived from Manger (1963). How was the range from 0.1 to 0.3 selected from the values provided in Manger (1963). The higher porosity units in this reference are related to high clay content shales or claystones. Is this valid for the Janusfjellet formation? Given that the higher porosity ranges were required to produce a water flux that could sustain the pingo flows, further details should be provided on the derivation of these values.**

*Indeed, by being the dominant geological unit Janusfjellet Subgroup is where most permafrost aggradation takes place. The porosity values (0.1-0.3) were chosen because Agardhfjellet Formation (Janusfjellet subgroup) is the Svalbard analogue to the Kimmeridge Clay Fm, which in (Manger, 1963) is reported to have porosities between 0.19-0.307 (only two samples from one outcrop location). We realize that basing the most crucial parameter for permafrost aggradation (and thus equivalent recharge) on so little empirical data is insufficient. We have now regarded the estimated burial depths of the Janusfjellet Subgroup (Grundvåg et al., 2019; Marshall et al., 2015) and inferred a porosity range based on empirical works on compressibility of clays and mudstone as function of effective vertical stress (Burland, 1990; Okiongbo, 2011; Yang and Aplin, 2004). Using these new references, we find a possible porosity range of 0.08 to 0.3. See changes to the manuscript below the answer to the comment regarding the hydraulic conductivity range of Carolinefjellet and Helvetiafjellet formations.*

**How is the lower boundary of the 1D heat transfer model specified? If the geothermal gradient from surface is maintained, does that imply that temperature of the lower boundary changes with time?**

**How would the rate of permafrost aggradation be changed if the depth of the 1D model was extended such that the heat flux at the bottom boundary could be kept constant through time? How would a cessation of permafrost aggradation up valley effect results?**

*The lower boundary (z=300 m) has a BC defined by a geothermal gradient of 0.025 C/m. Because the geothermal properties in the lowermost cell are constant throughout all simulations (i.e. the pore water does not freeze), this BC is equivalent to a constant heat flux. The reviewer is right that the temperature of the lower boundary changes with time, and we agree that this is problematic. Keeping the remainder of the model setup as before, we have now lowered the lower boundary to a depth of 1 km, so that the temperature change here is close constant (ΔT less than 0.22, 0.42, and 0.65°C for max, mid, and min porosity scenarios, respectively). The new setup simulates results in different simulated permafrost aggradation rates and we thank the reviewer for improving the estimate of these. Correction made and included in the modeling experiments of the revised manuscript.*

Old version, Sect. 4.3.3, Lines 311-317:

*"The model domain contained 12 one-dimensional grids, each 300 m long and consisting of 150 cells with a height of 2 m. Each individual grid was associated to the model area zones (Fig. 4) and the geothermal properties were defined accordingly. The names of the zones refer to the age of subaerial exposure (Table 1), which defined the simulation run time (e.g. for zone 0-1 the simulation period was 0.5 to 0 ka, for zone 1-2 it was 1.5 to 0 ka, etc. For zone 10, the simulation period was 10 to 0 ka). The initial ground temperature distribution followed the geothermal gradient reported by Liestøl (1977) (0.025 °C m$^{-1}$) from a surface temperature of 0 °C. At any subsequent time, the lower boundary condition was defined from the same geothermal gradient."*

New version:

*"The model domain contained 12 columns, each 1000 m long and consisting of 500 cells with a height of 2 m. One column was associated to each of the model area zones and the geothermal properties were defined according to the associated geological 1D simplifications (see insert on Fig. 5b). Deeper than 300 m b.g.l., the properties were that of Janusfjellet Subgroup. The simulation run time was defined by the valley floor age inferred for that zone (Fig. 5), so that, for zone 0-1 the simulation period was 0.5 to 0 ka, for zone 1-2 it was 1.5 to 0 ka, etc. For zone 10, the simulation period was 10 to 0 ka. The initial ground temperature distribution followed the geothermal gradient reported by Liestøl (1977) (0.025 °C m$^{-1}$) from a surface temperature of 0 °C. At any subsequent time, the lower boundary condition was defined from the same geothermal gradient resulting in a basal temperature change of less than 0.65°C."*

**In Table 3, the rock unit hydraulic conductivities derived from literature (the Festingen sandstone, the Janusfjellet subgroup, and the detachment zone) range within one order of magnitude across the three scenarios. While this could be considered a large range in this type of study, comparison to the observed hydraulic head at DH4 indicates that Scenario 1 (low hydraulic conductivity) values are unlikely, leaving a more reasonable half order of magnitude range.**

*We agree with this consideration and narrow the conductivity range of these units in the modeling experiments of the revised manuscript.*

Old version, in Table 3:

| | | | | | |
|---|---|---|---|---|---|
| *Festingen Sandstone* | *Fractured sandstone* | $10^{-2}$ | $5 \cdot 10^{-2}$ | $0.1$ | $0.1$ |
| *Janusfjellet Subgroub* | *Shale* | $10^{-4}$ | $5 \cdot 10^{-4}$ | $10^{-3}$ | $0.1$ |
| *Detachment zone* | *Fractured shale* | $10^{-3}$ | $5 \cdot 10^{-3}$ | $10^{-2}$ | $0.1$ |

New version, in Table 3:

| | | | | | |
|---|---|---|---|---|---|
| *Festingen Sandstone* | *Fractured sandstone* | $5 \cdot 10^{-2}$ | $7.5 \cdot 10^{-2}$ | $0.1$ | $0.1$ |
| *Janusfjellet Subgroub* | *Shale* | $5 \cdot 10^{-4}$ | $7.5 \cdot 10^{-4}$ | $10^{-3}$ | $0.1$ |
| *Detachment zone* | *Fractured shale* | $5 \cdot 10^{-3}$ | $7.5 \cdot 10^{-3}$ | $10^{-2}$ | $0.1$ |

**For the rock units with field data (the Carolinefjellet and Hevetiafjellet formations), the hydraulic conductivities applied range over two orders of magnitude. Does this range represent the maximum and minimum of tested values? It would be of value to plot the probability density function for the hydraulic conductivity values for each formation, selecting the geometric mean as scenario 2, and a more realistic percentile as scenarios 1 and 3 to tighten the potential range for these formations. As stated on line 406, the hydraulic conductivity is the most important parameter in determining the distribution of outflows between the pingos and the Fjord. Assignment of this parameter should be constrained where possible.**

*We agree with the author that the hydraulic conductivity ranges should be tightened where possible. The proposed approach is included in the revised manuscript and the ranges tightened accordingly.*

Old version, Section 4.3.2., Lines 290-301

[revised manuscript text omitted]

Old version, Sect. 4.3.2, in Table 2:

*Shale    4.73·10$^7$    800    2600    0.1$^{III}$    0.2$^{III}$    0.3$^{III}$*

New version

| | | | | | | |
|---|---|---|---|---|---|---|
| *Shale* | $4.73 \cdot 10^7$ | *800* | *2600* | *0.08[III]* | *0.19[III]* | *0.3[III]* |

Old version, Section 4.3.2., in Table 3:

| | | | | | |
|---|---|---|---|---|---|
| *[I]Carolinefjellet Fm* | *Sandstone* | $10^{-4}$ | $10^{-3}$ | $10^{-2}$ | *0.1* |
| *[I]u. Helvetiafjellet Fm* | *Sandstone* | $10^{-4}$ | $10^{-3}$ | $10^{-2}$ | *0.1* |

 New version

| | | | | | |
|---|---|---|---|---|---|
| *[I]Carolinefjellet Fm* | *Sandstone* | $2 \cdot 10^{-4}$ | $5 \cdot 10^{-4}$ | $10^{-3}$ | *0.1* |
| *[I]u. Helvetiafjellet Fm* | *Sandstone* | $2 \cdot 10^{-4}$ | $5 \cdot 10^{-4}$ | $10^{-3}$ | *0.1* |

In new version of Supplement

*"S3       Hydraulic conductivity of Carolinefjellet and Helvetiafjellet Formations*

*The vertical permeability, $\kappa_v$, of the sandstone-dominated Carolinefjellet and Helvetiafjellet formations were measured as part of the Longyearbyen $CO_2$ Laboratory Project (Olaussen et al., 2020, and references therein). The small-scale horizontal permeability, $\kappa_h$, for sandstones is typically a factor two higher than $\kappa_v$ (Domenico and Schwartz, 1998) and we converted the horizontal hydraulic conductivity, $K_h$, accordingly (Eq. 7). The ranges of hydraulic conductivity of these units were defined by the 25 %, 50 % and 75 % percentiles of a Weibull probability fit to the measured values (Fig. S5).*

[Figure]

**Figure S5** *Weibull fit to measured hydraulic conductivities of Carolinefjellet and Helvetiafjellet formations. Original data from the Longyearbyen CO₂ Laboratory Project (Olaussen et al., 2020, and references therein).*"

**On line 515 it is stated that the steady-state assumption for the groundwater flow model results in an underestimate of the present-day pressures. This statement may oversimplify the transient groundwater dynamics that could occur as permafrost aggrades and the sea level retreats. Permafrost aggradation is highest in proximity to the Fjord, which is also where the greatest potential for discharge to the Fjord occurs. It is possible that any excess pressure would be dissipated as the sea level retreats, and that a transient simulation may not show higher pressures.**

*As stated in Section 3.2 (lines 182-183), sea-level reached close to present levels ~ 5 ka. As such, the sea retreat since then is explained to progradation of the fluvio-deltaic system. Pressure dissipation due to land emergence is therefore not plausible. The statement referred to by the reviewer (Old version, Line 515-516) considers that the omission of dynamic storage effects implies that the predominantly greater permafrost aggradation rates simulated in the past are not taken into account in our model. Effectively, this implies an underestimate of the equivalent recharge. In the revised manuscript, we account for this by applying a moving average to the simulated permafrost aggradation, where the time window is determined by the range of possible adjustment times.*

New version, paragraph added to Sect. 4.3.4.:

*"The only source of water in the groundwater model was defined from the basal permafrost aggradation rate simulated by the 1DHT model and assigned as recharge to the uppermost active cells in the model domain. To compensate for the lack of dynamic storage effects in the steady-state model, we applied a moving time-average to the simulated basal permafrost growth (or decay) before calculating the*

*recharge equivalent (Eq. 6). The time window of the moving average was based on the possible range of the adjustment time, $t_a$, which is the time needed for fluids to redistribute to a pressure perturbation (e.g. Neuzil, 2012; Šuklje, 1969):*

$$t_a = l^2 \, S_s \, K^{-1} \qquad\qquad\qquad\qquad (8)$$

*where $l$ is half of the shortest dimension of the system (the characteristic length), $S_s$ is the specific storage, and $K$ is the hydraulic conductivity. We found $t_a$ to be shortest in the vertical dimension, but assumed that hydraulic pressures could only dissipate in the horizontal dimension after the formation of continuous permafrost no earlier than 6 ka (Humlum, 2005; this research). Specifically, we estimated the horizontal $t_a$ to be between 20 and 19000 yrs. To quantify this, we used a characteristic length of 1 km. For $S_s$, we used a matrix compressibility of $7 \cdot 10^{-10}$ to $7 \cdot 10^{-8}$ Pa$^{-1}$ (based on common estimates for fractured rocks, e.g. Domenico and Mifflin, 1965; Domenico and Schwartz, 1998; Fitts, 2002) yielding a $S_s$ of $7 \cdot 10^{-6}$ to $7 \cdot 10^{-4}$ m$^{-1}$ (in line with literature values, c.f. Singhal and Gupta, 2010). For K, we used the values estimated for the dominating geological unit (Janusfjellet Subgroup, Table 3). The time window used to compensate for dynamic storage effects were defined from the above, but no longer than the age of permafrost (i.e. 6000 yrs or less)."*

**On line 635 it was stated that simulated flows to the pingo springs are likely underestimated as basal permafrost aggradation outside of the model domain is not included. Were this aggradation to contribute to the pingo spring flows, it would imply that the lateral boundaries of the model domain are not hydraulic divides (i.e., no flow boundaries). This statement should be reconciled with the boundary selection.**

*We have not changed the boundary conditions of the groundwater model. See comments above.*

**Technical Corrections:**

**Line 13: ...wet-based glaciers are not present in the adjacent highlands**

*done*

**Line 18: ..and groundwater (3D -Steady-state)**

*done*

**Line 229 Equation 1: The _z in the denominator should be _z2**

*done*

**Line 235: ...heat conduction will flow heat will be conducted through a matrix of solids (i.e sediment or rock) and liquid water, ice or a mixture**

*done*

**Line 288: ..The fraction of liquid water**

*Done (we assume this correction concerns line 238)*

**Line 247: ...When temperature change occurs**

*done*

**References**

**Rossi, G., Accaino, F., Boaga, J., Petronio, L., Romeo, R., Wheeler, W., 2018. Seismic survey on an open pingo system in Adventdalen Valley, Spitsbergen, Svalbard. Near Surface Geophysics 16, 89–103. https://doi.org/10.3997/1873-0604.2017037**

**Author response to Referee #2**

**In this paper, the authors investigate the question: "Why are there pingos in Adventdalen when there seem to be no groundwater recharge? They do this with a conceptual model, and find that the observed pingos are sustained by groundwater supply with residence times exceeding the duration of Holocene. In addition to the modeling, the work relies on a wide range of field observations.**

**The work presented in this paper is impressive. A new model is developed and data from many different sources are used in the study. However, the manuscript would benefit from revisions to improve readability and clarity. Currently, the manuscript contains many terms and context that need to be explained to expand readership beyond immediate experts and to the diverse backgrounds of the readers of The Cryosphere. The minor comments below provide many examples.**

We are very pleased to read that Referee#2 acknowledge our work as impressive and its use of many different data sources. Further, the Referee#2's view is that the manuscript has room for improvement when it comes to clarity and readability, and a number of specific changes to the manuscript are suggested. We agree with the majority of these and abide to them in the revised manuscript.

1. **Many terms and concepts are provided without much context. I give several examples in the minor comments.**

*We have strived to use a more concise language in the revised manuscript. The reviews of our manuscript have been very helpful with this.*

2. **Organize the manuscript into sections according to the traditional structure of a scientific paper (Abstract, intro, study site, methods and data, results, discussion, conclusions). In the current paper, there is no clearly defined method section and sections 2 to 4 gradually move from introduction to results. a. For example, the section 3.2 title suggests this section contains background information about the study site. However, besides that background information it also explains how climate history was reconstructed. Separate these two so that the background info on lines 150-154 goes in a study site description section, and the rest in a methods subsection about reconstructed temperatures b. Another benefit of a dedicated method section is that you can provide an overview of the methods at the beginning providing the reader with a sort of road map.**

*The major headings now read:*

- *Abstract*
- *Introduction*
- *Conceptual model of permafrost aggradation driven pingo formation*
- *Study site*
- *Method – Numerical modelling*
- *Results*
- *Discussion*

- *Conclusions*

*We have moved the information of how the Holocene temperature was constructed to the method subsection 4.3.3. and rephrased both sections:*

Old version, Sect. 3.2., L156-165:

*"The mean summer air temperature (MSAT) has consistently been 10 °C warmer than the MAAT (on the 30-yr scale, Førland et al., 1997).*

*Holocene temperatures on and around Svalbard are relatively well constrained by fossil-based temperature reconstructions. Mangerud and Svendsen (2017) infer a mean summer sea temperature (MSST) curve from the distribution and $^{14}$C–dating of thermophilous bivalves and point out that the MSST is essentially identical to the MSAT. As illustrated on Fig. 3a, their MSST curve is largely in agreement with MSST temperature reconstructions from west and southwest of Svalbard (van der Bilt et al., 2018; Hald et al., 2007). Assuming that the present difference between MAAT and MSAT was alike for the entire Holocene, we use a MAAT reconstruction inferred by subtracting 10 °C from the MSST curve by Mangerud and Svendsen (2017) for the modelling of ground temperatures in this work (Fig. 3b). We choose to rely on Mangerud and Svendsen (2017) because; a) their curve is more local to our field area than the alternatives and; b) the suggested timing of the Holocene thermal minimum at ~ 3 to 2 ka is in agreement with the maximum of perennial or semi-permanent land fast sea ice at ~ 2.5 to 2 ka inferred from the minimal occurrence of dated driftwood (Dyke et al., 1997; Farnsworth, 2019; Funder et al., 2011). Furthermore, their curve is better supported by geomorphological evidence of glacier dynamics (Farnsworth, 2019)."*

New version:

*"For the entire temperature record, the mean summer air temperature (MSAT) has consistently been 10 °C warmer than the MAAT (on the 30-yr scale, Førland et al., 1997). Further back in time, Holocene mean summer sea temperatures (MSST) in and around Svalbard are relatively well constrained by fossil-based temperature reconstructions (Fig. 3a, van der Bilt et al., 2018; Hald et al., 2007; Mangerud and Svendsen, 2017). Mangerud and Svendsen (2017) point out that the MSST is essentially identical to the MSAT."*

Old version, Sect. 4.3.3, L317:

*"The Holocene temperature curve (Fig. 3) defined the upper boundary condition."*

New version:

*"The upper boundary condition was defined by the Holocene MAAT curve presented in Fig. 3b. Assuming that the present 10 °C difference between MAAT and MSAT (Førland et al., 1997) was alike for the entire Holocene, we constructed this curve (Fig. 3b) by subtracting 10 °C from the MSST curve by Mangerud and Svendsen (2017) (Fig. 3a). As illustrated on Fig. 3a, their MSST curve is largely in agreement with MSST temperature reconstructions from west and southwest of Svalbard (van der Bilt et al., 2018; Hald et al., 2007). We chose to rely on Mangerud and Svendsen (2017) because; a) their curve is more local to our*

*field area than the alternatives and; b) the suggested timing of the Holocene thermal minimum at ~ 3 to 2 ka is in agreement with the maximum of perennial or semi-permanent land fast sea ice at ~ 2.5 to 2 ka inferred from the minimal occurrence of dated driftwood (Dyke et al., 1997; Farnsworth et al., 2020; Funder et al., 2011). Furthermore, their curve is better supported by geomorphological evidence of glacier dynamics (Farnsworth et al., 2020)."*

3. **None of the tables has captions. It should be possible for a reader to understand tables and figures based on the caption.**

*All tables now have captions. See changes in our reply to the comment on tables.*

4. **Adding a description of the model simulations (number of simulations, parameterizations etc). in the methods would be helpful for the reader to anticipate the results.**

*New paragraph added at the end of section 4.3.4:*

New version:

*"To represent the uncertainty of how permafrost aggradation affects sub-permafrost groundwater flow, we simulated nine different scenarios that were defined by having three sets of values for the two fundamental parameters in any combination; hydraulic conductivity (Scenarios Sc1–3x, Table 3) and equivalent recharge (Scenarios ScXa–b, values calculated as described above and by Eq. 6). The nine scenarios are all labelled ScXx where X and x indicates the minimum, intermediate or maximum value sets of hydraulic conductivity and equivalent recharge, respectively. We further simulated a tenth scenario that takes additional pressure sources into account."*

5. **The simulations with the groundwater model need to be better explained. Clarify what scenarios were run and why. What is the significance of the 3 kyr catchment. How were they drawn in the first place? Figure 7 is very complicated and should be simplified and ideally split up into several figures. In the text, it is stated that each of the 12 zones were run with a different REq. However, the y-axis suggests that only three values were used. Explain why head is visualized with two different types of units?**

Referee#2 is not sure which equivalent recharge rates (REq) were used for which scenario of the groundwater model. We read this confusion as due to a misunderstanding that the subzones of the groundwater model were run individually ("it is stated that each of the 12 zones were run with a different REq"). This was not the case (without specific reference to the text, we can also not find this statement in our manuscript). Instead, the entire surface of the model domain was assigned with the maximum, intermediate or minimum estimates of the REq rate as calculated from heat transfer model results (Fig. 6c). As illustrated on Fig. 6c, the maximum, intermediate or minimum REq value was not uniform, but decreased in the inland direction, and each subzone was thus assigned with the relevant REq rate. As indicated by the y-axis on Fig. 7 and as stated in the text (Old version, L381-382) this resulted in three different total inflow rates of water to the groundwater model.

We have clarified and corrected the manuscript according to the suggestions in this comment (see changes in replies to minor comments) except for splitting up figure 7. Figure 7 was simplified as a consequence of changes arriving from comments by referee#1.

**Minor comments:**

**L13: "methane emissions/release"**

*Is the suggestion here that " methane" is followed by "emissions" or "release"? This would not be grammatically correct, but we could rephrase. No changes made to the revised manuscript.*

**L40: Clarify what "they" refers to, e.g. springs**

*"they" refer to springs. Replaced.*

*"they" replaced with "springs".*

**L47-L50. Clarify where the liquid water is coming from if not from the surface (e.g. relic groundwater?)**

Old version, L50:

*"This would remove the need to invoke groundwater recharge from the surface."*

New version:

*"This would remove the need to invoke groundwater recharge from the surface as spring outflow derives from relict groundwater."*

**L70: Here and other places. Be careful with using words like "this" without being more specific about what "this" refers too (here pressure). It is easy for the reader to get lost.**

*"this" replaced with "pressures"*

**L72: Define "talik"**

*Definition added after first use of talik:*

Old version, L72

*"[…] closed talik (Mackay, 1998)"*

*New version:*

*"[…] closed talik (i.e. a perennially unfrozen part of the permafrost) (Mackay, 1998)"*

**L73: "suggest that and open: : :"**

*done*

**L65-72: Somewhere here it would be good to explain the difference between open and closed pingo systems.**

*Sentence added at the beginning of the section previously starting "Liestøl (1977) suggests…" (Old version, L73):*

New version:

*"In contrast to closed-system pingos, an open-system pingo is sourced from a body of groundwater that is not enclosed by frozen ground."*

**L75: be more specific about what "the system" refers to.**

Old version, L75:

*"[…] the system […]"*

New version:

*"[…] spring outflow […]"*

**L82: Avoid abbreviations as much as possible. If you need the, make sure to write them out the first time they are mentioned.**

In the old version, the abbreviation was written out, but not at the first occurrence.

Old version, L82:

*"MAAT"*

New version:

*"mean annual air temperature (MAAT)".*

Old version L152:

*"mean annual air temperatures (MAAT)"*

New version:

*"MAATs".*

**L89: Explain what a "through-talik" is and how is it different than a "talik".**

*Change added to previous sentence. See below.*

Old version L82-86:

*"Close to the sea, groundwater flows towards the shoreface, but at some distance inland, higher advective heat transfer, associated with higher groundwater velocities, prevents frozen ground formation and groundwater flows through an intrapermafrost talik (through-talik) towards the surface along the most hydraulically conductive path, and a spring (or pingo) forms (as modelled by Scheidegger et al., 2012)."*

Replaced with:

*"Close to the sea, groundwater flows towards the shoreface, but at some distance inland, higher advective heat transfer, associated with higher groundwater velocities, prevents frozen ground formation. As a consequence, groundwater may flow through a talik that perforates the permafrost (i.e. a through-talik) towards the surface along the most hydraulically conductive path, resulting in a spring (or pingo) (as modelled by Scheidegger et al., 2012)."*

**L112: Reformulate. It is not clear from Fig 2b that the sediments are fine-grained or pre-Cenozoic.**

*Simple geological timescale added to the legend on Fig. 2b.*

Old version L113:

*"As illustrated on Fig. 2b, fine-grained Quaternary sediments (< 70 m thick Gilbert et al., 2018) overly pre-Cenozoic, well-consolidated sedimentary strata in Adventdalen, […]"*

New version:

*"In Adventdalen, fine-grained Quaternary sediments (< 70 m thick Gilbert et al., 2018) overly pre-Cenozoic, well-consolidated sedimentary strata (Fig. 2b), […]"*

**L116: What does "these" refer to. All layers or just the surface layers < 70 m? Needs clarification.**

*"these" refers to all of the aforementioned stratigraphy. Clarified:*

Old version, L116-117:

*"Together, these form a low-permeability groundwater system. "*

New version:

*"Together, all these units form a low-permeability groundwater system. "*

**L131: Explain the term "OSL"**

*First occurrence of OSL is written out:*

Old version, L131:

*"OSL"*

New version

*"optically-stimulated luminescence (OSL)"*

**L154: Specify period**

Old version L153:

*"The mean summer air temperature (MSAT) has consistently been […]"*

New version:

*"For the entire temperature record, the mean summer air temperature (MSAT) has consistently been […]"*

**L191: Clarify the source of these depth observations and the permafrost depths inferred from those observations.**

*We now mention the observations sites in the text and the associated references. However, we do not think that writing out the depths helps the reader without having the map (Fig. 2a) to reference the locations. As such, the depths still only appear from Fig. 2a.*

Old version, L190-191:

*"Specific depth observations support this regional characterisation (Fig. 2a)."*

New version:

*"One observation of the freezing front depth at Lagoon Pingo (Harada and Yoshikawa, 1996) and permafrost depth observations at well DH4 (Braathen et al., 2012), Endalen, Sarkofagen (both Liestøl, 1977), and Breinosa (Christiansen et al., 2005) support this regional characterisation (Fig. 2a)."*

**L220: Expand abbrevations, i.e. explain what GMS 10.4 is**

*In the first submission, we erroneously had the explanation for GMS (as below) at the second occurrence in the text. We do not expand the abbreviation (Groundwater Modelling System) as we think that "groundwater modelling software" is more eloquent.*

Old version, L220:

*"GMS"*

New version:

*"the groundwater modelling software GMS"*

Old version, L280:

*"the groundwater modelling software GMS"*

New version:

*"GMS"*

**L252: Rephrase "validate the model we" (you are validating the model, not the code itself)**

Old version, L252:

*"To validate the model code we […]"*

New version:

*"To validate the model we […]"*

**L253: Briefly explain the limitations of the analytical solutions (i.e. answering the question why can't you use these model for your study). Also summarize the findings about the model performance. Make sure to provide quantitative estimates of model performance (e.g. RMSE as in Supp. L79). Statements such as "relatively good performance" (Supp. L58) are not sufficient.**

*The problems dealt with by the analytical solutions are described in detail in the supplement and the reason why they cannot be used in our study is apparent there. We think it would be too lengthy to put it in the main article as the reasoning demands a description of the analytical solutions.*

*We now summarize the model performance by having added a sentence:*

Sentence added after L254 (Old version):

*"The model code was able to reproduce the analytical results with root-mean-square errors of respectively $1.1 \cdot 10^{-2}$ and $1.3 \cdot 10^{-5}$ and these numbers were regarded to represent reasonable accuracy."*

**L271: Explain the term "A" in equation 6.**

Old version, L271:

*"where $Q_{REq}$ is equivalent to the source term $Q_N$ in Eq. (5)."*

New version:

*"where $Q_{REq}$ is equivalent to the source term $Q_N$ in Eq. (5), and A is an area $[m^2]$."*

**L281: Better define subzones. The text says that 12 subzones are defined. However, figure 12 does not show these clearly. Table 1 says nothing about subzones.**

*(There is no Fig. 12 in our manuscript, but we assume that it is Fig. 5 (Fig. 4 in first submission), which is referred to here.)*

*In the old version of our manuscript, we used both of the terms "subzone" and "zone" to describe the same thing. All occurrences of "subzone" are now replaced with "zone". The definition of the zones is now*

*expressed more explicitly in the text as well as in the caption of Fig. 5. Figure 5 has also been subject to changes. See these in the list of changes to figures.*

Old version, L281-282:

*"For the 1DHT model, the geology was simplified into one-dimensional columns for a total of 12 subzones defined based age of the valley floor (Table 1).*

New version:

*"For the 1DHT model, the geology was simplified into one-dimensional columns for a total of 12 zones of the model area. The zones were defined as follows: The age of the valley floor (Table 1) was used to infer isochrones of valley floor exposure with intervals of 1000 yrs (Fig. 5a). The isochrones defined boundaries between zones and their names (i.e. the zone located between the 5 ka and 6 ka isochrones was named zone 5-6, Fig. 5b). The area between isochrones 9 ka and 10 ka was divided into two zones to incorporate geological variation. "*

**L293: Explain how you found out that porosity was the most important parameter**

Old version, L293-294:

*"Porosity was found to be the most important parameter for permafrost growth, and realistic minimum, mean, and maximum values were therefore defined for the 1DHT model (Table 2)."*

New version:

*"The considerable contrast between the thermal properties of water and ice implied that porosity was the most important parameter for permafrost growth, and realistic minimum, intermediate, and maximum values were therefore defined for the 1DHT model (Table 2)."*

**L311: Can a grid be one dimensional? Isn't it by definition 2D? I suggest rephrasing.**

*In our understanding, a grid can be 1D, 2D or 3D. By 1D grid we mean a 3D grid with only one grid line in two out of three dimensions, and where both these grid lines equal unity. However, we agree that rephrasing will add clarity. By rephrasing, we avoid confusion between the domains of the heat transfer model and groundwater model (columns and grid, respectively) We have employed rephrasing as suggested and replaced '1d grid' with 'column':*

Old version, L311:

*"The model domain contained 12 one-dimensional grids […]"*

New version:

*"The model domain contained 12 columns […]"*

**L312: It is not clear from Figure 4 where these 12 grids/columns are located. I suggest adding a point or arrow to identify the locations of the 12 grid/columns**

*See changes to Fig. 5 in the list of changes to figures.*

Old version, L312:

*"Each individual grid was associated to the model area zones (Fig. 4) and the geothermal properties were defined accordingly."*

New version:

*"One column was associated to each of the model area zones and the geothermal properties were defined according to the associated geological 1D simplifications (see insert on Fig. 5b)."*

**L313. Table 1 shows site locations, not geological units. Do you mean table 3? Either way, explain how the names of the zones refer to the age. It is unclear. Provide a better connection to Figure 4. I suggest making a table with parameters for each of the 12 zones.**

*Please, also see the changes arising from the comment to L281 (old version).*

Old version, L313:

*"The names of the zones refer to the age of subaerial exposure (Table 1), which defined the simulation run time (e.g. for zone 0-1 the simulation period was 0.5 to 0 ka, for zone 1-2 it was 1.5 to 0 ka, etc. For zone 10, the simulation period was 10 to 0 ka)."*

New version:

*"The simulation run time was defined by the valley floor age inferred for that zone (Fig. 5), so that, for zone 0-1 the simulation period was 0.5 to 0 ka, for zone 1-2 it was 1.5 to 0 ka, etc. For zone 10, the simulation period was 10 to 0 ka."*

**L319: Clarify that this grid is different from the one in the previous section.**

*This should be obvious now, as we no longer call the 1DHT model domain 'grid'. Clarified further by the following rephrasing:*

Old verison, L319-320:

*"Each grid cell measured 100 by 100 by 5 m (x y z) and their hydrogeological properties were defined from the geology (Fig. 4, Table 3)."*

New version:

*"For the groundwater model, each grid cell measured 100 by 100 by 5 m (x y z) and their hydrogeological properties were defined from the geology (Fig. 5a, Table 3)."*

**L331: Clarify how you dealt with the active layer at the surface (i.e. unfrozen too, but also inactivated?)**

*Annual temperature variations was not included in the model setup and the seasonal thaw of the active layer was therefore not considered. No changes made to the manuscript based on this comment.*

**L331: Clarify the meaning of "raw" and how it differs from non-raw simulations**

*Rephrased to clarify:*

Old version, L331-332:

*"The raw simulation results from the 1DHT model were […]"*

New version:

*"The direct output from running the 1DHT model code was […]"*

**L333: The interpolation is unclear. What was interpolated, time or space? Explain.**

Old version, L332-333:

*"The evolution of the permafrost and freezing front depths were evaluated by interpolating the associated temperatures for each time step."*

New version:

*"For each time step, the permafrost and freezing front depths were calculated by interpolating the depth at which the associated temperatures occurred."*

**L335: Be more specific and clarify. Isn't it the permafrost aggradation in m/s that is inferred from the freezing front?**

*Strictly speaking, permafrost aggradation is defined by the progression of the 0°C isotherm. What we attempt to express here is that the progression of the -0.7C isotherm is more relevant for the contribution of hydraulic pressures on the GW system, because this is the temperature where the phase change rate if fastest (in our model, that is). We have rephrased and added a sentence to clarify:*

Old version, L333-335:

*"The greatest phase change rate in the 1DHT model occurs at a temperature of -0.7 °C (following from Eq. 3) and the (always) downwards progression rate of this isotherm was therefore used for the calculation of the equivalent recharge (by net pore space loss, Eq. 6)."*

New version:

*"The greatest phase change rate in the 1DHT model occurs at a temperature of -0.7 °C (following from Eq. 3) and it was therefore the progression rate of this isotherm that was used for the calculation of the*

*equivalent recharge (by net pore space loss, Eq. 6). Hereafter, 'permafrost aggradation' therefore means the downwards progression rate of this isotherm (-0.7 °C), although this is not entirely congruent with the thermal definition of permafrost (ground perennially below 0 °C, French, 2017)."*

**L340: Table 2 does not mentioned "intermediate" values, only, min/mean/max for three materials. Clarify what porosity values was used, and why those were used over other values.**

*Old version, L294: "mean" replaced with "intermediate"*

*Table 2: "Mean" replaced with "Intermediate"*

**L371: Cross reference with tables so that the reader can check this statement. Also provide the name of the layer.**

*Column added to Table 2 with values of thermal diffusivity.*

Old version, L370-374:

*"This was due to the different properties of the sediments and bedrock undergoing freezing (Fig. 4). In zone 0-1, closest to the shore, phase change took place at < 60 m b.g.l. corresponding to the most porous and least thermally diffusive unit (the fluvio-deltaic succession). Thus, a relatively high amount of latent heat had to be released for the freezing front to aggrade. By contrast, the opposite was the case in zone 1-2, where the freezing front just entered the sandstone unit that possessed the lowest porosity."*

New version

*"This was due to the different properties of the sediments and bedrock undergoing freezing. In zone 0-1, closest to the shore, phase change took place at < 60 m b.g.l. corresponding to the most porous and least thermally diffusive unit (Qt1, insert on Fig. 5 and Table 2). Thus, a relatively high amount of latent heat had to be released for the freezing front to aggrade. By contrast, the opposite was the case in zone 1-2, where the freezing front just entered the sandstone unit that possessed the lowest porosity (Carolinefjellet Fm, insert on Fig. 5 and Table 2)."*

**L374: What about the thermal properties. Also cross-reference with table.**

*See change in answer to comment above.*

**L389: Explain how you know this was artesian.**

*Reference to Fig. 7. Added:*

Old version, L389:

*"Entirely or almost entirely [...]"*

New version:

*"As illustrated by the colour fill on Fig. 7, entirely or almost entirely [...]"*

**L390: Explain how you know that hydraulic pressures were below hydrostatic as well as the significance of this.**

*Wrong usage of hydrostatic.*

Old version, L390:

*"[…] where the up-valley part of the system has hydraulic pressures below hydrostatic."*

New version:

*[…] where the up-valley part of the system has hydraulic pressures below ground level.*

**L392: Explain the 3 kyr catchments. Why 3 and not 4 kyr? What is the significance of these "catchments"**

*The 3 kyr catctments were drawn to visualize a slow moving groundwater system. On Fig. 7, the mean pore water velocities are also provided and indicate the same, but not in a visual manner. The catchment size (time) could also have been 4 kyr, but not 50 yr or 15 kyr. 3 kyr years was chosen as this is the time order on which the modelled conditions have existed.*

Old version, L392-393:

*"This enabled us to draw 3 kyr catchment zones for each outlet point (the term 'catchment' is somewhat misleading in this context as no actual recharge takes place)."*

New version:

*"In order to visualise groundwater movement towards the outlet points, we used the particle tracking to draw 3 kyr catchment zones (the term 'catchment' is somewhat misleading in this context as no actual recharge takes place). The 3 kyr duration was chosen because it is on that order of time that the modelled permafrost and groundwater conditions likely existed."*

**L401: Rephrase and clarify. It reads as if both artesian and non-artesian determine spring sites and has small QREq and high K. Earlier it is written that most of the area was artesian.**

Old version, L401-402:

*"The artesian/non-artesian conditions clearly determine whether outflow takes place at all pingo spring sites, as was indeed the case for all but three simulations (small $Q_{REq}$ and high K, Figs. 7-2a and 7-3a–b)."*

New version:

*"The colour fill and the pie charts on Fig. 8 together show that outflow at a pingo site only takes place if the hydraulic pressure is artesian. Outflow from all pingo sites were simulated for four scenarios including all three with maximum $Q_{REq}$ and the one with intermediate $Q_{REq}$ and minimum K (Figs. 8-1b–c, 8-2c and 8-3c)."*

**L445: Clarify if the characteristic length is equal to l, and why 200 m was selected.**

*In the revised manuscript, the concept of adjustment time is introduced earlier in a new paragraph added to section 4.3.4 and we here clarify that l denotes the characteristic length (see the paragraph in our reply to referee#1's comment to line 515, old version). As a consequence, section 6.1.1 is now completely rewritten:*

New version, Sect. 6.1.1:

*"A possible interpretation of anomalous overpressure is that a previous perturbation was long-lived enough to redistribute groundwater (and other fluids) and recent enough for groundwater not to have adjusted to the present conditions (Bahr et al., 1994). The notion of adjustment time (Sect. 4.3.4, Eq. 8) becomes convenient when assessing whether an ice-load removed > $10^4$ yr ago could be responsible for present anomalous pressures. For shallow, low-permeable and well-consolidated bedrock systems like the one investigated in this work, we found the vertical $t_a$ to be between 80 and 7500. To calculate this, we used the lowest K-estimate of the dominant hydrogeological unit (Janusfjellet Subgroup, Table 3) and a characteristic length of 200 (approximating half of the thickness of the aforementioned unit). The specific storage was defined like when calculating the horizontal adjustment time (Sect. 4.3.4). Conclusively, we argue that overpressures in systems like the investigated case cannot be explained by equilibration from past glacial loading."*

**L453: Clarify. Pertubations of what**

Old version, L453:

*"Past geological or climatic events may be indirectly responsible for ongoing perturbations"*

New version:

*"Past geological or climatic events may be indirectly responsible for ongoing pressure perturbations"*

**L536: Be specific about which scenarios you are referring to here.**

*Rephrased and specific references to scenarios added.*

Old version, L534-536:

*"Further, discharge from all of the observed pingo springs was not simulated for scenarios with high K's and low equivalent recharge. This could indicate that the real K-values are in effect lower than those employed for these scenarios."*

New version:

*"Further, discharge from the up-valley pingo springs was not simulated for scenarios with minimum $Q_{REq}$ and maximum K (ScXa and Sc3b, Figs. 8-1a, 8-2a, 8-3a–b)."*

**L547: Explain why max K values where used. In the previous text you argued for intermediate K values to be the most reasonable.**

*The high K values were used in order to allow the highest amount of recharge to enter the system. We add this statement in the revised manuscript.*

*The argument for intermediate K values was an error in the first version of the manuscript and it has been removed from the revised manuscript.*

Old version, L549:

*"[…] alone (Fig. 7). Figure 9 […]"*

New verison:

*"[…] alone (Fig. 8). We used the maximum K-values to allow for the highest amount of recharge to enter the system. Figure 10 […]"*

**Comments on figures and tables ===================**

**Tables: Add captions**

*Changes to the tables are explained below*

***Table 1:***

*First row removed from table (i.e. number and title)*

Added caption:

*"**Table 1** Absolute ages from Adventdalen constraining delta propagation. Depending on the dated material and the host sediment, the dating indicates minimum, approximate or maximum valley floor ages. See Fig. 2 for site locations. HML = Holocene marine limit. Compiled from [I]Åhman (1973), [II]Svensson (1970), [III]Gilbert et al. (2018), [IV]Yoshikawa and Nakamura (1996) and [V]Lønne and Nemec (2004)."*

*Sentence removed in revised manuscript:*

Old version, L185-186:

*"Depending on the dated material and the host sediment, the dating indicates minimum, approximate or maximum ages of subaerial exposure (i.e. exposure to MAATs)."*

***Table 2***

*First row removed from table (i.e. number and title)*

*Column added showing thermal diffusivities*

*Reference indicated for all values*

Caption added:

*"**Table 2** Geothermal material properties used in the heat transfer model. Density and thermal properties compiled from [I]Williams and Smith (1989), [II]Robertson (1988), and [III]Manger (1963). Porosities from [IV]Fitts (2002), [V]Braathen et al. (2012), and based on works by [VI]Burland (1990), Grundvåg et al. (2019), Marshall et al. (2015), Okiongbo (2011), and Yang and Aplin (2004) (see text)."*

**Table 3:**

*"Table 3" removed from table*

*References for values indicated for all hydrogeological units*

Caption added:

*"**Table 3** Properties of the hydrogeological units used in the groundwater model. Values based on [I]Fitts (2002) and [II]Singhal and Gupta (2010) or evaluated from [III]Braathen et al. (2012) (see Supplement)."*

**Table 2: Be more specific about what source you sued for what properties by using. The way it is done for porosity is great, do this for all parameters and other tables as well.**

*See changes in reply to the previous comment.*

**Figure S1 should be in the main article. However, I suggest creating different types of boxes in the flow chart to distinguish between data outputs and inputs, calculations/algorithms, and decisions. Make sure the groundwater model MODFLOW in the diagram**

*The figure is now removed from the supplement and placed in the main article as a Fig. 4. Figure changed according to suggestions.*

Old version, Supplement, caption to Figure S1:

*"Figure S1 Schematic overview of the inner workings of the decoupled heat and groundwater model."*

New version, Caption fig. 4:

*"**Figure 4** Schematic overview of the inner workings of the decoupled heat and groundwater model. Model setup, calculations and algorithms are indicated with sharp corners and bold text. Validation and comparison of simulations and observations are indicated with round corners and italics. Input and output data are indicated with rounded corners, grey background and normal text."*

**Figure 1: This is a great figure to illustrate the conceptual model. To improve., consider using another shading to identify permafrost**

*Shading added to indicate frozen ground.*

**Figure 2 caption. Several words, terms, sites mentioned within parenthesis are unclear without context. Clarify the meaning/context of these, including: "temperature loggings", "Sarkofagen",**

**"Breinosa", "geophysics". Be specific about what "Map data" you refer to, e.g. topographic data in panel a. Add at the end "description of the layers shown in cross sections A, B, and C". Add compass to the map since directions are discussed in the text later. Also point out the direction towards Longyearbyen**

- *Compass added. The location of Longyearbyen is now indicated. Site name "Longyeardalen" removed to make room for this. The location of Longyearbyen airport is now shown.*
- *"(temperature logging)" and "(geophysics)" was written to indicate the methods that were used to measure the depths. Now removed to avoid confusion. Site names in parenthesis (i.e. Sarkofagen, DH4 and Breinosa) all appear from the figure. Sentence rephrased to clarify that these are site names.*
- *"Map data" specified.*
- *Sentence added at the end of the caption as suggested.*

Old version, caption to Figure 2, L104-111:

*"**Figure 2 (a)** Map of Lower Adventdalen with the location of data resources, pingos and the Holocene marine limit. LP=Lagoon Pingo, LYRP=Longyear Pingo, FHP=Førstehytte Pingo, IHP=Innerhytte Pingo, RP=River Pingo. Core logs from boreholes S1–3 and D1–D7 (respectively, Gilbert et al., 2018, and Olaussen et al., 2020, and references therein), seismic lines (Bælum et al., 2012, and unpublished commercial lines from Norsk Hydro) and a geological map (Norwegian Polar Institute, 2019) were used to build the geological model (Fig. 4) (see details in Hornum, 2018). Permafrost depths (temperature logging) are from Liestøl (1977) (Sarkofagen), Braathen et al. (2012) (DH4), and (Christiansen et al., 2005) (Breinosa). The freezing front depth (geophysics) is from Yoshikawa and Harada (1995). Map data by courtesy of Norwegian Polar Institute (2019). **(b)** Geological cross sections constructed based on the resources mentioned above. See Sect. 3.1 for a (hydro)geological description."*

New version:

*"**Figure 2 (a)** Map of Lower Adventdalen with the location of data resources, pingos and the Holocene marine limit. LP=Lagoon Pingo, LYRP=Longyear Pingo, FHP=Førstehytte Pingo, IHP=Innerhytte Pingo, RP=River Pingo. Core logs from boreholes S1–3 and D1–D7 (respectively, Gilbert et al., 2018, and Olaussen et al., 2020, and references therein), seismic lines (Bælum et al., 2012, and unpublished commercial lines from Norsk Hydro) and a geological map (Norwegian Polar Institute, 2019) were used to build the geological model (Fig. 5a) (see details in Hornum, 2018). Permafrost depth measurements at the Sarkofagen, DH4 and Breinosa sites are from Liestøl (1977), Braathen et al. (2012), and Christiansen et al. (2005), respectively. The freezing front depth at LP is from Yoshikawa and Harada (1995). Data used to develop the map including topography, glacial extent, and fluvial network by courtesy of Norwegian Polar Institute (2019). **(b)** Geological cross sections constructed based on the resources mentioned above. The Quaternary unit overly well-consolidated sedimentary strata of pre-Cenozoic age (i.e. Cretaceous or older). See Sect. 3.1 for a (hydro)geological description of the layers shown in the cross sections A, B and C."*

**Figure 3. Explain the time axis. Time in relation to what (i.e. what is time 0 = present day). Use colors instead of dashed lines. The grey shaded are of driftwood arrival is almost impossible to see, use colors for this too.**

- *The time axis is in 'ka' and with this unit we means 1000 years before present. This was already stated in the first version of the manuscript (at the first occurrence, L129). To further clarify, we now also explain the unit in the caption.*
- *We now use colors instead of the dashed lines.*
- *We have kept the grey dashed area, but darkened it to make it more visible.*

Old version, caption for Figure 3, L170-172:

*"Figure 3 Holocene temperature reconstructions. Dashed light grey area indicate time of minimal driftwood arrival 170 (Farnsworth, 2019). (a) MSST curves (= MSAT, see text). Solid line from Mangerud and Svendsen (2017). Dotted line from Hald et al. (2007). Dashed line from van der Bilt et al. (2018). (b) MAAT used in this work. Based on Mangerud and Svendsen (2017) and Førland et al. (1997)."*

New version:

*"Figure 3 Holocene temperature reconstructions in and around Svalbard. Dashed grey area indicate time of minimal driftwood arrival (Farnsworth et al., 2020). The unit of the time axis is ka = $10^3$ years before present. (a) MSST curves (= MSAT, see text). Red line from Mangerud and Svendsen (2017). Orange line from Hald et al. (2007). Blue line from van der Bilt et al. (2018). (b) MAAT used in this work. Based on Mangerud and Svendsen (2017) and Førland et al. (1997) (see Sect. 4.3.3)."*

**Figure 4: Clarify that the inset maps shows the 1D simplifications of the model area. Choose one word to describe the 1D simplification (simplifications or interpretation). Explain the numbering of the subzones. Why does it start with zero and why is there a 10b. Why not go from 1 to 12? Explain everything shown in the figure, e.g. the red arrows pointing out pingo locations.**

- *Figure 4 is now fig 5*
- *We have clarified the meaning of the inset by making the labelling of the zones (now only called 'zones') more visible.*
- *1D simplifications are now only called so.*
- *The labelling of the zones is now explained in the caption. Isochrones added to the figures.*
- *Legend added to make the figure easier to read.*
- *A zone map is now provided in a new panel b.*

Old caption:

*"Figure 4 3D geological model of subsurface below the valley floor in Adventdalen and vertical 1D simplifications below subzones of the model area. The former determines the hydrogeological properties in the groundwater model (Table 3), whereas the latter determines the geothermal properties in the 1DHT model (Table 2). The subzones were defined based on the reconstruction of fjord retreat and their names indicate the exposure ages (i.e. zone 0-1 became sub-aerially exposed between 1 and 0 ka, zone 1-2 between 2 and 1 ka, etc.)."*

New caption:

*"**Figure 5 (a)** 3D geological model of the subsurface below the valley floor in Adventdalen and vertical 1D simplifications (inset in upper right corner) below zones of the model area. The former determines the hydrogeological properties in the groundwater model (Table 3), whereas the latter determines the geothermal properties in the 1DHT model (Table 2). The domain of the 1DHT model extends to 1000 m b.g.l. (not shown). Deeper than 300 m b.g.l., geothermal properties were defined as for Janusfjellet Subgroup. The sea retreat reconstruction was inferred from absolute datings (Table 1) and is illustrated by valley floor exposure isochrones (red dashed lines). Pingos are indicated with blue arrows. **(b)** Vertical view of the model area showing the zonation. The aforementioned isochrones defined the zone names so that the valley floor exposure age of a zone is apparent from its name (i.e. zone 0-1 became sub-aerially exposed between 1 and 0 ka, zone 1-2 between 2 and 1 ka, etc.). "*

**Figure 5: This figures shows better what the zones are. Something like this is needed in figure 4 but indicating all zones. Why not just label the zones 1, 2, 3 and so on. But you also need to explain in the caption what all elements in the figure are (including the zone map)**

Old caption:

*"**Figure 5** Development of simulated freezing front and permafrost front depths from zones 2-3, 7-8 and 9-10b (Fig. 4). These simulation results derive when using the intermediate porosity values (Table 2). Note that completely frozen ground does not establish permanently until ~ 6.5 ka."*

New caption:

*"**Figure 5** Development of simulated freezing front and permafrost front depths from zones 2-3, 7-8 and 9-10b of the model area (see Fig. 4). These simulation results derive when using the intermediate porosity values (Table 2). Note that completely frozen ground does not establish permanently until ~ 6.5 ka."*

**Figure 6: Clarify that the uncertainty fields are uncertainty due to porosity. Why are symbols include in the middle chart, but not in the top and bottom chart? Make the chart constant.**

*Now figure 7.*

*Figure changed due to new model setup.*

**Figure 7: The nine scenarios need to be explained in the method section. Explain how were they selected and parameterized. The figure is difficult to understand. Simplify and split up into multiple figures and tables. What does the colors mean for the DH4 borehole?**

*Now figure 8.*

*Scenarios explained in the methods sections.*

*The figure was simplified as a result of changes suggested by Referee #1. In order to ease comparison and keep an overview of the modelling results from all scenarios, we have not split up the figure. However, if the reviewer still thinks that this would add clarity, we will be happy to do so.*

**References in replies to comments**

[revised manuscript text omitted]

by / Condensed by

---

## Author Response (AR2)

**Author response to Reviewer's and Editor's comments regarding resubmission of TC-2020-07**

Thanks for another round of constructive reviews. The comments and suggestions were highly appropriate and we have changed the manuscript accordingly. Details of changes are provided in the point-to-point reply and in the marked-up versions of the manuscript and the supplement, all three found on the following pages and in that order. In short, we did the following:

- Changed the last part of the abstract to emphasize that our study shows that permafrost aggradation deserves more attention as driver for the transport of deep groundwater and methane to the surface.
- Replaced pie charts with bar charts in figures 8 and 10
- Added two new statistical fits to the measured K-values of Carolinefjellet and Helvetiafjellet Fms (in the supplement).
- Rephrased text and added/removed sentences and words as suggested by the referees.

With great thanks,

Mikkel Toft Hornum on behalf of all authors

**Point-to-point reply to referee comments**

**Referee comments are written in red.** Author replies in black.

**Referee #1 (Melissa Bunn)**

The following minor revisions are offered for the consideration of the authors:

- Line 53-54: Heat transport and groundwater flow
  - corrected
- Table 1 – Does the Asterisk in Site/*Event have a purpose, is there a note missing?
  - The asterisk was leftover from a previous manuscript version. Now deleted.
- Line 258: In discussing the RMS error for the heat flow model validation, opposed to reasonable accuracy, could it be stated that it is an acceptable level of accuracy for the given purpose.
  - *Old version, line 258:* "[…] to represent reasonable accuracy."
  - *New version:* "[…] to represent an acceptable level of accuracy for the given purpose."
- Line 333: The properties were that of THE Janusfjellet Group.
  - corrected
- Line 353: would tidal flat be more appropriate relative to Fjord, given the extent of the model boundary?
  - We think 'fjord' is more appropriate, because the general head boundary conditions was assigned to cells located where the fjord begins (and the model domain ends). We have added a sentence to clarify:
  - *Old version:* "The fjord was simulated with a general head of 0 m a.s.l. assigned to the relevant cells and the conductance was determined according to hydraulic conductivity (Table 3)."
  - *New version:* "The fjord was simulated with a general head of 0 m a.s.l. and the conductance was determined according to the hydraulic conductivity (Table 3). The area assigned with this boundary conditions comprised 24 cells located within the uppermost layer at the north-western end of the model domain."
- Line 360: were assigned no-flow conditions
  - corrected
- Line 447 to 448: Partial sentence left over?
  - Yes. Partial sentence deleted.
- Line 449: Two exceptions to this result were the minimum…
  - corrected
- Line 468: and the lack of discharge at the up-valley….
  - corrected
- Line 479: all of this outflow discharge to the fjord.
  - corrected
- Line 479: Instead of referring to the water flux that is redistributed to subpermafrost liquid water as "outflow to permafrost thaw" perhaps it would be clearer to explain this as a phase change and redistribution to storage opposed to an outflow, as it does not leave the model domain.
  - We agree with the referee and have rephrased here and in the caption for Figure 8:

- *Old version, Lines 478-480:* "For intermediate and maximum $Q_{REq}$ scenarios (Sc*X*b–c, Fig. 8), all of this discharged to the fjord, while, for the minimum $Q_{REq}$ scenarios (Sc*X*a, Fig. 8), a minor proportion of the outflow represents phase change associated with permafrost thaw, which causes a storage redistribution within the model domain."
  - *New version:* "For intermediate and maximum $Q_{REq}$ scenarios (Sc*X*b–c, Fig. 8), all of this discharged to the fjord, while, for the minimum $Q_{REq}$ scenarios (Sc*X*a, Fig. 8), a minor proportion of the outflow represents phase change associated with permafrost thaw, which causes a storage redistribution within the model domain."
  - *Old version, Lines 489-490:* "*For minimun $Q_{REq}$ scenarios (**1a**, **2a** and **3a**) part of the outflow is caused by basal permafrost thaw (Figs. 7b–c)."
  - *New version:* "*For minimum $Q_{REq}$ scenarios (**1a**, **2a** and **3a**) part of the outflow represents phase change associated with permafrost thaw (Figs. 7b–c), which causes a storage redistribution within the model domain."
- Line 480: Larger proportions of outflow to the fjord were simulated…..
  - corrected
- Line 520: low-permeability
  - corrected
- Line 590: Negelecting this effect represents an uncertainty in the simulation results,….
  - corrected
- Line 600: These conditions should be taken into account….
  - corrected
- Supplement S2 – Figure S5: The higher measured hydraulic conductivity values do not seem to fit the Weibull distribution presented. Were other distributions presented, with this distribution yielding the best fit? If so, it may be useful to include a statement indicating such. Was a log-normal distribution (using the negative log of hydraulic conductivity) attempted?
  - *Supplement, old version, Lines 81-82:* "The ranges of hydraulic conductivity of these units were defined by the 25 %, 50 % and 75 % percentiles of a Weibull probability fit to the measured values (Fig. S5)."
  - *New version:* "To tighten the $K_h$-range used in the groundwater model, we defined the realistic minimum, mean, and maximum values as the first, second and third quartiles of a statistical fit to the measured values. As illustrated on Fig. S4, three types of statistical distribution were tested; log-normal, Weibull and log-logistic (panels a), b) and c), respectively). While the log-logistic fit showed the best correlation, all three fits predicted the same hydraulic conductivities at the quartiles."
  - *Supplement, old version, caption for Figure S4:* "Figure S5 Weibull fit to measured hydraulic conductivities of Carolinefjellet and Helvetiafjellet formations. Original data from the Longyearbyen CO$_2$ Laboratory Project (Olaussen et al., 2020, and references therein)."
  - *New version:* "**Figure S4** Log-normal (a), Weibull (b) and log-logistic (c) fits to the measured hydraulic conductivities of Carolinefjellet and Helvetiafjellet formations. Original data from the Longyearbyen CO$_2$ Laboratory Project (Olaussen et al., 2020, and references therein). The log-logistic fit shows the best correlation with the measured values and the values used in the groundwater model were thus defined by the quartiles predicted by this fit. Note that the two other fits both predict the same quartiles."

**Referee #2**
The authors have made many edits to address my points in the previous review. I have the following suggestions for the revised manuscript.

- L25-27: I suggest rewriting/replacing the last two sentences and try and make it more impactful. I think the interesting implication is how methane emissions are affected by the processes investigated in this paper, or I suppose what can be inferred about methane from the processes. The introduction also set up this implication. Also, give that this is an unusual hydraulic system, where would you expect to find other sites like it?
  - *Old version, Lines 20-28:* "Simulations also suggest that the generally low-permeability hydrogeological units cause groundwater residence times to exceed the duration of the Holocene. The likelihood of such pre-Holocene groundwater ages is also supported by the geochemistry of the pingo springs, which demonstrates an unexpected sea-ward freshening of groundwater, potentially caused by a paleo-subglacial melt water "wedge" from the Weichselian. This sub-permafrost wedge progressively thins inland, as permafrost thickness (and age) increases and so, less unfrozen freshwater is available for mixing. Although this unusual hydraulic system is most likely governed by permafrost aggradation, the potential for additional pressurisation is also explored. We find that methane production and methane clathrate formation/dissolution deserve particular attention on account of their likely effects upon the hydraulic pressure at the study site and others like it."
  - *New version:* "We also explore the potential for additional pressurisation, and find that methane production and methane clathrate formation/dissolution deserve particular attention on account of their likely effects upon the hydraulic pressure. Our model simulations also suggest that the generally low-permeability hydrogeological units cause groundwater residence times to exceed the duration of the Holocene. The likelihood of such pre-Holocene groundwater ages is supported by the geochemistry of the pingo springs, which demonstrates an unexpected sea-ward freshening of groundwater, potentially caused by a paleo-subglacial melt water "wedge" from the Weichselian. This sub-permafrost wedge progressively thins inland, as permafrost thickness (and age) increases and so, less unfrozen freshwater is available for mixing. Although this is an unusual hydraulic system our observations confirm that, also in other uplifted Arctic valleys, millennial-scale permafrost aggradation deserves attention as a possible driver of sustained flow of sub-permafrost groundwater and methane to the surface."
- L210: Maybe clarify "by the eye" a bit. Was any quantitative measurement involved, or is this just a guess? Perhaps add an uncertainty range to indicate a potential range, since this probably is an unprecise method of estimation.
  - This is indeed a guess based on personal experience, and we agree that this is an imprecise method. We have further emphasized the uncertainty by rephrasing:
  - *Old version:* "Based on our own "by the eye" observations, the discharge rate at Førstehytte Pingo was in orders of 0.1 L s$^{-1}$ when visited in fall of 2015, 2016 and 2017, and less than 0.01 L s$^{-1}$ when rediscovered in October 2019."
  - *New version:* "Based on our own "by the eye" estimates involving no quantitative measurements, the discharge rate at Førstehytte Pingo was in orders of 0.1 L s$^{-1}$ when visited in fall of 2015, 2016 and 2017, and less than 0.01 L s$^{-1}$ when rediscovered in October 2019."

- L431: When using terms such as "surprising" it should also be clarified why the results are "surprising".
  - *Old version, Lines 431-433:* "At first surprising, the model simulations generally did not suggest that the highest aggradation rate occurred where permafrost is youngest (zone 0-1), but instead at zone 1-2. This was due to the different properties of the sediments and bedrock undergoing freezing."
  - *New version:* "For a homogeneous medium with one-sided freezing, the freezing front progression rate will decrease with time (as exemplified by Figure S2). At first, it therefore seems surprising that the model simulations generally did not suggest that the highest aggradation rate occurred where permafrost is youngest (zone 0-1), but instead at zone 1-2. This was due to the heterogeneity in the model domain as expressed by the different properties of the sediments and bedrock undergoing freezing."
- L418: explain what labels were used to identify the pingo springs in panel 1c.
  - Change to figure 7: We have emphasized the symbols and labelling used to show the freezing front and permafrost depth observations. The references of these observations have been added to the figure caption.
- L473: I would rephrase: It is the model that shows that outflow only take place if hydraulic pressure is artesian, the colour fill and figure 8 is only visualizing the model result. So the figure text says that red is artesian flow, and blue is not. But several figures have both (1a, 2a, 3a). Clarify that the pressure needs to be artesian at the pingos. I guess this is the case. I think converting the pie charts into bar charts where the data is grouped based on simulation and the artesian conditions may be more effective, but perhaps this is an issue of taste so I leave it for the authors to consider.
  - Sentence rephrased to clarify:
  - *Old version, Lines 473-474:* "The colour fill and the pie charts on Fig. 8 together show that outflow at a pingo site only takes place if the hydraulic pressure is artesian."
  - *New version: "*The colour fill and the bar charts on Fig. 8 together show that the hydraulic pressure below a pingo site needs to be artesian in order for outflow to take place at that pingo."
  - Pie charts replaced with bar charts in Figures 8 and 10 text changed accordingly.
- L487: Clarify what distribution in the term "outflow distribution" refer to
  - *Old version, Lines 488-489:* "The outflow distribution and discharge rates are illustrated by pie charts with the location of the discharge points (pingos and fjord) indicated on **(1c)**."
  - *New version:* "The groundwater outflow rates and distribution are illustrated by bar charts with the location of the discharge points (pingos and fjord) indicated on **(1c)**."
- Figure 8: I still think there is too much information in one figure, but I guess the authors disagree. Pie charts are typically not recommended for scientific data, since it is difficult to discern magnitudes. Bar charts are typically more effective than pie charts. Of course, the authors have provided the actual data in the pie charts, but that adds clutter and information overload.
  - Pie charts replaced with bar charts in Figures 8 and 10 text changed accordingly.
- Figure 10. Add a line that the figure is using the same structure, legend etc. as Figure 8.
  - done

[revised manuscript text omitted]

**S2    Hydraulic conductivity of Carolinefjellet and Helvetiafjellet Formations**

The vertical permeability, $\kappa_v$, of the sandstone-dominated Carolinefjellet and Helvetiafjellet formations were measured as part of the Longyearbyen CO$_2$ Laboratory Project (Olaussen et al., 2020, and references therein). The small-scale horizontal

80    permeability, $\kappa_h$, for sandstones is typically a factor two higher than $\kappa_v$ (Domenico and Schwartz, 1998) and we converted the horizontal hydraulic conductivity, $K_h$, accordingly (Eq. 7). To tighten the $K_h$-range used in the groundwater model, we defined the realistic minimum, mean, and maximum values as the first, second and third quartiles of a statistical fit to the measured values. As illustrated on Fig. S4, three types of statistical distribution were tested; log-normal, Weibull and log-logistic (panels a), b) and c), respectively). While the log-logistic fit showed the best correlation, all three fits predicted the same hydraulic

85    conductivities at the quartiles.

[Figure]

**Figure**  S4  Log-normal (a), Weibull (b) and log-logistic (c) fits  to the measured hydraulic conductivities of Carolinefjellet and Helvetiafjellet formations. Original data from the Longyearbyen $CO_2$ Laboratory Project (Olaussen et al., 2020, and references therein). The log-logistic fit shows the best correlation with the measured values and the values used in the groundwater model were thus defined by the quartiles predicted by this fit. Note that the two other fits both predict the same quartiles.

**S3      Major ions in Adventdalen pingo spring waters**

Hydrochemical data from 25 spring water samples from 2014 to 2017 presented by Hodson et al. (2020) and publicly
95    available from DOI:10.5285/3d82fd3f-884b-47b6-b11c-6c96d66b950d give insights into the groundwater system in
Adventdalen. As illustrated in Fig. S6a, water samples from Lagoon (LP), Førstehytte (FHP), Innerhytte (IHP) and River
Pingos (RP) reveal that all these springs share the same sodium-bicarbonate ($NaHCO_3$) water type. The only exception is four
samples taken near River Pingo in 2017 of a magnesium-sulfate water type (Fig. S6b). These four samples were excluded from
the discussion because they might not be associated with a pingo according to Hodson et al. (2020).
100

[Figure]

**Figure S5** Stiff plots of major ion concentrations from pingo spring water samples in Adventdalen (data first presented in
Hodson et al., 2020). The corners of the Stiff polygons represent mean concentrations.

---

## Author Response (AR3)

**Author response to Editor's comments regarding resubmission of TC-2020-07**

Thanks for another round of constructive review. The suggestion of adding a paragraph to the discussion was highly appropriate and has helped to improve the manuscript. The changes we have made are summarized in the list below and a marked-up version of the manuscript is found on the following pages. In the new version of the manuscript we have:

- Moved detailed information about table contents from the table titles to notes below the tables.
- Rephrased the last two sentences in the manuscript.
- Added a new section (Sect. 6.5) to the discussion, where we discuss the pan-Arctic significance.
- Added a new paragraph at the end of the discussion, which links to the new section and the last part of the abstract.
- Changed the doi-reference to the 1DHT model code. This references the updated code, which was ch anged because of the first review of this manuscript.

With great thanks,

Mikkel Toft Hornum on behalf of all authors

Non-public comment to Editor:

Dear Peter,
Thanks a lot for all your work reviewing and editing this manuscript. Much appreciated! One little detail, which is not mentioned above: The University of Copenhagen does not have an official online archive for master dissertations (don't ask me why!). I have changed the reference to my master thesis to the project site in the Research in Svalbard Database (https://researchinsvalbard.no/project/8729), which I guess is a bit more official than my researchgate site. Hope that works.
Best // Mikkel

[revised manuscript text omitted]

---

## Author Response (AR4)

**Author response to corrections made by Editor regarding resubmission of TC-2020-07**

Dear Editor,

Thanks for the additional corrections. As suggested, we made the following two corrections to our manuscript:

- Old version, line 727: "[…] Canadian high Arctic […]"
    - New version, Line 727:  "[…] Canadian High Arctic […]"
- Old version, line 729: "[…] we argue that […]"
    - New version, line 729: "[…] we suggest that […]"

With great thanks,

Mikkel Toft Hornum on behalf of all authors